# Balancing Gradient and Hessian Queries in Non-Convex Optimization

Deeksha Adil[*]  Brian Bullins[†]  Aaron Sidford[‡]  Chenyi Zhang[‡]

## Abstract

We develop optimization methods which offer new trade-offs between the number of gradient and Hessian computations needed to compute the critical point of a non-convex function. We provide a method that for a twice-differentiable $f \colon \mathbb{R}^d \to \mathbb{R}$ with $L_2$-Lipschitz Hessian, an input initial point with $\Delta$-bounded sub-optimality, and a sufficiently small $\epsilon > 0$, outputs an $\epsilon$-critical point, i.e., a point $x$ such that $\|\nabla f(x)\| \leq \epsilon$, using $\tilde{O}(\Delta L_2^{1/4} n_H^{-1/2} \epsilon^{-9/4})$ queries to a gradient oracle and $n_H$ queries to a Hessian oracle. As a consequence, we obtain an improved gradient query complexity of $\tilde{O}(d^{1/3} L_2^{1/2} \Delta \epsilon^{-3/2})$ in the case of bounded dimension and of $\tilde{O}(\Delta^{3/2} L_2^{3/4} \epsilon^{-9/4})$ in the case where we are allowed only a *single* Hessian query. We obtain these results through a more general algorithm which can handle approximate Hessian computations and recovers known prior state-of-the-art bounds of computing an $\epsilon$-critical point, under the additional assumption that $f$ has an $L_1$-Lipschitz gradient, with $O(\Delta L_2^{1/4} \epsilon^{-7/4})$ gradient queries.

## 1 Introduction

We consider the problem of computing an $\epsilon$-*critical point* of a differentiable function $f \colon \mathbb{R}^d \to \mathbb{R}$, that is $x$ with $\|\nabla f(x)\| \leq \epsilon$, given an initial point $x^{(0)} \in \mathbb{R}^d$ with bounded function error or sub-optimality, $\Delta := f(x^{(0)}) - \inf_{x \in \mathbb{R}^d} f(x)$.[4] This *critical point computation problem*—also referred to as making the gradient norm small [2, 33] or finding stationary points [1]—is a foundational and well-studied optimization problem. It is ubiquitous in machine learning research and has been studied extensively for decades; see e.g., [10, 11] for references.

Obtaining an $\epsilon$-critical point is a natural stopping condition for many optimization methods. For general smooth non-convex functions, guarantees of this type—as opposed to reaching a globally optimal point—may be provably established without incurring exponential dimension dependence in the rates [10, 11]. Furthermore, there are even instances of non-convex objectives, such as regression tasks with non-convex regularization [31] and matrix completion [21], for which reaching what is known as a second-order critical point [1, 9, 26, 27]—which generalize $\epsilon$-critical points—suffices to establish global optimality.

In certain foundational settings, optimal query complexities for critical point computation are known. For example, consider the following simple variant of gradient descent, e.g.,

$$x^{(t+1)} \leftarrow x^{(t)} - \frac{1}{L_1} \nabla f(x^{(t)}) = \operatorname{argmin}_{x \in \mathbb{R}^d} T^1_{x^{(t)}}(x) + \frac{L_1}{2} \|x^{(t)} - x\|^2 \qquad (1)$$

---

[*]Institute for Theoretical Studies, ETH Zürich deeksha.adil@eth-its.ethz.ch

[†]Department of Computer Science, Purdue University bbullins@purdue.edu

[‡]Stanford University {sidford,chenyiz}@stanford.edu

 Preprint available at arXiv:2510.20786.

[4]Throughout the paper we let $\|\cdot\|$ denote the Euclidean or $\ell_2$ norm, and when applied to a square matrix, we let it denote the $\ell_2$-operator norm, i.e., $\|A\| = \sup_{x \in \mathbb{R}^d} \|Ax\|/\|x\|$ for all $A \in \mathbb{R}^{d \times d}$.

39th Conference on Neural Information Processing Systems (NeurIPS 2025).

where $T_{x^{(t)}}^p(x)$ is the $p$th-order Taylor approximation of $f$ evaluated at $x^{(t)}$ and $f$ is $L_1$-smooth, i.e., has an $L_1$-Lipschitz gradient where $\|\nabla f(x) - \nabla f(y)\| \leq L_1\|x - y\|$ for all $x, y \in \mathbb{R}^d$. Eq. (1) computes an $\epsilon$-critical point with at most $O(L_1\Delta\epsilon^{-2})$ iterations [32], and it is known that for sufficiently large $d$, any method (even a randomized one that succeeds with high probability) must compute at least $\Omega(L_1\Delta\epsilon^{-2})$ gradients in the worst case [10]. More broadly, for $f$ that has $L_p$-Lipschitz $p$th-order derivatives define a *$p$th-order oracle* as one that when queried at a point $x$ returns all partial derivatives of $f$ at $x$ up to order $p$. It is known that the method

$$x^{(t+1)} \leftarrow \operatorname{argmin}_{x \in \mathbb{R}^d} T_{x^{(t)}}^p(x) + c_p L_p \|x^{(t)} - x\|^{p+1},$$

for suitable choice of constant $c_p > 0$, solves the problem in $O(c_p L_p^{1/p} \Delta\epsilon^{-(p+1)/p})$ queries and iterations [7], and these rates are optimal in terms of dimension-independent query complexity [10].

Unfortunately, the faster rates for critical point computation obtained via higher-order oracles, e.g., $p$th-order oracles, generally come at a cost. Simply specifying the output of a $p$th-order oracle at a point involves outputting $d^p$ numbers in the worst case, which even for a small constant $p$ can be prohibitively expensive when the problem dimension is large. Correspondingly, there has been extensive research [1, 9, 10, 11, 34] on what rates are obtainable given various assumptions on $f$ and access only to a *gradient oracle*, i.e., an oracle that when queried at $x \in \mathbb{R}^d$ outputs $\nabla f(x)$, the gradient of $f$ at $x$, and a *Hessian oracle*, i.e., an oracle that when queried at $x \in \mathbb{R}^d$ outputs $\nabla^2 f(x)$, the Hessian of $f$ at $x$. One particularly relevant line of work has shown that with only a gradient oracle for $f$ that has $L_1$-Lipschitz gradient *and* $L_2$-Lipschitz Hessian, it is possible to obtain a rate of $O(L_1^{1/2} L_2^{1/4} \Delta\epsilon^{-7/4})$ if $0 < \epsilon \leq \min\{L_1^2 L_2^{-1}, \Delta^{2/3} L_2^{1/3}\}$ [1, 9, 29], and [11] provided an $\Omega(L_1^{3/7} L_2^{2/7} \Delta\epsilon^{-12/7})$ lower bound for this setting. In other words, when the Hessian of the function is also Lipschitz continuous, it is possible to improve upon the $O(L_1\Delta\epsilon^{-2})$ query complexity of gradient descent. Meanwhile, if we further allow for querying Hessian information, this rate can be improved to $O(L_2^{1/2} \Delta\epsilon^{-3/2})$ [34], which is optimal under this stronger oracle model [10, 12].

In this paper, we perform a more fine-grained study of the problem of critical point computation. We ask, *what trade-offs are possible between the number of gradient and Hessian computations that are needed to compute $\epsilon$-critical points for functions with $L_1$-Lipschitz gradients and $L_2$-Lipschitz Hessians?* Our main result is an algorithm which offers a new such trade-off, even when the Hessian is computed only approximately. Furthermore, we show that this result generalizes the bounds of [29] and yields further improvements in dimension-dependent settings.

## 1.1 Our Results

Our main result is a new trade-off in the number of approximate Hessian oracle queries and gradient queries for $f$ needed for $\epsilon$-critical point computation. Here we define the approximate Hessian oracle we consider and provide our main theorem.

**Definition 1** (Approximate Hessian Oracle). *We call a procedure a $\delta$-approximate Hessian oracle for twice differentiable $f : \mathbb{R}^d \to \mathbb{R}$ if when queried at $x \in \mathbb{R}^d$ it outputs symmetric $H_x \in \mathbb{R}^{d \times d}$ such that $\|H_x - \nabla^2 f(x)\| \leq \delta$.*

**Theorem 1** (Main Result). *Let $f : \mathbb{R}^d \mapsto \mathbb{R}$ have $L_1$-Lipschitz gradient and $L_2$-Lipschitz Hessian. There is an algorithm which given any $x^{(0)} \in \mathbb{R}^d$ with $\Delta$-bounded sub-optimality with respect to $f$, positive integer $n_H$, and $0 < \epsilon \leq \min\{L_1^2 L_2^{-1}, \Delta^{2/3} L_2^{1/3}\}$, outputs an $\epsilon$-critical point of $f$ with at most $n_H$ queries to a $\delta$-approximate Hessian oracle and at most*

$$O\left(\frac{\Delta L_2^{1/4} c_\delta^{1/2}}{\epsilon^{7/4}} \cdot \operatorname{poly}\log\left(\frac{c_\ell}{c_\delta}\right)\right)$$

*queries to a gradient oracle for $f$, where*

$$c_\delta := \min\left\{L_1, \delta + \frac{\Delta L_2}{n_H \epsilon}\right\} \quad and \quad c_\ell := \min\left\{L_1, \frac{L_2^2 \Delta^3}{\epsilon^4} + \frac{\Delta \delta^2}{\epsilon^2} + \delta\right\}.$$

As outlined in our overview in Section 2, Theorem 1 follows from a careful combination of Theorem 3 and Corollary 2, which characterize Algorithm 2 and Algorithm 3, respectively.

In the remainder of this section we compare this result to previous studied problems and discuss its implications.

**Generalizing Prior Gradient Methods.** First, we note that Theorem 1 recovers known prior results on gradient-only methods. Observe that $\left\|\nabla^2 f(x)\right\| \le L_1$ if and only if $f$ has an $L_1$-Lipschitz gradient, and consequently, an $L_1$-approximate Hessian oracle for $f$ can be implemented with no queries, by simply outputting the all-zero matrix. In this case, the approximation error is $\delta = L_1$, which leads to $c_\delta = c_\ell = L_1$. Thus, as a corollary of Theorem 1, we obtain a method which computes an $\epsilon$-critical point using $O(L_1^{1/2} L_2^{1/4} \Delta \epsilon^{-7/4})$ queries, which matches the prior best known algorithms in this setting [1, 8, 9, 29].

**Gradient-Hessian Trade-offs for Functions with Unbounded Smoothness.** Interestingly, our results apply even without a bound on $L_1$. In this case, it follows from Theorem 1 that, with exact Hessian queries, i.e., when $\delta = 0$, it is possible to obtain methods that compute an $\epsilon$-critical point with $n_H$ Hessian queries and $\tilde{O}(\Delta^{3/2} L_2^{3/4} n_H^{-1/2} \epsilon^{-9/4})$ gradient queries.[5] Excitingly, this result shows it is possible to compute an $\epsilon$-critical point with only a *single* Hessian query and $\tilde{O}(\Delta^{3/2} L_2^{3/4} \epsilon^{-9/4})$ gradient queries. The previous best algorithms in this setting are essentially due to Doikov et al. [18]. Though their paper studies a different setting, their results seem to imply an $\epsilon$-critical point with $n_H$ Hessian queries and $O(\Delta^2 L_2 n_H^{-2} \epsilon^{-3})$ gradient queries, and so for a single Hessian query, we improve these results, up to polylogarithmic factors, by a factor of $O(\Delta^{1/2} L_2^{1/4} \epsilon^{-3/4})$.

**Dimension-Dependent Critical Point Computation.** As another application of Theorem 1, we obtain improved bounds on the number of gradients needed to compute a critical point of functions where $d$, the dimension, is bounded. Specifically, we note that by a finite differencing argument (e.g., Lemma 3 in [23] for $h = 2\delta d^{-1/2} L_2^{-1}$), a $\delta$-approximate Hessian oracle for $f$ can be implemented with just $2d$ queries to a gradient oracle by approximateing each column of the Hessian using by finite differences and wo gradient queries. Applying this fact with Theorem 1 and optimizing over the choice of $n_H$ yields the following corollary.

**Corollary 1.** *Let $f\colon \mathbb{R}^d \mapsto \mathbb{R}$ have $L_1$-Lipschitz gradient and $L_2$-Lipschitz Hessian. There is a method which given any $x^{(0)} \in \mathbb{R}^d$ with $\Delta$-bounded sub-optimality with respect to $f$ and $0 < \epsilon \le \min\{L_1^2 L_2^{-1}, \Delta^{2/3} L_2^{1/3}\}$, outputs an $\epsilon$-critical point of $f$ with at most $\tilde{O}(d^{1/3} L_2^{1/2} \Delta \epsilon^{-3/2} + d)$ queries to a gradient oracle for $f$.*

Interestingly, while optimal query complexities are known for the low-dimensional $d = 1$ [15] and $d = 2$ [24] cases—the latter following from the more general $O(\max\{2^d, \epsilon^{-2d/(d+2)}\})$ query complexity results of Vavasis [37]—for $d \ge 7$, our results improve, up to polylogarithmic factors, upon the previous state-of-the-art dimension-dependent bound of $O(\sqrt{dL_2} \Delta \epsilon^{-3/2})$ for this problem [18] for any $\epsilon \le \min\{L_1^2 L_2^{-1}, \Delta^{2/3} L_2^{1/3}\}$. In addition, we improve upon the $O(L_1^{1/2} L_2^{1/4} \Delta \epsilon^{-7/4})$ bounds of Li and Lin [29] for $\epsilon \le \min\{O(L_1^{3/2} L_2^{-1} d^{-4/3}), L_1^2 L_2^{-1}, \Delta^{2/3} L_2^{1/3}\}$. This expands the range of $\epsilon$ for which the rate of [29] can be improved over [18], which offers improvement when $\epsilon \le O(L_1^2 L_2^{-1} d^{-2})$; see Table 1 for details. Work by Jiang et al. [25] also provides dimension-dependent results of $O(d^{1/4} L_1^{1/4} L_2^{3/8} \Delta \epsilon^{-13/8})$ gradient queries under the additional assumption that $\epsilon \le \Delta L_2 / L_1$. However, as shown in Appendix B, for any $d \ge 1$ and $\epsilon \le \min\{L_1^2 L_2^{-1}, \Delta^{2/3} L_2^{1/3}, \Delta L_2 L_1^{-1}\}$, this is always at least the minimum of $O(\sqrt{dL_2} \Delta \epsilon^{-3/2} + d)$ [18] and $O(L_1^{1/2} L_2^{1/4} \Delta \epsilon^{-7/4})$ [29].

It remains unclear whether our bound, particularly the $d^{1/3}$ dependence, is asymptotically optimal. Although there are relevant lower bounds in the dimension-independent setting [10, 11], the dimension-dependent complexity of critical point computation is still not well understood: Existing dimension-dependent lower bounds apply only to low-dimensional settings and do not have Hessian Lipschitzness assumptions, see e.g., [15, 24]. The development of tight lower bounds in our regime as an independent and interesting open problem.

While not the focus of the paper, we briefly comment on the computational complexity of our algorithm. Each iteration involves an approximate eigendecomposition step, which may seem more involved than the computation in the classical Newton method, where a dense linear system is solved in each iteration. Nevertheless, in the worst case, it can be implemented in $O(d^\omega)$ time, where $\omega$

---

[5]We use $\tilde{O}(\cdot)$ to hide polylogarithmic factors in $n_H$, $\max\{L_2, L_2^{-1}\}$, $\max\{\Delta, \Delta^{-1}\}$, $\max\{\epsilon, \epsilon^{-1}\}$, and $\delta$.

| Algorithm | # Gradient Queries | Assumption |
|-----------|--------------------|-----------| 
| Vavasis [37] | $O(2^d + \epsilon^{-2d/(d+2)})$ | $L_1$ |
| Li & Lin [29] | $O(L_1^{1/2} L_2^{1/4} \Delta \epsilon^{-7/4})$ | $L_1, L_2$ |
| Nesterov & Polyak [34] | $O(d\sqrt{L_2} \Delta \epsilon^{-3/2})$ | $L_2$ |
| Doikov et al. [18] | $O(\sqrt{dL_2} \Delta \epsilon^{-3/2} + d)$ | $L_2$ |
| Jiang et al. [25] | $O(d^{1/4} L_1^{1/4} L_2^{3/8} \Delta \epsilon^{-13/8})^6$ | $L_1, L_2$ |
| **Corollary 1 (Ours)** | $\tilde{O}(d^{1/3} L_2^{1/2} \Delta \epsilon^{-3/2} + d)$ | $L_2$ |

Table 1: Comparison with previous results in terms of the number of gradient queries needed to reach an $\epsilon$-critical point, i.e., $\|\nabla f(x)\| \leq \epsilon$, under $L_1$-Lipschitz gradient, $L_2$-Lipschitz Hessian assumptions, for $0 < \epsilon \leq \min\{L_1^2 L_2^{-1}, \Delta^{2/3} L_2^{1/3}\}$. This is a standard range of $\epsilon$ to consider as noted in [8]. If $\epsilon > L_1^2 L_2^{-1}$, gradient descent achieves better query complexity. If $\epsilon > \Delta^{2/3} L_2^{1/3}$, our algorithm halts after at most a single iteration and makes at most $\tilde{O}(d^{1/3})$ queries. Up to polylogarithmic factors, our results improve upon [29] when $\epsilon \leq \min\{O(L_1^{3/2} L_2^{-1} d^{-4/3}), L_1^2 L_2^{-1}, \Delta^{2/3} L_2^{1/3}\}$, and upon [18] for any $\epsilon \leq \min\{L_1^2 L_2^{-1}, \Delta^{2/3} L_2^{1/3}\}$.

denotes the matrix multiplication exponent [17]. This is the same as the per-iteration cost of the Newton method.

**Additional Related Work.**  The efficiency of critical point computation has been explored in a wide variety of non-convex optimization contexts and settings, some of which we now highlight. Although this work is concerned with exact gradient information, there has also been significant effort in understanding optimal complexities when instead given access to stochastic oracles [4, 2, 3, 5, 6, 20, 22, 40]. In addition, other works have considered methods based on alternative structural assumptions, such as a type of graded non-convexity [19], a particular spectral decay of the Hessian [30], or, in the case of *non-smooth* non-convex objectives, relaxed notions of approximate critical points [16, 28, 36, 39]. There is also a broader literature related to general Taylor descent algorithms [7, 10], including works that focus on efficient and adaptive methods [13, 14, 34].

## 2   Our Algorithms

Our approach is inspired by the advances in obtaining $\tilde{O}(L_1^{1/2} L_2^{1/4} \Delta \epsilon^{-7/4})$ rates over the last few years [1, 9, 29]. In particular, our algorithm builds on the work of [29], which proves that one can apply accelerated gradient descent with restarts to obtain improved rates. Our algorithm uses a similar method, but works with a norm induced by computations of the approximate Hessian $H$. In particular, we work in the norm induced by $\phi(H)$ where $\phi$ is a carefully chosen function which returns a symmetric matrix. We show that by applying their method in this carefully designed norm and recomputing the Hessian intermittently, we obtain our result.

First, in Section 2.1 we present a variant of accelerated gradient descent (Algorithm 1). This is similar to the algorithms of [29] without restarts, but in the norm induced by $\widehat{H} := \phi(H)$ (Eq. 2). We prove that with a single Hessian computation and a bounded number of gradient computations, the algorithm either finds a critical point or significantly reduces the function value (Theorem 2).

Second, to make use of this result, we either perform negative curvature descent whenever the approximate Hessian $H$ has a sufficiently negative eigenvalue, or apply a restart strategy similar to [29]. (Algorithm 2). Our algorithm additionally keeps track of the movement of the iterates and when the movement is too large, recomputes the approximate Hessian $H$ and the corresponding $\widehat{H}$. In Section 2.2 we analyze Algorithm 2 which essentially obtains our main result up to logarithmic factors (Theorem 3).

Unfortunately, the logarithmic factors for Algorithm 2 depend on $L_1$. Interestingly, we show that there is a fairly generic procedure that allows us to remove this dependence with at most one additional Hessian computation. In Section 2.3, we prove a general reduction that given an algorithm that finds critical points for a function with $L_1$-Lipschitz gradient and $L_2$-Lipschitz Hessian, there is an

---

[6]This result holds under the additional assumption that $\epsilon = O(\Delta L_2/L_1)$. See Appendix B for a detailed comparison between this work and [18, 29].

algorithm that uses one additional Hessian computation and finds a critical point for any function with only an $L_2$-Lipschitz Hessian (Theorem 4). Finally, with all of these tools in hand, we prove our main result, Theorem 1. Several proofs are deferred to the appendix.

## 2.1 Critical or Progress using Approximate Hessians

In this section, we describe the core subroutine of our critical point computation algorithm, `Critical-or-Progress`, which is a version of accelerated gradient descent (Algorithm 1). This procedure either finds an $\epsilon$-critical point or decreases the function value by at least $\tilde{\Omega}(\epsilon^{3/2}/L_2^{1/2})$. Suppose that we are given $H_{x^{(0)}}$, a $\delta$-approximate Hessian at $x^{(0)}$, such that $\|H_{x^{(0)}} - \nabla^2 f(x^{(0)})\| \leq \delta$, and additionally $-2\delta I \preceq H_{x^{(0)}} \preceq L_1 I$. Let the spectral decomposition of $H_{x^{(0)}}$ be

$$H_{x^{(0)}} = \sum_{j=1}^{d} \lambda_j h_j h_j^\top,$$

where $\{h_1, \ldots, h_d\}$ is an orthonormal basis. Define

$$\widehat{H} := \phi(\lambda_j) h_j h_j^\top, \tag{2}$$

where

$$\phi(\lambda) := (32\delta + |\lambda|) \cdot \frac{\lceil \log_2(L_1/\delta) \rceil}{\lceil \log_2(\max\{|\lambda|, 2\delta\}/\delta) \rceil}. \tag{3}$$

We consider an algorithm that performs AGD in the norm induced by $\widehat{H}$, as shown in Algorithm 1. The parameters used in Algorithm 1 are chosen as follows.

$$p_{\max} = \max\{\lceil \log(L_1/\delta) \rceil, 16\}, \quad \tilde{\epsilon} = \frac{\epsilon}{p_{\max}^8}, \quad \eta = \frac{1}{4},$$

$$B = \frac{1}{3}\sqrt{\frac{\tilde{\epsilon}}{L_2}}, \qquad\qquad \theta = \frac{1}{K}, \qquad K = \frac{\sqrt{\delta}}{(\tilde{\epsilon}L_2)^{1/4}} \tag{4}$$

---

**Algorithm 1:** `Critical-or-Progress`

---

1 **Input**: initial iterate $x^{(0)}$, a $\delta$-approximate Hessian $H_{x^{(0)}}$, target accuracy $\epsilon$, gradient Lipschitzness $L_1$, Hessian Lipschitzness $L_2$;

2 Initialize $x^{(-1)} \leftarrow x^{(0)}$;

3 Set $\widehat{H}$ according to (2), and set $\eta, B, \theta, K$ according to (4);

4 **for** $k = 0, \ldots, K$ **do**

5      $y^{(k)} \leftarrow x^{(k)} + (1 - \theta_1)(x^{(k)} - x^{(k-1)})$;

6      $x^{(k+1)} \leftarrow y^{(k)} - \eta \widehat{H}^{-1} \nabla f(y^{(k)})$;

7      **if** $k \sum_{\kappa=0}^{k-1} \|\widehat{H}^{1/2}(x^{(\kappa+1)} - x^{(\kappa)})\|^2 \geq 12\delta p_{\max} B^2$ **then Output** $x^{\text{out}} \leftarrow x^{(k)}$ ;

8 $K_0 \leftarrow \operatorname{argmin}_{\lfloor \frac{3K}{4} \rfloor \leq k \leq K-1} \|\widehat{H}^{1/2}(x^{(k+1)} - x^{(k)})\|$;

9 **Output** $x^{\text{out}} \leftarrow \frac{1}{K_0 + 1 - \lfloor K/2 \rfloor} \sum_{k=\lfloor K/2 \rfloor}^{K_0} y^{(k)}$

---

The main result of this section is the following Theorem 2, which shows that Algorithm 1, using a single query to a $\delta$-approximate Hessian oracle and a bounded number of gradient queries for $f$, either finds an $\epsilon$-critical point or decreases the function value by at least $\tilde{\Omega}(\epsilon^{3/2}/L_2^{1/2})$.

**Theorem 2.** *Let $\delta \leq L_1$, $\epsilon \leq \delta^2/L_2$, and $-\delta I \preceq H_{x^{(0)}} \preceq 2L_1 I$. Using the parameters in (4), Algorithm 1 makes 1 query to a $\delta$-approximate Hessian oracle and at most $K$ queries to a gradient oracle for $f$ and outputs $x^{\text{out}} \in \mathbb{R}^d$ with $\|x^{\text{out}} - x^{(0)}\| \leq 7B$ such that either $x^{\text{out}}$ is $\epsilon$-critical for $f$ or*

$$f(x^{\text{out}}) - f(x^{(0)}) \leq -L_2^{-1/2} \tilde{\epsilon}^{3/2} = -L_2^{-1/2} p_{\max}^{-12} \epsilon^{3/2}.$$

For any $x, y \in \mathbb{R}^d$, define the variables $\hat{x} := \widehat{H}^{1/2} x$, $\hat{y} := \widehat{H}^{1/2} y$, and the function in the norm induced by $\widehat{H}$,

$$\hat{f}(\hat{x}) := f(\widehat{H}^{-1/2} \hat{x}), \tag{5}$$

which satisfies

$$\nabla \hat{f}(\hat{x}) = \widehat{H}^{-1/2} \cdot \nabla f(\widehat{H}^{-1/2} \hat{x}). \tag{6}$$

In this norm induced by $\widehat{H}$, the updates in Line 5 and Line 6 of Algorithm 1 become

$$\begin{aligned}
\hat{y}^{(k)} &\leftarrow \hat{x}^{(k)} + (1 - \theta)(\hat{x}^{(k)} - \hat{x}^{(k-1)}), \\
\hat{x}^{(k+1)} &\leftarrow \hat{y}^{(k)} - \eta \nabla \hat{f}(\hat{y}^{(k)}),
\end{aligned} \tag{7}$$

which are similar to the standard accelerated gradient descent updates.

The proof of Theorem 2 proceeds by analyzing whether the "if condition" in Line 7 is triggered. In the case where the iterates move relatively far in $K$ iterations and the "if condition" is triggered, using a similar proof strategy as [29], we demonstrate that the function value must have decreased by at least $\tilde{\Omega}(L_2^{-1/2} \epsilon^{3/2})$. On the other hand, in the case where the iterates stay relatively close to $x^{(0)}$ and the "if condition" is not triggered, we show that averaging over several iterates yields a point with a small gradient. This part of the analysis is more intricate and relies sensitively on the choice of the matrix $\widehat{H}$. We first establish that $\nabla \hat{f}(\hat{x}^{\text{out}})$, the gradient of the output measured in the norm induced by $\widehat{H}$, is small. However, this does not immediately imply that $\nabla f(x^{\text{out}})$ is small. To bridge this gap, we leverage the specific structure of $\widehat{H}$ defined in (2), where spectral gaps are intentionally introduced through the use of a piecewise function defined in its construction. Using results from matrix perturbation theory, we show that this construction guarantees that the eigenvectors of $\nabla^2 \hat{f}(\hat{x}^{(0)})$ with large eigenvalues is nearly identical to that of $\widehat{H}$ and $\nabla^2 f(x^{(0)})$. Consequently, we can bound the component of $\nabla f(x^{\text{out}})$ in the strongly convex subspace using the corresponding component of $\nabla \hat{f}(\hat{x}^{\text{out}})$, and analyze the component in the non-strongly convex subspace similarly.

## 2.2 Restarted Approximate Hessian AGD

In this section, we present our main algorithm, Algorithm 2. In each iteration, Algorithm 2 maintains a Hessian estimate of $\nabla^2 f(x^{(t)})$ with bounded error. If the estimate has a negative eigenvalue smaller than $-3\tilde{\delta}$, we identify a direction of negative curvature and update along it. Otherwise, we invoke Critical-or-Progress (Algorithm 1) and set its output as the next iterate. We show that if $x^{(t+1)}$ is not an $\epsilon$-critical point, the function value decreases efficiently. As a result, the algorithm finds an $\epsilon$-critical point in a bounded number of iterations.

The Hessian estimate in Algorithm 2 is maintained via lazy updates. It initializes with $x^{(0)}$ as the reference point $\bar{x}$ and sets the Hessian estimate to $H_{x^{(0)}}$. Whenever the current iterate $x^{(t)}$ moves more than a threshold $R$ away from $\bar{x}$, we update the reference point $\bar{x} \leftarrow x^{(t)}$ and the Hessian estimate to $H_{x^{(t)}}$. Since $f$ has an $L_2$-Lipschitz Hessian, the error of the Hessian estimate is bounded by $\max\{2L_1, \delta + L_2 R\}$ in each iteration.

The parameters in Algorithm 2 are chosen as follows.

$$\begin{aligned}
R &\leftarrow \frac{3\Delta}{n_H \epsilon} \log^8 \left( \frac{L_1}{\delta + 3L_2 \Delta / (n_H \epsilon)} + 16 \right), \\
\tilde{\delta} &\leftarrow \min\{\delta + L_2 R, 2L_1\}, \\
\tilde{p} &\leftarrow \max\{\lceil \log(L_1/\tilde{\delta}) \rceil, 16\}.
\end{aligned} \tag{8}$$

The following theorem shows that Algorithm 2 outputs an $\epsilon$-critical point using a bounded number of queries to a $\delta$-approximate Hessian oracle and a gradient oracle.

**Theorem 3.** *Let $f : \mathbb{R}^d \mapsto \mathbb{R}$ have $L_1$-Lipschitz gradients and $L_2$-Lipschitz Hessian. For any $x^{(0)} \in \mathbb{R}^d$ with $\Delta$-bounded sub-optimality with respect to $f$, positive integer $n_H$, and $0 < \epsilon \leq$*

---

**Algorithm 2:** `Restarted-Approx-Hessian-AGD`

---

1 **Input**: initial iterate $x^{(0)}$, accuracy of the approximate Hessian oracle $\delta$, target accuracy $\epsilon$, gradient Lipschitzness $L_1$, Hessian Lipschitzness $L_2$;

2 Initialize $\bar{x} \leftarrow x^{(0)}$, $H \leftarrow H_{\bar{x}}$;

3 **for** $t = 0, 1, 2, \ldots$ **do**

4      **if** $\|\nabla f(x^{(t)})\| \leq \epsilon$ **then** **Output** $x^{(t)}$ ;

5      **if** $\|x^{(t)} - \bar{x}\| \geq R$ **then** $\bar{x} \leftarrow x^{(t)}$ and $H \leftarrow H_{\bar{x}}$ ;

6      **if** $H \prec -3\tilde{\delta} I$ **then**

7          Choose a unit vector $v \in \mathbb{R}^d$ such that $v^\top H v \leq -2\tilde{\delta}$ and $\langle v, \nabla f(x^{(t)}) \rangle \leq 0$;

8          $x^{(t+1)} \leftarrow x^{(t)} + Rv$

9      **else** $x^{(t+1)} \leftarrow$ `Critical-or-Progress`$\left(x^{(t)}, H, 4\tilde{\delta}, \epsilon, L_1, L_2\right)$ ;

---

$\min\{L_1^2 L_2^{-1}, \Delta^{2/3} L_2^{1/3}\}$, *Algorithm 2 outputs an $\epsilon$-critical point with at most $n_H$ queries to a $\delta$-approximate Hessian oracle and at most*

$$\frac{2\Delta L_2^{1/4} c_\delta^{1/2}}{\epsilon^{7/4}} \cdot \log^{18}\left(\frac{L_1}{c_\delta} + 16\right)$$

*queries to a gradient oracle for $f$, where*

$$c_\delta := \min\left\{L_1, \delta + \frac{\Delta L_2}{n_H \epsilon}\right\}.$$

The following describes the change in function value in every iteration of Algorithm 2 and is useful for proving Theorem 3.

**Lemma 1.** *Suppose $\epsilon \leq L_1^2/L_2$. In each iteration $t$ of Algorithm 2 before it terminates, we have*

$$f(x^{(t+1)}) - f(x^{(t)}) \leq -\tilde{p}^{-12}\sqrt{\epsilon^3/L_2}$$

*if $x^{(t+1)}$ is not $\epsilon$-critical for $f$, where we denote $\tilde{p} = \max\{\lceil \log(L_1/\tilde{\delta})\rceil, 16\}$.*

*Proof of Theorem 3.* By Lemma 1, before Algorithm 2 terminates, in each iteration the function value decreases by at least $\tilde{p}^{-12}\sqrt{\epsilon^3/L_2}$. Hence, Algorithm 2 terminates in at most $\tilde{p}^{12}\Delta\sqrt{L_2/\epsilon^3}$ iterations. From Theorem 2, the number of gradient queries in each iteration is at most

$$K = \frac{\tilde{p}^2\sqrt{\tilde{\delta}}}{(L_2\epsilon)^{1/4}}.$$

Therefore, the total number of gradient queries is at most

$$
\begin{aligned}
\tilde{p}^{12}\Delta\sqrt{\frac{L_2}{\epsilon^3}} \cdot \frac{\tilde{p}^2\sqrt{\tilde{\delta}}}{(L_2\epsilon)^{1/4}} &= \frac{2\tilde{p}^{18}\Delta L_2^{1/4}}{\epsilon^{7/4}}\sqrt{\min\left\{L_1, \delta + \frac{\Delta L_2}{n_H\epsilon}\right\}} \\
&= \frac{2\Delta L_2^{1/4}}{\epsilon^{7/4}}\sqrt{\min\left\{L_1, \delta + \frac{\Delta L_2}{n_H\epsilon}\right\}} \cdot \log^{18}\left(\frac{L_1}{\tilde{\delta}} + 16\right) \\
&\leq \frac{2\Delta L_2^{1/4}}{\epsilon^{7/4}}\sqrt{\min\left\{L_1, \delta + \frac{\Delta L_2}{n_H\epsilon}\right\}} \cdot \log^{18}\left(\frac{L_1}{c_\delta} + 16\right)
\end{aligned}
$$

given that $c_\delta < \tilde{\delta}$. Furthermore, from Theorem 2 and (4), in every iteration $t$ we have

$$\|x^{(t+1)} - x^{(t)}\| \leq \frac{7}{3\tilde{p}^4}\sqrt{\frac{\epsilon}{L_2}}.$$

Therefore, the number of Hessian computations is at most

$$\left\lceil \tilde{p}^{12}\Delta\sqrt{\frac{L_2}{\epsilon^3}} \cdot \frac{7}{3\tilde{p}^4}\sqrt{\frac{\epsilon}{L_2}} \cdot \frac{1}{R}\right\rceil,$$

which is at most $n_H$ when using the values of the parameters in (8). $\qquad \square$

## 2.3 Removing the $L_1$-Lipschitz Gradient Assumption

In this section, we give an algorithm that computes an $\epsilon$-critical point for functions with $L_2$-Lipschitz Hessians but no guarantees on the Lipschitzness of the gradient. In order to do so, we first provide a general reduction in Algorithm 3. Given an algorithm `Alg` for computing an approximate critical point for a function with $L_1$-Lipschitz gradient and $L_2$-Lipschitz Hessian, Algorithm 3 can compute an approximate critical point for any function with only $L_2$-Lipschitz Hessian while maintaining the number of queries to gradient oracle and using at most one more query to the Hessian oracle, if the output of `Alg` is not very far away from the initial iterate and has bounded suboptimality.

For any symmetric $M \in \mathbb{R}^{d \times d}$ with spectral decomposition $M = \sum_{j \in [d]} \lambda_j v_j v_j^\top$, we denote $M^\dagger := \sum_{j \in [d]} \mathbb{I}\{\lambda_j \neq 0\} \cdot \lambda_j^{-1} v_j v_j^\top$ to be its Moore–Penrose pseudoinverse, where $\mathbb{I}\{\cdot\}$ is the indicator function.

---

**Algorithm 3:** `Reduction-To-Unbounded-Hessian`

---

1 **Input**: initial iterate $x^{(0)}$, target accuracy $\epsilon$, Hessian Lipschitzness $L_2$;
2 Choose an eigenvalue threshold $\ell$;
3 Initialize $H \leftarrow H_{x^{(0)}}$;
4 $x^{\text{out}} \leftarrow \texttt{Alg}(f_{\leq \ell}, \ell, L_2, \delta, \Delta, \epsilon)$;
5 **Output** $y \leftarrow x^{\text{out}} - (\Pi_{>\ell} H \Pi_{>\ell})^\dagger \nabla f(x^{\text{out}})$

---

Given a $\delta$-approximate Hessian $H_{x^{(0)}}$ with spectral decomposition $H_{x^{(0)}} = \sum_{j=1}^d \lambda_j h_j h_j^\top$, where $\{h_1, \ldots, h_d\}$ is an orthonormal basis, we partition the indices into two sets based on the eigenvalue threshold $\ell$ picked in Line 2.

$$\mathcal{S}_{\leq \ell} := \{i : |\lambda_i| \leq \ell\}, \quad \mathcal{S}_{>\ell} := \{i : |\lambda_i| > \ell\}, \tag{9}$$

and define corresponding projection matrices

$$\Pi_{\leq \ell} := \sum_{i \in \mathcal{S}_{>\ell}} h_i h_i^\top, \quad \Pi_{>\ell} := \sum_{i \in \mathcal{S}_{\leq \ell}} h_i h_i^\top. \tag{10}$$

Moreover, we define the restricted function

$$f_{\leq \ell} := f\big(x^{(0)} + \Pi_{\leq \ell}(x - x^{(0)})\big). \tag{11}$$

Algorithm 3 begins by invoking the subroutine `Alg` to find an $\epsilon/2$-stationary point $x^{\text{out}}$ of the restricted function $f_{\leq \ell}$. It then performs a Newton step in the subspace spanned by $h_j : j \in \mathcal{S}_{>\ell}$:

$$y \leftarrow x^{\text{out}} - (\Pi_{>\ell} H_{x^{(0)}} \Pi_{>\ell})^\dagger \nabla f(x^{\text{out}})$$

using $H_{x^{(0)}}$ as the approximation of $\nabla^2 f(x^{\text{out}})$. Since $\|x^{\text{out}} - x^{(0)}\| \leq R_{\text{out}}$, we show that $H_{x^{(0)}}$ remains a sufficiently accurate estimate. Moreover, $\ell$ is chosen large enough so that the size of this Newton step is small. Otherwise it would incur a function value decrease larger than $\Delta_{\text{out}}$, contradicting to the suboptimality condition of $x^{\text{out}}$. As a result, $\nabla f(y)$ is close to $\nabla T_{x^{\text{out}}}^2(y)$ since $f$ is $L_2$-Hessian Lipschitz, and the latter is at most $\epsilon/2$. Formally, we prove the following:

**Theorem 4.** *Let* $\texttt{Alg}(f_{\leq L_1}, L_1, L_2, \delta, \Delta, \epsilon)$ *be a procedure that, for any function* $f_{\leq L_1} : \mathbb{R}^d \to \mathbb{R}$ *with* $L_1$-*Lipschitz gradient,* $L_2$-*Lipschitz Hessian, and* $\Delta$-*bounded suboptimality, uses*

- $n_H$ *queries to a* $\delta$-*approximate Hessian oracle for* $f_{\leq L_1}$, *and,*
- $n_g(L_1, L_2, \delta, \Delta, \epsilon)$ *queries to a gradient oracle for* $f_{\leq L_1}$,

*and returns an* $\epsilon/2$-*critical point* $x^{\text{out}}$ *satisfying* $\|x^{\text{out}} - x^{(0)}\| \leq R_{\text{out}}$ *and* $f(x^{\text{out}}) - \inf_z f(z) \leq \Delta_{\text{out}}$. *Then, for any* $f$ *with* $L_2$-*Lipschitz Hessian and* $\Delta$-*bounded suboptimality, any* $0 < \epsilon \leq \min\{L_1^2 L_2^{-1}, \Delta^{2/3} L_2^{1/3}\}$, *and any* $\ell$ *that satisfies*

$$\ell \geq \max\left\{ \frac{800\Delta}{\epsilon^2}(L_2 R_{\text{out}} + \delta)^2, \frac{48 L_2 \Delta_{\text{out}}}{\epsilon}, 24\Delta_{\text{out}}^{1/3} L_2^{2/3}, 2\delta \right\}. \tag{12}$$

*Algorithm 3 returns an* $\epsilon$-*critical point using*

- $n_H + 1$ *queries to a $\delta$-approximate Hessian oracle for $f$, and,*
- $n_g(\ell, L_2, \delta, \Delta, \epsilon)$ *queries to a gradient oracle for $f$.*

The proof of Theorem 4 is in Appendix G. Corollary 2 is obtained by running Algorithm 3 where we use Algorithm 2 as the subroutine `Alg`. The proof of Corollary 2 is in Appendix F.

**Corollary 2.** *Let $f\colon \mathbb{R}^d \mapsto \mathbb{R}$ $L_2$-Lipschitz Hessian. Given any $x^{(0)} \in \mathbb{R}^d$ with $\Delta$-bounded sub-optimality with respect to $f$, any positive integer $n_H \geq 1$, and $0 < \epsilon \leq \Delta^{2/3} L_2^{1/3}$, Algorithm 3 using Algorithm 2 as the subroutine `Alg` outputs an $\epsilon$-critical point of $f$ with at most $n_H$ queries to a $\delta$-approximate Hessian oracle and*

$$O\left(\frac{\Delta L_2^{1/4}}{\epsilon^{7/4}}\sqrt{\delta + \frac{\Delta L_2}{n_H\epsilon}} \cdot \mathrm{poly}\log\left(\frac{1}{c_\delta}\left(\frac{L_2^2\Delta^3}{\epsilon^4} + \frac{\Delta\delta^2}{\epsilon^2} + \delta\right)\right)\right)$$

*queries to a gradient oracle for $f$.*

## 2.4  Proof of Theorem 1

*Proof.* We consider the following algorithm. When

$$L_1 \leq \frac{L_2^2\Delta^3}{\epsilon^4} + \frac{\Delta\delta}{\epsilon^2} + \delta,$$

we run Algorithm 2, which outputs an $\epsilon$-critical point with at most $n_H$ queries to a $\delta$-approximate Hessian oracle and at most

$$\frac{2\Delta L_2^{1/4}c_\delta^{1/2}}{\epsilon^{7/4}} \cdot \log^{18}\left(\frac{L_1}{c_\delta} + 16\right) = O\left(\frac{\Delta L_2^{1/4}c_\delta^{1/2}}{\epsilon^{7/4}} \cdot \mathrm{poly}\log\left(\frac{c_\ell}{c_\delta}\right)\right).$$

Otherwise, we run Algorithm 3 using Algorithm 2 as the subroutine `Alg`, which outputs an $\epsilon$-critical point of $f$ with at most $n_H$ queries to a $\delta$-approximate Hessian oracle and

$$O\left(\frac{\Delta L_2^{1/4}}{\epsilon^{7/4}}\sqrt{\delta + \frac{\Delta L_2}{n_H\epsilon}} \cdot \mathrm{poly}\log\left(\frac{c_\ell}{c_\delta}\right)\right) = O\left(\frac{\Delta L_2^{1/4}c_\delta^{1/2}}{\epsilon^{7/4}} \cdot \mathrm{poly}\log\left(\frac{c_\ell}{c_\delta}\right)\right)$$

queries to a gradient oracle for $f$, where the last equality follows from the fact that

$$L_1 > \frac{L_2^2\Delta^3}{\epsilon^4} + \frac{\Delta\delta}{\epsilon^2} + \delta \geq \delta + \frac{\Delta L_2}{n_H\epsilon}$$

and thus $c_\delta = \delta + \frac{\Delta L_2}{n_H\epsilon}$. We conclude by noticing that the number of Hessian queries in both cases is $n_H$, while the number of gradient queries in both cases is

$$O\left(\frac{\Delta L_2^{1/4}c_\delta^{1/2}}{\epsilon^{7/4}} \cdot \mathrm{poly}\log\left(\frac{c_\ell}{c_\delta}\right)\right).$$

$\square$

# 3  Conclusion

In this paper we provided new algorithms for computing critical points of twice differentiable functions using gradient and $\delta$-approximate Hessian queries. We provided a general result which offered new trade-offs between the number of queries made to these oracles to compute an $\epsilon$-critical point of functions with $L_1$-Lipschitz gradients and $L_2$-Lipschitz Hessians given an intial point of bounded suboptimality. As a consequence of this result, for sufficiently small $\epsilon$, we recovered known bounds on the number gradient queries needed to compute critical points and improved upon the prior state-of-the-art bounds in the case where the function is either of bounded dimension or when a single Hessian query is available.

Though our work provides new algorithms and tools for critical point computation, there are several limitations to the result. First, this work is primarily theoretical, no practical implementation or

experiments are provided, and in certain cases our bounds incur multiple logarithmic factors. Second, many functions in practice may be non-differentiable or of a large enough size that computing the Hessian is prohibitively expensive, limiting the direct applicability of the results. Third, though there are interesting relevant lower bounds [10, 11], it is unknown whether our query complexities are asymptotically optimal. Each of these limitations suggests natural open problems and directions for future work, e.g., finding practical applications of our techniques and seeking improved upper and lower bounds for the problems we consider. However, we hope this paper provides valuable tools for this potential future work.

## Acknowledgments

Thank you to anonymous reviewers for their feedback. Deeksha Adil is supported by Dr. Max Rössler, the Walter Haefner Foundation and the ETH Zürich Foundation. Aaron Sidford was supported in part by NSF Grant CCF-1955039. Chenyi Zhang was supported in part by the Shoucheng Zhang graduate fellowship.

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

# A Overview and Notation of the Appendix

The appendix is organized as follows. Appendix B provides a comparison between [25] and prior works. Appendix C presents key properties of the matrix $\widehat{H}$ defined in (2). Appendix D collects useful results from matrix perturbation theory, which are used to analyze the spectral properties of the matrices appearing in Algorithm 1. Then, we give the analysis of Algorithm 1 and prove Theorem 2 in Appendix E. The analyses of Algorithm 2 and Algorithm 3 are given in Appendix F and Appendix G, respectively.

**Notation.** For any symmetric matrix $M \in \mathbb{R}^{d \times d}$ with spectral decomposition $M = \sum_{j \in [d]} \lambda_j v_j v_j^\top$, and any subset $\mathcal{S} \in \mathbb{R}$, we denote

$$\Pi_{\mathcal{S}}(M) := \sum_{j \in [d]} \mathbb{I}\{\lambda_j \in \mathcal{S}\} v_j v_j^\top$$

to be the projector onto the eigenspace of $M$ with eigenvalues in $\mathcal{S}$. For any matrix $M \in \mathbb{R}^{d_1 \times d_2}$, we use $\lambda_{\min}(M)$ to denote its smallest eigenvalue. For any two invertible symmetric matrices $M, N \in \mathbb{R}^{d \times d}$ that commute, i.e., $MN = NM$, denote $\frac{M}{N} := MN^{-1} = N^{-1}M$. Moreover, we define

$$p_{\max} := \max\{\lceil \log_2(L_1/\delta) \rceil, 16\}, \quad p_{\min} := 16.$$

as in (4), where $L_1$ is the gradient Lipschitzness of $f$. For any positive integer $p \in \mathbb{N}^+$ we define

$$l_p := \frac{1}{p_{\max}} \cdot \frac{p+1}{1 + 2^{-(p-5)}}, \quad r_p := \frac{1}{p_{\max}} \cdot \frac{p+1}{1 + 2^{-(p-4)}}. \tag{13}$$

and

$$\xi_p := \frac{\sqrt{p}}{2^{p/2} p_{\max}}, \quad \bar{l}_p := l_p - \xi_p, \quad \forall p \in \mathbb{N}^+. \tag{14}$$

# B Comparison Between [25] and Prior Works

In this section, we provide a comparison between [25] and prior works [18] and [29]. In particular, [25] achieves a query complexity of $O(d^{1/4} L_1^{1/4} L_2^{3/8} \Delta \epsilon^{-13/8})$, with an implicit requirement that $\epsilon \leq O(\Delta L_2/L_1)$. This condition arises from the fact that, in the first displayed equation of [25, Section C.3], the third term in the bracket needs to dominate the first two. Substituting the choices of $D$ and $M$ specified in [25, Theorem 4.1] yields the corresponding inequality. In the following lemma, we show that within the parameter regime $\epsilon \leq \min\{L_1^2/L_2, \Delta^{2/3} L_2^{1/3}, \Delta L_2/L_1\}$, the query complexity of [25] is asymptotically at least the minimum of [18] and [29].

**Lemma 2.** *For $d, L_1, L_2, \Delta, \epsilon > 0$ satisfying*

$$\epsilon \leq \min \left\{ L_1^2/L_2, \Delta^{2/3} L_2^{1/3}, \Delta L_2/L_1 \right\},$$

*we have*

$$\min\{\sqrt{dL_2}\Delta \epsilon^{-3/2} + d, L_1^{1/2} L_2^{1/4} \Delta \epsilon^{-7/4}\} = O(d^{1/4} L_1^{1/4} L_2^{3/8} \Delta \epsilon^{-13/8}).$$

*Proof.* Define

$$A := \sqrt{dL_2}\Delta \epsilon^{-3/2}, \qquad B := L_1^{1/2} L_2^{1/4} \Delta \epsilon^{-7/4}, \qquad G := d^{1/4} L_1^{1/4} L_2^{3/8} \Delta \epsilon^{-13/8}$$

and let $\phi = B/A$ and $u = d/A$. Since $G = \sqrt{AB}$,

$$\frac{\min\{A + d, B\}}{G} = \min \left\{ \frac{1 + u}{\sqrt{\phi}}, \sqrt{\phi} \right\}.$$

where

$$\phi = \frac{L_1^{1/2}}{\sqrt{d} L_2^{1/4}} \epsilon^{-1/4}, \quad u = \frac{\sqrt{d}}{\sqrt{L_2}\Delta} \epsilon^{3/2} = \frac{L_1^{1/2}}{L_2^{3/4}\Delta} \frac{\epsilon^{5/4}}{\phi}.$$

Using the condition that $\epsilon \le \min\left\{ L_1^2/L_2, \Delta^{2/3} L_2^{1/3}, \Delta L_2/L_1 \right\}$, we obtain

$$u \le \frac{1}{\phi} \min\{C_1, C_2, C_3\}, \qquad C_1 = \frac{L_1^3}{L_2^2 \Delta}, \quad C_2 = L_1^{1/2} \Delta^{-1/6} L_2^{-1/3}, \quad C_3 = \Delta^{1/4} L_2^{1/2} L_1^{-3/4}.$$

When $L_1 = \Delta^{1/3} L_2^{2/3}$, we have $C_1 = C_2 = C_3 = 1$. Moreover, since $C_1$ and $C_2$ increase with $L_1$ while $C_3$ decreases,

$$\min\{C_1, C_2, C_3\} \le 1$$

for all $L_1, L_2, \Delta$, and thus $u \le 1/\phi$. Hence,

$$\frac{\min\{A + d, B\}}{G} \le \min_{\phi \in \mathbb{R}} \left\{ \frac{1 + 1/\phi}{\sqrt{\phi}}, \ \sqrt{\phi} \right\} \le \sqrt{\frac{1 + \sqrt{5}}{2}},$$

which gives

$$\min\{\sqrt{dL_2}\Delta\epsilon^{-3/2} + d, L_1^{1/2} L_2^{1/4} \Delta\epsilon^{-7/4}\} = O(d^{1/4} L_1^{1/4} L_2^{3/8} \Delta\epsilon^{-13/8}).$$

$\square$

# C Properties of $\widehat{H}$

In this section, we present several properties of the matrix $\widehat{H}$ used in Algorithm 1. Let us recall the setting of Section 2.1: We are given $H_{x^{(0)}}$, a $\delta$-approximate Hessian at $x^{(0)}$ that satisfies $\|H_{x^{(0)}} - \nabla^2 f(x^{(0)})\| \le \delta$ and $-3\delta I \preceq H_{x^{(0)}} \preceq L_1 I$. Let the spectral decomposition be

$$H_{x^{(0)}} = \sum_{j=1}^{d} \lambda_j h_j h_j^\top,$$

where $\{h_1, \ldots, h_d\}$ is an orthonormal basis. Then, $\widehat{H}$ is defined as

$$\widehat{H} := \phi(\lambda_j) h_j h_j^\top,$$

as in (2), where

$$\phi(\lambda) := (32\delta + |\lambda|) \cdot \frac{\lceil \log_2(L_1/\delta) \rceil}{\lceil \log_2(\max\{|\lambda|, 2\delta\}/\delta) \rceil}.$$

as in (3).

**Lemma 3.** *For any positive integer $p \in \mathbb{N}^+$ and any $\lambda$ such that $2^p\delta < |\lambda| \le 2^{p+1}\delta$, we have*

$$l_p < |\lambda| \cdot \phi(\lambda)^{-1} \le r_p.$$

*Proof.* Given that $\phi$ is symmetric with respect to 0, without loss of generality, we assume $\lambda > 0$. Then, we have

$$\frac{\lambda}{\phi(\lambda)} = \frac{1}{1 + 32\delta\lambda^{-1}} \cdot \frac{\lceil \log_2(\lambda/\delta) \rceil}{p_{\max}} \le \frac{1}{p_{\max}} \cdot \frac{p+1}{1 + 2^{-(p-4)}} = r_p,$$

and

$$\frac{\lambda}{\phi(\lambda)} = \frac{1}{1 + 32\delta\lambda^{-1}} \cdot \frac{\lceil \log_2(\lambda/\delta) \rceil}{p_{\max}} > \frac{1}{p_{\max}} \cdot \frac{p+1}{1 + 2^{-(p-5)}} = l_p.$$

$\square$

**Lemma 4.** $\lambda\phi(\lambda)^{-1}$ *is monotonically increasing for* $\lambda \in [-L_1, L_1]$.

*Proof.* Note that $\lambda \cdot \phi(\lambda)^{-1}$ is a odd function. Therefore, it suffices to prove that

$$\lambda \cdot \phi(\lambda)^{-1} = \frac{\lambda}{(32\delta + \lambda)} \cdot \frac{\lceil \log_2(\max\{\lambda, 2\delta\}/\delta) \rceil}{\lceil \log_2(L_1/\delta) \rceil}.$$

is increasing for $\lambda \in [0, L_1]$, where it suffices to check monotonicity of

$$\frac{\lambda \cdot \lceil \log_2(\max\{\lambda, 2\delta\}/\delta) \rceil}{32\delta + \lambda}.$$

First observe that the function is strictly increasing for any $\lambda \in [0, 2\delta)$. For any positive integer $p \geq 0$ and any $\lambda \in [2^p\delta, 2^{p+1}\delta)$, we have

$$\frac{\lambda \cdot \lceil \log_2(\max\{\lambda, 2\delta\}/\delta) \rceil}{32\delta + \lambda} = \frac{\lambda \cdot p}{32\delta + \lambda}.$$

where

$$\frac{d}{d\lambda}\left(\frac{\lambda \cdot p}{32\delta + \lambda}\right) = p \cdot \frac{(32\delta + \lambda) - \lambda}{(32\delta + \lambda)^2} = \frac{p \cdot 32\delta}{(32\delta + \lambda)^2} > 0.$$

Thus, the function is strictly increasing on each interval $[2^p\delta, 2^{p+1}\delta)$. Moreover, at each boundary point $\lambda = 2^p\delta$, the function increases from $\frac{2^p\delta \cdot (p-1)}{32\delta + 2^p\delta}$ to $\frac{2^p\delta \cdot p}{32\delta + 2^p\delta}$. We can thus conclude that $\lambda\phi^{-1}(\lambda)$ is monotonically increasing. $\qquad \square$

**Lemma 5.** *The function $\phi$ defined in (3) satisfies*

$$\min_{\lambda \in (-\infty, +\infty)} \phi(\lambda) \geq 12\delta p_{\max}.$$

*Proof.* Since $\phi$ is symmetric with respect to 0, we only need to consider values of $\lambda$ in $[0, +\infty)$. We analyze the function in pieces. For $\lambda \in [0, 2\delta)$, we have

$$\phi(\lambda) = (32\delta + \lambda) \cdot \frac{p_{\max}}{\lceil \log_2(2\delta/\delta) \rceil} = (32\delta + \lambda) \cdot p_{\max},$$

which is minimized at $\lambda = 0$, yielding

$$\phi(0) = 32\delta \cdot p_{\max}.$$

For $\lambda \in [2^k\delta, 2^{k+1}\delta)$ with $k \geq 1$, we have

$$\phi(\lambda) = (32\delta + \lambda) \cdot \frac{p_{\max}}{\lceil \log_2(\lambda/\delta) \rceil} = (32\delta + \lambda) \cdot \frac{p_{\max}}{k},$$

which is minimized at $\lambda = 2^k\delta$, giving

$$\phi(2^k\delta) = \frac{32 + 2^k}{k} \cdot \delta p_{\max}.$$

Hence, we have

$$\min_{\lambda \in (-\infty, +\infty)} \phi(\lambda) = \min\left\{32, \min_{k \geq 1} \frac{32 + 2^k}{k}\right\} \cdot \delta p_{\max} = 12\delta p_{\max}.$$

$\qquad \square$

**Corollary 3.** *For any symmetric $H \in \mathbb{R}^{d \times d}$, the matrix $\widehat{H}$ defined in (2) satisfies $\widehat{H} \succeq 12\delta p_{\max} \cdot I$ and $\widehat{H}^{-1} \preceq (12\delta p_{\max})^{-1} I$.*

## D   Tools from Matrix Perturbation Theory and Extensions

In this section, we present several useful results from matrix perturbation theory that characterize how the eigenspaces of symmetric matrices change under perturbations. Recall the setting from Section 2.1, where we are given a $\delta$-approximate Hessian $H_{x^{(0)}}$ at the point $x^{(0)}$, satisfying $\|H_{x^{(0)}} - \nabla^2 f(x^{(0)})\| \leq \delta$ and $-3\delta I \preceq H_{x^{(0)}} \preceq L_1 I$. Throughout this section, we denote $G := \nabla^2 f(x^{(0)})$. Our goal is to show that the spectra of $\widehat{H}$ and $\widehat{H}^{-1/2} G \widehat{H}^{-1/2}$ can be partitioned into $\Theta(p_{\max})$ contiguous subsets such that, for any $p_{\min} \leq p \leq p_{\max}$, the principal angles between the eigenspaces spanned by the $p$-th spectral subsets of $\widehat{H}$ and $\widehat{H}^{-1/2} G \widehat{H}^{-1/2}$ are small. We also derive several additional properties of these eigenspaces, which are used in the analysis in Appendix E.

### D.1 Definition of Projectors

This part includes the definition of a series of projectors that we use to analyze the spectrum of $\widehat{H}$ and $\widehat{H}^{-1/2}G\widehat{H}^{-1/2}$. Define

$$\widehat{\Pi}_p^+ := \Pi_{(l_p, L_1]}(H_{x^{(0)}}) = \Pi_{(l_p, \infty)}(H_{x^{(0)}}), \quad \widehat{\Pi}_p^- := \Pi_{[-2\delta, l_{p+1}]}(H_{x^{(0)}}), \tag{15}$$

and

$$\widehat{\Pi}_p := \Pi_{(l_p, l_{p+1}]}(H_{x^{(0)}}). \tag{16}$$

Similarly, we define

$$\overline{\Pi}_p^+ := \Pi_{(\bar{l}_p, \infty)}(\widehat{H}^{-1/2}G\widehat{H}^{-1/2}), \quad \overline{\Pi}_p^- := \Pi_{(-\infty, \bar{l}_{p+1}]}(\widehat{H}^{-1/2}G\widehat{H}^{-1/2}), \tag{17}$$

and

$$\overline{\Pi}_p := \Pi_{(\bar{l}_p, \bar{l}_{p+1}]}(\widehat{H}^{-1/2}G\widehat{H}^{-1/2}). \tag{18}$$

Furthermore, we define

$$\overline{\Pi}_{\text{base}} := \overline{\Pi}_{p-1}^-. \tag{19}$$

### D.2 The Davis-Kahan Theorem

In this subsection, we present the celebrated Davis–Kahan theorem, and provide an equivalent formulation that we use in our paper.

**Definition 2** (Principal angles of subspaces). *Let $V, \widetilde{V} \subset \mathbb{R}^n$ be $k$-dimensional subspaces. The principal angles $\theta_1(V, \widetilde{V}), \dots \theta_k(V, \widetilde{V})$ between $V$ and $\widetilde{V}$ are defined recursively by*

$$(v_j, \tilde{v}_j) := \arg \max_{v_j \in V, \tilde{v}_j \in \widetilde{V}} \langle v_j, \tilde{v}_j \rangle, \quad \theta_j(V, \widetilde{V}) := \arccos\langle v_j, \tilde{v}_j \rangle,$$

*subject to the constraint*

$$\|v_j\| = \|\tilde{v}_j\| = 1, \quad \langle v_j, v_i \rangle = 0, \quad \langle \tilde{v}_j, \tilde{v}_i \rangle = 0, \qquad \forall 1 \le i < j.$$

**Lemma 6** (Davis-Kahan Theorem, see e.g., Theorem 1 of [38]). *Let $M, \widetilde{M} \in \mathbb{R}^{d \times d}$ be two symmetric matrices satisfying $\|M - \widetilde{M}\| \le \xi$ for some $\xi > 0$. For any $a < b$, we use $\mathcal{S} = \{v_1, \dots, v_k\}$ and $\widetilde{\mathcal{S}} = \{\tilde{v}_1, \dots, \tilde{v}_{\tilde{k}}\}$ to denote the set of normalized eigenvectors of $M$ and $\widetilde{M}$ associated with eigenvalues contained in the interval $[a, b]$ and $[a - \xi, b + \xi]$, respectively, and denote*

$$V := \text{span}(\mathcal{S}), \quad \widetilde{V} := \text{span}(\widetilde{\mathcal{S}}).$$

*Then, if the remaining eigenvalues of $M$ lie outside the interval $[a - \gamma, b + \gamma]$, we have $k = \tilde{k}$ and*

$$\|\sin(\Theta(V, \widetilde{V}))\| \le \frac{\xi}{\gamma},$$

*where*

$$\sin\Theta(V, \widetilde{V}) := \text{diag}\left(\sin\theta_1(V, \widetilde{V}), \dots, \sin\theta_k(V, \widetilde{V})\right)^\top \tag{20}$$

**Lemma 7** (Theorem I.5.5 of [35]). *Let $V, \widetilde{V} \subset \mathbb{R}^n$ be $k$-dimensional subspaces and let $\Pi, \widetilde{\Pi}$ be their projectors. Then, we have*

$$\|\Pi - \widetilde{\Pi}\| = \|\sin\Theta(V, \widetilde{V})\| = \sin\theta_1(V, \widetilde{V}),$$

*where $\sin\Theta(V, \widetilde{V})$ is defined in (20).*

**Lemma 8** (Equivalent form of the Davis-Kahan Theorem). *Let $M, \widetilde{M} \in \mathbb{R}^{d \times d}$ be two symmetric matrices satisfying $\|M - \widetilde{M}\| \le \xi$ for some $\xi > 0$. For any $a < b$ and $\gamma > \xi$, if there are no eigenvalues of $M$ in intervals $[a - \gamma, a)$ and $(b, b + \gamma]$, we have $k = \tilde{k}$ and*

$$\left\|\Pi_{[a,b]}(M) - \Pi_{[a-\gamma, b+\gamma]}(\widetilde{M})\right\| = \left\|\sum_{j=1}^k v_j v_j^\top - \sum_{j=1}^k \tilde{v}_j \tilde{v}_j^\top\right\| \le \frac{\xi}{\gamma}.$$

*Proof.* The proof follows by combining Lemma 6 and Lemma 7. $\qquad\square$

Intuitively, Lemma 6 and Lemma 8 states that a small perturbation to a symmetric matrix leads to only a small change in its eigenspaces, provided the corresponding eigenvalues are well separated from the other eigenvalues.

## D.3 Properties of $\widehat{\Pi}_p^+$ and $\overline{\Pi}_p^+$

**Proposition 1.** *For any positive integer $p \geq 5$, we have*

$$\left\| \widehat{\Pi}_p^+ - \overline{\Pi}_p^+ \right\| \leq 2^{2-p/2}\sqrt{p}.$$

Before proving Proposition 1, we first present the following two lemmas.

**Lemma 9.** *For any positive integer $p \geq 5$, denote*

$$\widetilde{G} := \widehat{\Pi}_p^+ H_{x^{(0)}} \widehat{\Pi}_p^+ + (I - \widehat{\Pi}_p^+)G(I - \widehat{\Pi}_p^+)$$

*with $\widehat{\Pi}_p^+$ defined in (15). Then, we have*

$$\left\| \widehat{H}^{-1/2}(G - \widetilde{G})\widehat{H}^{-1/2} \right\| \leq \xi_p,$$

*where $\widehat{H}$ is defined in (2).*

*Proof.* Given that $H_{x^{(0)}}$ and $\widehat{H}$ have the same set of eigenvectors, we have

$$\widehat{\Pi}_p^+ \widehat{H}(I - \widehat{\Pi}_p^+) = (I - \widehat{\Pi}_p^+)\widehat{H}\widehat{\Pi}_p^+ = 0.$$

Then, $\widetilde{G}$ can also be written as

$$\widetilde{G} = H_{x^{(0)}} - (I - \widehat{\Pi}_p^+)H_{x^{(0)}}(I - \widehat{\Pi}_p^+) + (I - \widehat{\Pi}_p^+)G(I - \widehat{\Pi}_p^+) = H_{x^{(0)}} + (I - \widehat{\Pi}_p^+)(G - H_{x^{(0)}})(I - \widehat{\Pi}_p^+),$$

which implies

$$
\begin{aligned}
G - \widetilde{G} &= (G - H_{x^{(0)}}) - (I - \widehat{\Pi}_p^+)(G - H_{x^{(0)}})(I - \widehat{\Pi}_p^+) \\
&= \widehat{\Pi}_p^+(G - H_{x^{(0)}}) + (G - H_{x^{(0)}})\widehat{\Pi}_p^+ - \widehat{\Pi}_p^+(G - H_{x^{(0)}})\widehat{\Pi}_p^+.
\end{aligned}
$$

Therefore,

$$
\begin{aligned}
\left\| \widehat{H}^{-1/2}(G - \widetilde{G})\widehat{H}^{-1/2} \right\| &\leq \left\| \widehat{H}^{-1/2}\widehat{\Pi}_p^+(G - H_{x^{(0)}})\widehat{H}^{-1/2} \right\| + \left\| \widehat{H}^{-1/2}(G - H_{x^{(0)}})\widehat{\Pi}_p^+\widehat{H}^{-1/2} \right\| \\
&\quad + \left\| \widehat{H}^{-1/2}\widehat{\Pi}_p^+(G - H_{x^{(0)}})\widehat{\Pi}_p^+\widehat{H}^{-1/2} \right\|.
\end{aligned}
$$

From the definitions of $\widehat{H}$ and $\phi$ in equations (2) and (3), it follows that $\phi(\lambda) \geq 12\delta$ for all $\lambda \in [-L, L]$, yielding

$$\left\| \widehat{H}^{-1/2} \right\| \leq \sqrt{\frac{1}{12\delta p_{\max}}} < \frac{1}{3}\sqrt{\frac{1}{\delta p_{\max}}}$$

and

$$\left\| \widehat{\Pi}_p^+\widehat{H}^{-1/2} \right\| = \left\| \widehat{H}^{-1/2}\widehat{\Pi}_p^+ \right\| \leq \sqrt{\frac{p}{2^p \delta p_{\max}}}.$$

Combining these bounds gives

$$\left\| \widehat{H}^{-1/2}\widehat{\Pi}_p^+(G - H_{x^{(0)}})\widehat{H}^{-1/2} \right\| = \left\| \widehat{H}^{-1/2}(G - H_{x^{(0)}})\widehat{\Pi}_p^+\widehat{H}^{-1/2} \right\| \leq \frac{\sqrt{p}}{3 \times 2^{p/2}p_{\max}},$$

and

$$\left\| \widehat{H}^{-1/2}\widehat{\Pi}_p^+(G - H_{x^{(0)}})\widehat{\Pi}_p^+\widehat{H}^{-1/2} \right\| \leq \frac{\sqrt{p}}{2^p p_{\max}}.$$

Thus, we can conclude that

$$\left\| \widehat{H}^{-1/2}(G - \widetilde{G})\widehat{H}^{-1/2} \right\| \leq \frac{p}{2^{p/2}p_{\max}} = \xi_p.$$

$\qquad\qquad\qquad\qquad\qquad\qquad\qquad\qquad\qquad\qquad\qquad\qquad\qquad\qquad\qquad\qquad\qquad\qquad\square$

**Lemma 10.** *For any positive integer $p \geq 5$, we have*

$$\widehat{\Pi}_p^+ \widehat{H}^{-1/2} H_{x^{(0)}} \widehat{H}^{-1/2}\widehat{\Pi}_p^+ \succeq l_p \cdot \widehat{\Pi}_p^+$$

*and*

$$\left\| (I - \widehat{\Pi}_p^+)\widehat{H}^{-1/2}G\widehat{H}^{-1/2}(I - \widehat{\Pi}_p^+) \right\| \leq \frac{1}{12p_{\max}} + r_{p-1},$$

*where $\widehat{\Pi}_p^+$ and $\widehat{H}$ are defined in (15) and (2), respectively.*

*Proof.* Given that $H_{x^{(0)}}$ and $\widehat{H}$ share the same eigenvectors, the minimal eigenvalue of $\widehat{H}^{-1/2} H_{x^{(0)}} \widehat{H}^{-1/2}$ restricted to the subspace projected by $\widehat{\Pi}_p^+$ is bounded below by

$$\min_{\lambda > 2^p \delta} \lambda \phi^{-1}(\lambda) > l_p,$$

as established in Lemma 3. Consequently, we have

$$\widehat{\Pi}_p^+ \widehat{H}^{-1/2} H_{x^{(0)}} \widehat{H}^{-1/2} \widehat{\Pi}_p^+ \succeq l_p \cdot \widehat{\Pi}_p^+.$$

By the triangle inequality, it follows that

$$\left\| (I - \widehat{\Pi}_p^+) \widehat{H}^{-1/2} G \widehat{H}^{-1/2} (I - \widehat{\Pi}_p^+) \right\| \leq \left\| (I - \widehat{\Pi}_p^+) \widehat{H}^{-1/2} H_{x^{(0)}} \widehat{H}^{-1/2} (I - \widehat{\Pi}_p^+) \right\|$$
$$+ \left\| (I - \widehat{\Pi}_p^+) \widehat{H}^{-1/2} (G - H_{x^{(0)}}) \widehat{H}^{-1/2} (I - \widehat{\Pi}_p^+) \right\|.$$

By the definition of $\widehat{H}$ in (2) and Lemma 3, we have the bound

$$\left\| (I - \widehat{\Pi}_p^+) \widehat{H}^{-1/2} H_{x^{(0)}} \widehat{H}^{-1/2} (I - \widehat{\Pi}_p^+) \right\| \leq \frac{1}{p_{\max}} \cdot \frac{p}{1 + 2^{-(p-5)}} = r_{p-1}.$$

Moreover, since

$$\left\| \widehat{H}^{-1/2} \right\| \leq \sqrt{\frac{1}{12\delta p_{\max}}},$$

we obtain

$$\left\| (I - \widehat{\Pi}_p^+) \widehat{H}^{-1/2} (G - H_{x^{(0)}}) \widehat{H}^{-1/2} (I - \widehat{\Pi}_p^+) \right\| \leq \left\| \widehat{H}^{-1/2} (G - H_{x^{(0)}}) \widehat{H}^{-1/2} \right\|$$
$$\leq \left\| \widehat{H}^{-1/2} \right\|^2 \cdot \left\| G - H_{x^{(0)}} \right\| \leq \frac{1}{12 p_{\max}}.$$

By combining these bounds we can conclude that

$$\left\| (I - \widehat{\Pi}_p^+) \widehat{H}^{-1/2} G \widehat{H}^{-1/2} (I - \widehat{\Pi}_p^+) \right\| \leq \frac{1}{12 p_{\max}} + r_{p-1}.$$

$\square$

*Proof of Proposition 1.* Define

$$\widetilde{G} := \widehat{\Pi}_p^+ H_{x^{(0)}} \widehat{\Pi}_p^+ + (I - \widehat{\Pi}_p^+) G (I - \widehat{\Pi}_p^+).$$

Since $\widehat{H}$ and $\widehat{\Pi}_p^+$ share the same set of eigenvectors, in the basis $\{\hat{h}_1, \ldots, \hat{h}_d\}$, where the eigenvectors are arranged in descending order according to their eigenvalues, the matrix $\widehat{H}^{-1/2} \widetilde{G} \widehat{H}^{-1/2}$ takes the following block-diagonal form:

$$\widehat{H}^{-1/2} \widetilde{G} \widehat{H}^{-1/2} = \begin{bmatrix} \widehat{\Pi}_p^+ \widehat{H}^{-1/2} H_{x^{(0)}} \widehat{H}^{-1/2} \widehat{\Pi}_p^+ & 0 \\ 0 & (I - \widehat{\Pi}_p^+) \widehat{H}^{-1/2} G \widehat{H}^{-1/2} (I - \widehat{\Pi}_p^+) \end{bmatrix}.$$

By Lemma 10, the top left block has eigenvalues bounded below by $l_p$, while the bottom right block has eigenvalues bounded above by $r_{p-1} + \frac{1}{12\lceil \log_2(L/\delta) \rceil}$, with eigenvalue gap

$$l_p - r_{p-1} - \frac{1}{12 p_{\max}} \geq \frac{1}{3 p_{\max}} > \xi_{p-1}.$$

Additionally, by Lemma 9, we have the bound

$$\left\| \widehat{H}^{-1/2} G \widehat{H}^{-1/2} - \widehat{H}^{-1/2} \widetilde{G} \widehat{H}^{-1/2} \right\| \leq \xi_p.$$

We can then conclude by applying Lemma 8, which gives

$$\left\| \widehat{\Pi}_p^+ - \overline{\Pi}_p^+ \right\| \leq 2^{2-p/2} \sqrt{p}.$$

$\square$

### D.4   Properties of $\widehat{\Pi}_p^-, \widehat{\Pi}_p$ and $\overline{\Pi}_p^-, \overline{\Pi}_p$

**Lemma 11.** *For any positive integer $p \geq 5$, we have*
$$\left\|\widehat{\Pi}_p^- - \overline{\Pi}_p^-\right\| \leq 2^{2-p/2}\sqrt{p}.$$

*Proof.* Given that
$$\widehat{\Pi}_p^- + \widehat{\Pi}_{p+1}^+ = \overline{\Pi}_p^- + \overline{\Pi}_{p+1}^+ = I,$$
we have
$$\left\|\widehat{\Pi}_p^- - \overline{\Pi}_p^-\right\| = \left\|\widehat{\Pi}_p^+ - \overline{\Pi}_{p+1}^+\right\| \leq 2^{2-p/2}.$$
$\square$

**Lemma 12.** *For any positive integer $p > 1$, we have*
$$\left\|\widehat{\Pi}_p - \overline{\Pi}_p\right\| \leq 2^{3-p/2}\sqrt{p}.$$

*Proof.* Given that
$$\widetilde{\Pi}_{p-1}^- + \widehat{\Pi}_p + \widehat{\Pi}_{p+1}^+ = I, \quad \overline{\Pi}_p^- + \overline{\Pi}_p + \overline{\Pi}_{p+1}^+ = I,$$
we have
$$\left\|\widehat{\Pi}_p - \overline{\Pi}_p\right\| \leq \left\|\widetilde{\Pi}_{p-1}^- - \overline{\Pi}_{p-1}^-\right\| + \left\|\widehat{\Pi}_{p+1}^+ - \overline{\Pi}_{p+1}^+\right\| \leq 2^{2-p/2}\sqrt{p} + 2^{2-p/2}\sqrt{p} \leq 2^{3-p/2}\sqrt{p}.$$
$\square$

**Lemma 13.** *For any positive integer $p \geq 5$ and any $\hat{v} \in \mathbb{R}^d$, we have*
$$\left\|\widehat{H}^{1/2}\widehat{\Pi}_{p+1}^+\overline{\Pi}_p\hat{v}\right\| \leq \sqrt{\delta} \cdot p_{\max} \cdot \left\|\overline{\Pi}_p\hat{v}\right\|$$
*and*
$$\left\|\widehat{H}^{1/2}\widehat{\Pi}_{p+1}^+\overline{\Pi}_p^-\hat{v}\right\| \leq \sqrt{\delta} \cdot p_{\max} \cdot \left\|\overline{\Pi}_p^-\hat{v}\right\|.$$

*Proof.* For the first inequality, observe that
$$\left\|\widehat{H}^{1/2}\widehat{\Pi}_{p+1}^+\overline{\Pi}_p\hat{v}\right\| \leq \sum_{q=p+1}^{p_{\max}} \left\|\widehat{H}^{1/2}\widehat{\Pi}_q\overline{\Pi}_p\hat{v}\right\|,$$
where for each $q > p$ we have
$$\left\|\widehat{H}^{1/2}\widehat{\Pi}_q\overline{\Pi}_p\hat{v}\right\| \leq 2^{2-q/2}\sqrt{q}\left\|\widehat{H}^{1/2}\widehat{\Pi}_q\right\| \cdot \left\|\overline{\Pi}_p\hat{v}\right\| \leq 2^{2-q/2}\sqrt{q\phi(2^{q+1}\delta)} \cdot \left\|\overline{\Pi}_p\hat{v}\right\|$$
by Lemma 11. Summing over $q$ gives
$$\sum_{q=p+1}^{p_{\max}} 2^{2-p/2} \cdot \phi(2^{q+1}\delta) = \sum_{q=p+1}^{p_{\max}} 2^{2-q/2} \cdot \sqrt{(32 + 2^q) \cdot \delta p_{\max}}$$
$$\leq \sqrt{\delta} \cdot p_{\max}.$$
Therefore,
$$\left\|\widehat{H}^{1/2}\widehat{\Pi}_{p+1}^+\overline{\Pi}_p\hat{v}\right\| \leq \sqrt{\delta} \cdot p_{\max} \cdot \left\|\overline{\Pi}_p\hat{v}\right\|.$$
The proof of the second inequality is similar. Note that
$$\left\|\widehat{H}^{1/2}\widehat{\Pi}_{p+1}^+\overline{\Pi}_p^-\hat{v}\right\| \leq \sum_{q=p+1}^{p_{\max}} \left\|\widehat{H}^{1/2}\widehat{\Pi}_q\overline{\Pi}_p^-\hat{v}\right\|,$$
where for each $q > p$ we have
$$\left\|\widehat{H}^{1/2}\widehat{\Pi}_q\overline{\Pi}_p^-\hat{v}\right\| \leq 2^{2-q/2}\sqrt{q}\left\|\widehat{H}^{1/2}\widehat{\Pi}_q\right\| \cdot \left\|\overline{\Pi}_p^-\hat{v}\right\| \leq 2^{2-p/2}\sqrt{q\phi(2^{q+1}\delta)} \cdot \left\|\overline{\Pi}_p^-\hat{v}\right\|$$
by Lemma 11. Summing over $q$ gives
$$\sum_{q=p+1}^{p_{\max}} 2^{2-p/2} \cdot \phi(2^{q+1}\delta) = \sum_{q=p+1}^{p_{\max}} 2^{2-q/2} \cdot \sqrt{(32 + 2^q) \cdot \delta p_{\max}} \leq \sqrt{\delta} \cdot p_{\max}.$$
Therefore,
$$\left\|\widehat{H}^{1/2}\widehat{\Pi}_{p+1}^+\overline{\Pi}_p^-\hat{v}\right\| \leq \sqrt{\delta} \cdot p_{\max} \cdot \left\|\overline{\Pi}_p^-\hat{v}\right\|.$$
$\square$

**Proposition 2.** *For any positive integer $p \geq 5$ and any $\hat{v} \in \mathbb{R}^d$, we have*

$$\left\|\widehat{H}^{1/2}\overline{\Pi}_p\hat{v}\right\| \leq 2^{p/2}\sqrt{\delta} \cdot p_{\max} \cdot \left\|\overline{\Pi}_p\hat{v}\right\|$$

*and*

$$\left\|\widehat{H}^{1/2}\overline{\Pi}_p\hat{v}\right\| \geq \left\|\overline{\Pi}_p\hat{v}\right\|$$

*Moreover, we have*

$$\left\|\widehat{H}^{-1/2}\overline{\Pi}_p\hat{v}\right\| \geq \frac{1}{2\sqrt{\phi(2^{p+1}\delta)}}\left\|\overline{\Pi}_p\hat{v}\right\|.$$

*Proof.* For the first inequality, note that

$$\overline{\Pi}_p\hat{v} = \widehat{\Pi}_p^-\overline{\Pi}_p\hat{v} + \sum_{q=p+1}^{p_{\max}} \widehat{\Pi}_q\overline{\Pi}_p\hat{v},$$

which gives

$$\left\|\widehat{H}^{1/2}\overline{\Pi}_p\hat{v}\right\| \leq \left\|\widehat{H}^{1/2}\widehat{\Pi}_p^-\overline{\Pi}_p\hat{v}\right\| + \sum_{q=p+1}^{p_{\max}} \left\|\widehat{H}^{1/2}\widehat{\Pi}_q\overline{\Pi}_p\hat{v}\right\|,$$

where

$$\left\|\widehat{H}^{1/2}\widehat{\Pi}_p^-\overline{\Pi}_p\hat{v}\right\| \leq \left\|\widehat{H}^{1/2}\widehat{\Pi}_p^-\right\| \cdot \left\|\overline{\Pi}_p\hat{v}\right\| \leq \sqrt{\phi(2^{p+1}\delta)} \cdot \left\|\overline{\Pi}_p\hat{v}\right\|$$

and

$$\sum_{q=p+1}^{p_{\max}} \left\|\widehat{H}^{1/2}\widehat{\Pi}_q\overline{\Pi}_p\hat{v}\right\| \leq \sqrt{\delta} \cdot p_{\max} \cdot \left\|\overline{\Pi}_p\hat{v}\right\|$$

by Lemma 13. Hence,

$$\left\|\widehat{H}^{1/2}\overline{\Pi}_p\hat{v}\right\| \leq 2^{p/2}\sqrt{\delta} \cdot p_{\max} \cdot \left\|\overline{\Pi}_p\hat{v}\right\|.$$

The second inequality follows from

$$\left\|\widehat{H}^{1/2}\overline{\Pi}_p\hat{v}\right\| \geq \left\|\widehat{H}^{1/2}\widehat{\Pi}_p\overline{\Pi}_p\hat{v}\right\| \geq \left\|\widehat{H}^{1/2}\widehat{\Pi}_p\right\| \cdot \left\|\widehat{\Pi}_p\overline{\Pi}_p\hat{v}\right\| \geq \frac{\sqrt{\phi(2^p\delta)}}{2}\left\|\overline{\Pi}_p\hat{v}\right\|$$

As for the third inequality,

$$\overline{\Pi}_p\hat{v} = \widehat{\Pi}_p\overline{\Pi}_p\hat{v} + \widehat{\Pi}_{p-1}^-\overline{\Pi}_p\hat{v} + \widehat{\Pi}_{p+1}^+\overline{\Pi}_p\hat{v},$$

which leads to

$$\left\|\widehat{H}^{-1/2}\overline{\Pi}_p\hat{v}\right\| \geq \left\|\widehat{H}^{-1/2}\widehat{\Pi}_p\overline{\Pi}_p\hat{v}\right\| \geq \frac{1}{\sqrt{\phi(2^{p+1}\delta)}} \cdot \left\|\widehat{\Pi}_p\overline{\Pi}_p\hat{v}\right\|,$$

where

$$\left\|\widehat{\Pi}_p\overline{\Pi}_p\hat{v}\right\| \geq \left\|\overline{\Pi}_p\overline{\Pi}_p\hat{v}\right\| - \left\|(\widehat{\Pi}_p - \overline{\Pi}_p)\overline{\Pi}_p\hat{v}\right\| \geq \frac{1}{2}\left\|\overline{\Pi}_p\hat{v}\right\|,$$

by which we can conclude that

$$\left\|\widehat{H}^{-1/2}\overline{\Pi}_p\hat{v}\right\| \geq \frac{1}{2\sqrt{\phi(2^{p+1}\delta)}}\left\|\overline{\Pi}_p\hat{v}\right\|.$$

$\square$

## D.5  The Connection between $\widehat{H}^{1/2}\hat{v}$ and $\widehat{H}^{1/2}\overline{\Pi}_p\hat{v}$

In this subsection, we prove the following result.

**Proposition 3.** *For any positive integer $p \geq p_{\min}$ and any $\hat{v} \in \mathbb{R}^d$, we have*

$$\|\widehat{H}^{1/2}\hat{v}\| \geq \frac{2^{-5}}{1 + 2^{-p/4}\sqrt{p_{\max}}} \cdot \|\widehat{H}^{1/2}\overline{\Pi}_p\hat{v}\|. \tag{21}$$

Denote $v_p := \widehat{H}^{1/2}\overline{\Pi}_p\hat{v}$, $v_p^- := \widehat{H}^{1/2}\overline{\Pi}_{p-1}^-\hat{v}$, and $v_p^+ := \widehat{H}^{1/2}\overline{\Pi}_{p+1}^+\hat{v}$. Then, we have $\widehat{H}^{1/2}\hat{v} = v_p + v_p^- + v_p^+$. If $v_p = 0$, (21) holds directly. Hence, we only need to prove the case where $v_p \neq 0$.

The following inequalitie are useful for proving Proposition 3.

**Lemma 14.** *For any integer $p \geq 5$ and any $\hat{v} \in \mathbb{R}^d$, the following inequalities hold.*

1. $\|\widehat{H}^{1/2}\widehat{\Pi}_p\overline{\Pi}_p\hat{v}\| \geq \sqrt{\phi(2^p\delta)}\|\overline{\Pi}_p\hat{v}\|/2$;

2. $\|\widehat{H}^{1/2}\widehat{\Pi}_{p-1}^-\overline{\Pi}_p\hat{v}\| \leq 2^{2-p/2}\sqrt{p\phi(2^p\delta)}\|\overline{\Pi}_p\hat{v}\|$;

3. $\|\widehat{H}^{1/2}\widehat{\Pi}_p\overline{\Pi}_{p-1}^-\hat{v}\| \leq 2^{2-p/2}\sqrt{p\phi(2^{p+1}\delta)}\|\overline{\Pi}_{p-1}^-\hat{v}\|$;

4. $\|\widehat{H}^{1/2}\widehat{\Pi}_{p-1}^-\overline{\Pi}_{p-1}^-\hat{v}\| \geq \sqrt{\delta \cdot p_{\max}} \cdot \|\overline{\Pi}_{p-1}^-\hat{v}\|$;

5. $\|\widehat{H}^{1/2}\widehat{\Pi}_{p+1}^+\overline{\Pi}_{p-1}^-\hat{v}\| \leq \sqrt{\delta \cdot p_{\max}} \cdot \|\overline{\Pi}_{p-1}^-\hat{v}\|$;

6. $\|\widehat{H}^{1/2}\widehat{\Pi}_p\overline{\Pi}_{p+1}^+\hat{v}\| \leq 2^{2-p/2}\sqrt{p\phi(2^{p+1}\delta)}\|\overline{\Pi}_{p+1}^+\hat{v}\|$;

7. $\|\widehat{H}^{1/2}\widehat{\Pi}_{p-1}^-\overline{\Pi}_{p+1}^+\hat{v}\| \leq 2^{1-p/2}\sqrt{p\phi(2^p\delta)}\|\overline{\Pi}_{p+1}^+\hat{v}\|$;

8. $\|\widehat{H}^{1/2}\widehat{\Pi}_{p+1}^+\overline{\Pi}_{p+1}^+\hat{v}\| \geq \sqrt{\phi(2^{p+1}\delta)}\|\overline{\Pi}_{p+1}^+\hat{v}\|/2$.

*Proof.* Proof of the first entry:

$$\|\widehat{H}^{1/2}\widehat{\Pi}_p\overline{\Pi}_p\hat{v}\| \geq \sqrt{\phi(2^p\delta)}\|\widehat{\Pi}_p\overline{\Pi}_p\hat{v}\|$$
$$\geq \sqrt{\phi(2^p\delta)}\big(\|\overline{\Pi}_p\overline{\Pi}_p\hat{v}\| - \|(\widehat{\Pi}_p - \overline{\Pi}_p)\overline{\Pi}_p\hat{v}\|\big) \geq \frac{\sqrt{\phi(2^p\delta)}}{2}\|\overline{\Pi}_p\hat{v}\|$$

by Lemma 12.

Proof of the second entry:

$$\|\widehat{H}^{1/2}\widehat{\Pi}_{p-1}^-\overline{\Pi}_p\hat{v}\| \leq \|\widehat{H}^{1/2}\widehat{\Pi}_{p-1}^-\| \cdot \|\widehat{\Pi}_{p-1}^-\overline{\Pi}_p\hat{v}\|$$
$$\leq \sqrt{\phi(2^p\delta)}\|\widehat{\Pi}_{p-1}^-\overline{\Pi}_p\hat{v}\| \leq 2^{2-p/2}\sqrt{p \cdot \phi(2^p\delta)}\|\overline{\Pi}_p\hat{v}\|$$

by Lemma 11.

Proof of the third entry:

$$\|\widehat{H}^{1/2}\widehat{\Pi}_p\overline{\Pi}_{p-1}^-\hat{v}\| \leq \|\widehat{H}^{1/2}\widehat{\Pi}_p\| \cdot \|\widehat{\Pi}_p\overline{\Pi}_{p-1}^-\hat{v}\|$$
$$\leq \sqrt{\phi(2^{p+1}\delta)} \cdot \|\widehat{\Pi}_p\overline{\Pi}_{p-1}^-\hat{v}\|$$
$$\leq 2^{2-p/2}\sqrt{p\phi(2^{p+1}\delta)}\|\overline{\Pi}_{p-1}^-\hat{v}\|,$$

by Lemma 12.

Proof of the fourth entry:

$$\|\widehat{H}^{1/2}\widehat{\Pi}_{p-1}^-\overline{\Pi}_{p-1}^-\hat{v}\| \geq \frac{1}{\|\widehat{H}^{-1/2}\|}\|\widehat{\Pi}_{p-1}^-\overline{\Pi}_{p-1}^-\hat{v}\|$$
$$\geq \sqrt{12\delta\lceil\log_2(L/\delta)\rceil}\big(\|\overline{\Pi}_{p-1}^-\hat{v}\| - \|(\widehat{\Pi}_{p-1}^- - \overline{\Pi}_{p-1}^-)\overline{\Pi}_{p-1}^-\hat{v}\|\big)$$
$$\geq \sqrt{\delta\lceil\log_2(L/\delta)\rceil}\|\overline{\Pi}_{p-1}^-\hat{v}\|$$

by Lemma 11.

Proof of the fifth entry:

$$\|\widehat{H}^{1/2}\widehat{\Pi}_{p+1}^{+}\overline{\Pi}_{p-1}^{-}\hat{v}\| \leq \sqrt{\delta} \cdot p_{\max} \cdot \|\overline{\Pi}_{p-1}^{-}\hat{v}\|$$

by Lemma 13.

Proof of the sixth entry:

$$\|\widehat{H}^{1/2}\widehat{\Pi}_{p}\overline{\Pi}_{p+1}^{+}\hat{v}\| \leq \|\widehat{H}^{1/2}\widehat{\Pi}_{p}\| \cdot \|(\widehat{\Pi}_{p} - \overline{\Pi}_{p})\overline{\Pi}_{p+1}^{+}\hat{v}\|$$
$$\leq 2^{2-p/2}\sqrt{p\phi(2^{p+1}\delta)}\|\overline{\Pi}_{p+1}^{+}\hat{v}\|$$

by Lemma 12.

Proof of the seventh entry:

$$\|\widehat{H}^{1/2}\widehat{\Pi}_{p-1}^{-}\overline{\Pi}_{p+1}^{+}\hat{v}\| \leq \|\widehat{H}^{1/2}\widehat{\Pi}_{p-1}^{-}\| \cdot \|(\widehat{\Pi}_{p-1}^{-} - \overline{\Pi}_{p-1}^{-})\overline{\Pi}_{p+1}^{+}\hat{v}\|$$
$$\leq 2^{1-p/2}\sqrt{p\phi(2^{p}\delta)}\|\overline{\Pi}_{p+1}^{+}\hat{v}\|$$

by Lemma 11.

Proof of the eighth entry:

$$\|\widehat{H}^{1/2}\widehat{\Pi}_{p+1}^{+}\overline{\Pi}_{p+1}^{+}\hat{v}\| \geq \sqrt{\phi(2^{p+1}\delta)}\big(\|\overline{\Pi}_{p+1}^{+}\hat{v}\| - \|(\widehat{\Pi}_{p+1}^{+} - \overline{\Pi}_{p+1}^{+})\overline{\Pi}_{p+1}^{+}\hat{v}\|\big)$$
$$\geq \frac{\sqrt{\phi(2^{p+1}\delta)}}{2}\|\overline{\Pi}_{p+1}^{+}\hat{v}\|$$

by Proposition 1. $\qquad\square$

**Lemma 15.** *For any positive integer $p \geq p_{\min}$ and any $\hat{v} \in \mathbb{R}^d$, we have*

$$\big\|\widehat{H}^{1/2}\widehat{\Pi}_{p}^{-}\overline{\Pi}_{p}^{-}\hat{v}\big\| \geq \sin\Big(\frac{\pi}{p_{\min}}\Big) \cdot \big\|\widehat{H}^{1/2}\widehat{\Pi}_{p}^{-}\overline{\Pi}_{p}\hat{v}\big\|.$$

*Proof.* Note that

$$\widehat{H}^{1/2}\widehat{\Pi}_{p}^{-}\overline{\Pi}_{p}^{-}\hat{v} = \widehat{H}^{1/2}\widehat{\Pi}_{p}^{-}\overline{\Pi}_{p}\hat{v} + \widehat{H}^{1/2}\widehat{\Pi}_{p}^{-}\overline{\Pi}_{p-1}^{-}\hat{v}$$

If either $\widehat{H}^{1/2}\widehat{\Pi}_{p}^{-}\overline{\Pi}_{p}\hat{v}$ or $\widehat{H}^{1/2}\widehat{\Pi}_{p}^{-}\overline{\Pi}_{p-1}^{-}\hat{v}$ equals 0, the inequality holds directly. Otherwise, by entries 1 through 4 of Lemma 14, we have

$$\arccos\left(\frac{\langle\widehat{H}^{1/2}\widehat{\Pi}_{p}^{-}\overline{\Pi}_{p}\hat{v}, \widehat{H}^{1/2}\widehat{\Pi}_{p}^{-}\overline{\Pi}_{p-1}^{-}\hat{v}\rangle}{\|\widehat{H}^{1/2}\widehat{\Pi}_{p}^{-}\overline{\Pi}_{p}\hat{v}\| \cdot \|\widehat{H}^{1/2}\widehat{\Pi}_{p}^{-}\overline{\Pi}_{p-1}^{-}\hat{v}\|}\right) \geq \frac{\pi}{16},$$

which leads to

$$\big\|\widehat{H}^{1/2}\widehat{\Pi}_{p}^{-}\overline{\Pi}_{p}^{-}\hat{v}\big\| \geq \sin(\pi/16)\big\|\widehat{H}^{1/2}\widehat{\Pi}_{p}^{-}\overline{\Pi}_{p}\hat{v}\big\|.$$

$\qquad\square$

**Lemma 16.** *For any positive integer $p \geq 5$ and any $\hat{v} \in \mathbb{R}^d$, we have*

$$\big\|\widehat{H}^{1/2}\widehat{\Pi}_{p+1}^{+}\overline{\Pi}_{p}^{-}\hat{v}\big\| \leq \big\|\widehat{H}^{1/2}\widehat{\Pi}_{p}^{-}\overline{\Pi}_{p}^{-}\hat{v}\big\|.$$

*Proof.* By Lemma 13 we have

$$\big\|\widehat{H}^{1/2}\widehat{\Pi}_{p+1}^{+}\overline{\Pi}_{p}^{-}\hat{v}\big\| \leq \sqrt{\delta} \cdot p_{\max}\big\|\overline{\Pi}_{p}^{-}\hat{v}\big\|.$$

Moreover, by entry 4 of Lemma 14 we have

$$\big\|\widehat{H}^{1/2}\widehat{\Pi}_{p}^{-}\overline{\Pi}_{p}^{-}\hat{v}\big\| \geq \sqrt{\delta \cdot p_{\max}} \cdot \big\|\overline{\Pi}_{p}^{-}\hat{v}\big\|,$$

which leads to

$$\big\|\widehat{H}^{1/2}\widehat{\Pi}_{p+1}^{+}\overline{\Pi}_{p}^{-}\hat{v}\big\| \leq \big\|\widehat{H}^{1/2}\widehat{\Pi}_{p}^{-}\overline{\Pi}_{p}^{-}\hat{v}\big\|.$$

$\qquad\square$

**Lemma 17.** *For any positive integer $p \geq 5$ and $\hat{v} \in \mathbb{R}^d$, we have*

$$\left\|\widehat{H}^{1/2}\overline{\Pi}_p\hat{v}\right\| \leq \left(2 + 2^{-p/4}\sqrt{p_{\max}}\right)\left\|\widehat{H}^{1/2}\widehat{\Pi}_p\overline{\Pi}_p\hat{v}\right\|.$$

*Proof.*

$$\left\|\widehat{H}^{1/2}\overline{\Pi}_p\hat{v}\right\| \leq \left\|\widehat{H}^{1/2}\widehat{\Pi}_p\overline{\Pi}_p\hat{v}\right\| + \left\|\widehat{H}^{1/2}\widehat{\Pi}_{p-1}^-\overline{\Pi}_p\hat{v}\right\| + \left\|\widehat{H}^{1/2}\widehat{\Pi}_{p+1}^+\overline{\Pi}_p\hat{v}\right\|,$$

where

$$\left\|\widehat{H}^{1/2}\widehat{\Pi}_{p-1}^-\overline{\Pi}_p\hat{v}\right\| \leq 2^{2-p/2}\sqrt{p\phi(2^p\delta)}\left\|\overline{\Pi}_p\hat{v}\right\| \leq 2^{2-(p-\log p)/2}\left\|\widehat{H}^{1/2}\widehat{\Pi}_p\overline{\Pi}_p\hat{v}\right\|$$

by the first two inequalities in Lemma 14, and

$$\begin{aligned}
\left\|\widehat{H}^{1/2}\widehat{\Pi}_{p+1}^+\overline{\Pi}_p\hat{v}\right\| &\leq \sqrt{\delta} \cdot p_{\max} \cdot \left\|\overline{\Pi}_p\hat{v}\right\| \\
&\leq 2\sqrt{\frac{\delta}{\phi(2^p\delta)}} \cdot p_{\max} \cdot \left\|\widehat{H}^{1/2}\widehat{\Pi}_p\overline{\Pi}_p\hat{v}\right\| \\
&\leq 2^{-p/4}\sqrt{p_{\max}}\left\|\widehat{H}^{1/2}\widehat{\Pi}_p\overline{\Pi}_p\hat{v}\right\|
\end{aligned}$$

by Proposition 2 and the first inequality of Lemma 14. We can therefore conclude that

$$\left\|\widehat{H}^{1/2}\overline{\Pi}_p\hat{v}\right\| \leq \left(2 + 2^{-p/4}\sqrt{p_{\max}}\right)\left\|\widehat{H}^{1/2}\widehat{\Pi}_p\overline{\Pi}_p\hat{v}\right\|.$$

$\square$

Equipped with these results, we are ready to prove Proposition 3.

*Proof of Proposition 3.* Note that $\widehat{H}^{1/2}\hat{v} = \widehat{H}^{1/2}\overline{\Pi}_p^-\hat{v} + \widehat{H}^{1/2}\overline{\Pi}_{p+1}^+\hat{v}$. By entries 6 through 8 of Lemma 14, we have

$$\left\|\widehat{H}^{1/2}\widehat{\Pi}_p^-\overline{\Pi}_{p+1}^+\hat{v}\right\| \leq 2^{4-p/2}\left\|\widehat{H}^{1/2}\overline{\Pi}_{p+1}^+\hat{v}\right\|,$$

indicating that the angle between $\widehat{H}^{1/2}\overline{\Pi}_{p+1}^+\hat{v}$ and the subspace projected by $\widehat{\Pi}_p^-$ is at most $\arcsin\left(2^{4-p/2}\right)$. On the other hand, by Lemma 16 we know that the angle between $\widehat{H}^{1/2}\overline{\Pi}_p^-\hat{v}$ and the subspace projected by $\widehat{\Pi}_p^-$ is at least $\pi/4$, which leads to

$$\begin{aligned}
\left\|\widehat{H}^{1/2}\hat{v}\right\| = \left\|\widehat{H}^{1/2}\overline{\Pi}_p^-\hat{v} + \widehat{H}^{1/2}\overline{\Pi}_{p+1}^+\hat{v}\right\| &\geq \frac{1}{2}\left\|\widehat{H}^{1/2}\widehat{\Pi}_p^-\overline{\Pi}_p^-\hat{v}\right\| \\
&\geq \frac{1}{2}\sin\left(\frac{\pi}{p_{\min}}\right) \cdot \left\|\widehat{H}^{1/2}\widehat{\Pi}_p^-\overline{\Pi}_p\hat{v}\right\| \geq 2^{-4} \cdot \left\|\widehat{H}^{1/2}\widehat{\Pi}_p^-\overline{\Pi}_p\hat{v}\right\|
\end{aligned}$$

by Lemma 15. Then by Lemma 17, we can conclude that

$$\left\|\widehat{H}^{1/2}\hat{v}\right\| \geq \frac{2^{-5}}{1 + 2^{-p/4}\sqrt{p_{\max}}} \cdot \left\|\widehat{H}^{1/2}\overline{\Pi}_p\hat{v}\right\|.$$

$\square$

## D.6 Properties of $\overline{\Pi}_{\text{base}}$

**Lemma 18.** *For any $\hat{v} \in \mathbb{R}^d$, we have*

$$\left\|\widehat{H}^{1/2}\overline{\Pi}_{\text{base}}\hat{v}\right\| \leq 40 p_{\max}^{3/2}\left\|\widehat{H}^{1/2}\hat{v}\right\|.$$

*Proof.* Given that

$$\widehat{H}^{1/2}\hat{v} = \widehat{H}^{1/2}\overline{\Pi}_{\text{base}}\hat{v} + \sum_{p=p_{\min}}^{p_{\max}} \widehat{H}^{1/2}\overline{\Pi}_p\hat{v},$$

we have

$$\left\|\widehat{H}^{1/2}\overline{\Pi}_{\mathrm{base}}\hat{v}\right\| \leq \left\|\widehat{H}^{1/2}\hat{v}\right\| + \sum_{p=p_{\min}}^{p_{\max}} \left\|\widehat{H}^{1/2}\overline{\Pi}_p\hat{v}\right\|,$$

where

$$\left\|\widehat{H}^{1/2}\hat{v}\right\| \geq \frac{2^{-5}}{1 + 2^{-p/4}\sqrt{p_{\max}}} \cdot \left\|\widehat{H}^{1/2}\overline{\Pi}_p\hat{v}\right\|, \quad \forall p_{\min} \leq p \leq p_{\max}$$

by Proposition 3. We can thus conclude that

$$\left\|\widehat{H}^{1/2}\overline{\Pi}_{\mathrm{base}}\hat{v}\right\| \leq \left(1 + p_{\max}(32 + 2\sqrt{p_{\max}})\right)\left\|\widehat{H}^{1/2}\hat{v}\right\| \leq 40p_{\max}^{3/2}\left\|\widehat{H}^{1/2}\hat{v}\right\|.$$

$\square$

**Corollary 4.** *For any $\hat{v} \in \mathbb{R}^d$, we have*

$$\left\|\widehat{H}^{1/2}\overline{\Pi}_{\mathrm{base}}\hat{v}\right\| \leq p_{\max}\sqrt{\delta} \cdot \left\|\overline{\Pi}_{\mathrm{base}}\hat{v}\right\|.$$

*Proof.* The proof follows from Lemma 13 by noticing that

$$\overline{\Pi}_{\mathrm{base}} = \overline{\Pi}_{p_{\min}-1}^{-}, \qquad \overline{\Pi}_{\mathrm{base}} + \overline{\Pi}_{p_{\min}}^{+} = I$$

given the definition of $\mathcal{S}_{\mathrm{base}}$ in (30). $\square$

# E  Analysis of Algorithm 1

## E.1  Quadratic Approximation of $f$ Near $x^{(0)}$

Given that $f$ is $L_2$-Hessian Lipschitz, in the neighborhood of $x^{(0)}$, it is close to $T_{x^{(0)}}^2(x)$, is the 2nd order Taylor approximation of $f$ evaluated at $x^{(0)}$. Throughout this section, we denote

$$g(x) := T_{x^{(t)}}^2(x) = f(x^{(0)}) + \langle \nabla f(x^{(0)}), x - x^{(0)} \rangle + \frac{1}{2}(x - x^{(0)})^\top \nabla^2 f(x^{(0)})(x - x^{(0)}). \quad (22)$$

Similarly to the definition of $\hat{f}$, we define $\hat{g} := (\widehat{H}^{-1/2}\hat{x})$, which satisfies

$$\nabla\hat{g}(\hat{x}) = \widehat{H}^{-1/2} \cdot \nabla g(\widehat{H}^{-1/2}\hat{x}),$$

and

$$\overline{H} := \nabla^2\hat{g}(\hat{x}) = \widehat{H}^{-1/2} \cdot \nabla^2 g(\widehat{H}^{-1/2}\hat{x}) \cdot \widehat{H}^{-1/2} = \widehat{H}^{-1/2} \cdot \nabla^2 f(x^{(0)}) \cdot \widehat{H}^{-1/2},$$

For any iteration $k$, we define

$$\iota^{(k)} := \nabla f(y^{(k)}) - \nabla g(y^{(k)}), \qquad \hat{\iota}^{(k)} := \nabla\hat{f}(\hat{y}^{(k)}) - \nabla\hat{g}(\hat{y}^{(k)}).$$

Then, we have $\iota^{(k)} = \widehat{H}^{1/2}\hat{\iota}^{(k)}$.

## E.2  Movement Bounds of the Iterates

In the case where Line 7 is triggered during Algorithm 1, we denote

$$\mathcal{K} := \underset{k}{\arg\min}\left\{k \sum_{t=0}^{k-1} \|x^{(t+1)} - x^{(t)}\|^2 > B^2\right\}.$$

Otherwise, we denote $\mathcal{K} = K + 1$.

**Lemma 19.** *For any iteration $k < \mathcal{K}$, we have*

1. *$k \sum_{t=1}^{k-1}\left\|x^{(t+1)} - x^{(t)}\right\|^2 < B^2$;*

2. *$\left\|x^{(k)} - x^{(0)}\right\| \leq B$;*

3. $\left\|y^{(k)} - x^{(0)}\right\| \le 2B$;

4. $\left\|\iota^{(k)}\right\| = \left\|\widehat{H}^{1/2}\hat{\iota}^{(k)}\right\| \le 2L_2B^2$

*Proof.* The first entry follows from Corollary 3:

$$k\sum_{k=1}^{k-1}\left\|x^{(k+1)} - x^{(k)}\right\|^2 < k\sum_{t=1}^{k-1}\left(\left\|\widehat{H}^{-1/2}\right\| \cdot \left\|\widehat{H}^{1/2}\big(x^{(t+1)} - x^{(t)}\big)\right\|\right)^2$$

$$< \frac{k}{12\delta p_{\max}}\sum_{t=1}^{k-1}\left\|\widehat{H}^{1/2}\big(x^{(t+1)} - x^{(t)}\big)\right\|^2 < B^2.$$

Then by Cauchy-Schwartz, we have

$$\left\|x^{(k)} - x^{(0)}\right\| \le \sqrt{k\sum_{t=1}^{k-1}\left\|\widehat{H}^{1/2}\big(x^{(t+1)} - x^{(t)}\big)\right\|^2} \le B,$$

which leads to

$$\left\|y^{(k)} - x^{(0)}\right\| \le \left\|x^{(k)} - x^{(0)}\right\| + (1-\theta)\left\|x^{(k)} - x^{(k-1)}\right\| \le 2B.$$

Since $f$ is $L_2$-Hessian Lipschitz, we can further derive that

$$\left\|\iota^{(k)}\right\| = \left\|\widehat{H}^{1/2}\hat{\iota}^{(k)}\right\| \le \frac{1}{2}L_2\left\|y^{(k)} - x^{(0)}\right\|^2 \le 2L_2B^2.$$

$\square$

**Lemma 20.** *Let $\eta \le 1/4$. In the case where the "if condition" in Line 7 of Algorithm 1 is triggered, we have $\|x^{(\mathcal{K})} - x^{(0)}\| \le 7B$.*

*Proof.* By Lemma 19, for any $k < \mathcal{K}$ we have $\|x^{(k)} - x^{(0)}\| \le B$ and $\|y^{(k)} - x^{(0)}\| \le 2B$. Hence, to bound $\|x^{(\mathcal{K})} - x^{(0)}\|$, it suffices bound $\|x^{(\mathcal{K})} - y^{(\mathcal{K}-1)}\|$, which satisfies

$$\left\|x^{(\mathcal{K})} - y^{(\mathcal{K}-1)}\right\| = \eta\left\|\widehat{H}^{-1}\nabla f(y^{(\mathcal{K}-1)})\right\|,$$

where

$$\widehat{H}^{-1}\nabla f(y^{(\mathcal{K}-1)})$$

$$= \widehat{H}^{-1}\nabla f(y^{(\mathcal{K}-2)}) + \widehat{H}^{-1}\int_{y=y^{(\mathcal{K}-2)}}^{y^{(\mathcal{K}-1)}}\nabla^2 f(y)\mathrm{d}y$$

$$= \widehat{H}^{-1}\nabla f(y^{(\mathcal{K}-2)}) + \widehat{H}^{-1}H(y^{(\mathcal{K}-1)} - y^{(\mathcal{K}-2)}) + \int_{y=y^{(\mathcal{K}-2)}}^{y^{(\mathcal{K}-1)}}\widehat{H}^{-1}\big(H - \nabla^2 f(y)\big)\mathrm{d}y. \quad (23)$$

The first and the second term satisfy

$$\left\|\widehat{H}^{-1}\nabla f(y^{(\mathcal{K}-2)})\right\| = \left\|x^{(\mathcal{K}-1)} - y^{(\mathcal{K}-2)}\right\|/\eta \le 3B/\eta,$$

and

$$\left\|\widehat{H}^{-1}H(y^{(\mathcal{K}-2)} - y^{(\mathcal{K}-1)})\right\| \le \left\|y^{(\mathcal{K}-2)} - y^{(\mathcal{K}-1)}\right\| \le 4B,$$

respectively. As for the third term, given that $\|\widehat{H}^{-1}\| \le (12\delta p_{\max})^{-1}$ by Corollary 3 and

$$\left\|\widehat{H} - \nabla^2 f(y)\right\| \le \delta + 4L_2B \le 2\delta,$$

it follows that

$$\left\|\int_{y=y^{(\mathcal{K}-2)}}^{y^{(\mathcal{K}-1)}}\widehat{H}^{-1}\big(H - \nabla^2 f(y)\big)\mathrm{d}y\right\| \le B.$$

Therefore,

$$\left\|x^{(\mathcal{K})} - y^{(\mathcal{K}-1)}\right\| \le (3 + 4\eta + 1)B = 5B,$$

and we can conclude that

$$\left\|x^{(\mathcal{K})} - x^{(0)}\right\| \le \left\|y^{(\mathcal{K}-1)} - x^{(0)}\right\| + \left\|x^{(\mathcal{K})} - y^{(\mathcal{K}-1)}\right\| \le 7B.$$

$\square$

## E.3 Function Value Decrease Case

In this subsection, we discuss the decrease in the function value of Algorithm 1 in the case where the "if condition" in Line 7 is triggered.

Given that $\overline{H}$ is symmetric, we can find a set of orthonormal basis $\{\hat{h}_1, \ldots, \hat{h}_d\}$ such that each $\hat{h}_i$ is an eigenvector of $\overline{H}$ with eigenvalue $\lambda_i$. We decompose these coordinates into two sets

$$\mathcal{S}_{\mathrm{sc}} := \left\{ i : \lambda_i \geq -\frac{\theta}{\eta} \right\}, \qquad \mathcal{S}_{\mathrm{nc}} := \left\{ i : \lambda_i < -\frac{\theta}{\eta} \right\}, \tag{24}$$

where sc and nc abbreviate strongly convex and not strongly convex, respectively. We further define the corresponding projectors

$$\Pi_{\mathrm{sc}} := \sum_{i \in \mathcal{S}_{\mathrm{sc}}} \hat{h}_i \hat{h}_i^\top, \quad \Pi_{\mathrm{nc}} := \sum_{i \in \mathcal{S}_{\mathrm{nc}}} \hat{h}_i \hat{h}_i^\top. \tag{25}$$

For any $\hat{v} \in \mathbb{R}^d$, denote

$$\hat{v}_{\mathrm{sc}} := \Pi_{\mathrm{sc}} \hat{v}, \qquad \hat{v}_{\mathrm{nc}} := \Pi_{\mathrm{nc}} \hat{v}$$

and

$$\hat{g}_{\mathrm{sc}}(\hat{v}) := \left\langle \nabla \hat{f}(\hat{x}^{(0)}), \hat{v}_{\mathrm{sc}} - \hat{x}_{\mathrm{sc}}^{(0)} \right\rangle + \frac{1}{2} \left( \hat{v}_{\mathrm{sc}} - \hat{x}_{\mathrm{sc}}^{(0)} \right)^\top \overline{H} \left( \hat{v}_{\mathrm{sc}} - \hat{x}_{\mathrm{sc}}^{(0)} \right)$$

$$\hat{g}_{\mathrm{nc}}(\hat{v}) := \left\langle \nabla \hat{f}(\hat{x}^{(0)}), \hat{v}_{\mathrm{nc}} - \hat{x}_{\mathrm{nc}}^{(0)} \right\rangle + \frac{1}{2} \left( \hat{v}_{\mathrm{nc}} - \hat{x}_{\mathrm{nc}}^{(0)} \right)^\top \overline{H} \left( \hat{v}_{\mathrm{nc}} - \hat{x}_{\mathrm{nc}}^{(0)} \right) \tag{26}$$

Then, we have $\hat{g}(\hat{v}) = \hat{g}_{\mathrm{sc}}(\hat{v}) + \hat{g}_{\mathrm{nc}}(\hat{v})$.

### E.3.1 Function Value Decrease of $\hat{g}_{\mathrm{sc}}$

The proof structure in this part is similar to the proof of [29, Lemma 2].

**Lemma 21.** *Let $\eta \leq 1/4$. Then for any $0 \leq k \leq \mathcal{K} - 1$ and any $\alpha > 0$, we have*

$$\hat{g}_{\mathrm{sc}}(\hat{x}^{(k+1)}) - \hat{g}_{\mathrm{sc}}(\hat{x}^{(k)}) \leq -\frac{1}{2} \left( \hat{x}_{\mathrm{sc}}^{(k)} - \hat{y}_{\mathrm{sc}}^{(k)} \right)^\top \overline{H} \left( \hat{x}_{\mathrm{sc}}^{(k)} - \hat{y}_{\mathrm{sc}}^{(k)} \right) + \frac{\| \iota^{(k)} \|^2}{2\alpha}$$

$$+ \frac{1}{2\eta} \left( \left\| \hat{x}_{\mathrm{sc}}^{(k)} - \hat{y}_{\mathrm{sc}}^{(k)} \right\|^2 - \left( 1 - \frac{\alpha \eta}{12 \delta p_{\max}} \right) \left\| \hat{x}_{\mathrm{sc}}^{(k+1)} - \hat{x}_{\mathrm{sc}}^{(k)} \right\|^2 \right).$$

*Proof.* Given that $\hat{g}_{\mathrm{sc}}$ is quadratic, for any two consecutive iterations, we have

$$\hat{g}_{\mathrm{sc}}(\hat{x}^{(k+1)}) = \hat{g}_{\mathrm{sc}}(\hat{x}^{(k)}) + \left\langle \nabla \hat{g}_{\mathrm{sc}}(\hat{x}^{(k)}), \hat{x}_{\mathrm{sc}}^{(k+1)} - \hat{x}_{\mathrm{sc}}^{(k)} \right\rangle$$

$$+ \frac{1}{2} \left( \hat{x}_{\mathrm{sc}}^{(k+1)} - \hat{x}_{\mathrm{sc}}^{(k)} \right)^\top \overline{H} \left( \hat{x}_{\mathrm{sc}}^{(k+1)} - \hat{x}_{\mathrm{sc}}^{(k)} \right),$$

where

$$\nabla \hat{g}_{\mathrm{sc}}(\hat{x}^{(k)}) = \nabla \hat{f}_{\mathrm{sc}}(\hat{y}^{(k)}) + \left( \nabla \hat{g}_{\mathrm{sc}}(\hat{y}^{(k)}) - \nabla \hat{f}_{\mathrm{sc}}(\hat{y}^{(k)}) \right) + \left( \nabla \hat{g}_{\mathrm{sc}}(\hat{x}^k) - \nabla \hat{g}_{\mathrm{sc}}(\hat{y}_{\mathrm{sc}}^k) \right)$$

$$= -\frac{1}{\eta} (\hat{x}_{\mathrm{sc}}^{(k+1)} - \hat{y}_{\mathrm{sc}}^{(k)}) + \overline{H} \left( \hat{x}_{\mathrm{sc}}^{(k)} - \hat{y}_{\mathrm{sc}}^{(k)} \right) - \hat{\iota}_{\mathrm{sc}}^{(k)}.$$

Hence,

$$\hat{g}_{\mathrm{sc}}(\hat{x}^{(k+1)}) - \hat{g}_{\mathrm{sc}}(\hat{x}_{\mathrm{sc}}^{(k)})$$

$$= -\frac{1}{\eta} \left\langle \hat{x}_{\mathrm{sc}}^{(k+1)} - \hat{y}_{\mathrm{sc}}^{(k)}, \hat{x}_{\mathrm{sc}}^{(k+1)} - \hat{x}_{\mathrm{sc}}^{(k)} \right\rangle - \left\langle \hat{\iota}_{\mathrm{sc}}^{(k)}, \hat{x}_{\mathrm{sc}}^{(k+1)} - \hat{x}_{\mathrm{sc}}^{(k)} \right\rangle$$

$$+ \left( \hat{x}_{\mathrm{sc}}^{(k)} - \hat{y}_{\mathrm{sc}}^{(k)} \right)^\top \overline{H} \left( \hat{x}_{\mathrm{sc}}^{(k+1)} - \hat{x}_{\mathrm{sc}}^{(k)} \right) + \frac{1}{2} \left( \hat{x}_{\mathrm{sc}}^{(k+1)} - \hat{x}_{\mathrm{sc}}^{(k)} \right)^\top \overline{H} \left( \hat{x}_{\mathrm{sc}}^{(k+1)} - \hat{x}_{\mathrm{sc}}^{(k)} \right),$$

where

$$\left\langle \hat{x}_{\mathrm{sc}}^{(k+1)} - \hat{y}_{\mathrm{sc}}^{(k)}, \hat{x}_{\mathrm{sc}}^{(k+1)} - \hat{x}_{\mathrm{sc}}^{(k)} \right\rangle = \left\| x_{\mathrm{sc}}^{(k)} - y_{\mathrm{sc}}^{(k)} \right\|^2 - \left\| x_{\mathrm{sc}}^{(k+1)} - y_{\mathrm{sc}}^{(k)} \right\|^2 - \left\| x_{\mathrm{sc}}^{(k+1)} - x_{\mathrm{sc}}^{(k)} \right\|^2$$

and

$$\big(\hat{x}_{\text{sc}}^{(k)} - \hat{y}_{\text{sc}}^{(k)}\big)^\top \overline{H}\big(\hat{x}_{\text{sc}}^{(k+1)} - \hat{x}_{\text{sc}}^{(k)}\big) + \frac{1}{2}\big(\hat{x}_{\text{sc}}^{(k+1)} - \hat{x}_{\text{sc}}^{(k)}\big)^\top \overline{H}\big(\hat{x}_{\text{sc}}^{(k+1)} - \hat{x}_{\text{sc}}^{(k)}\big)$$

$$= \frac{1}{2}\big(\hat{x}_{\text{sc}}^{(k+1)} - \hat{y}_{\text{sc}}^{(k)}\big)^\top \overline{H}\big(\hat{x}_{\text{sc}}^{(k+1)} - \hat{y}_{\text{sc}}^{(k)}\big) - \frac{1}{2}\big(\hat{x}_{\text{sc}}^{(k)} - \hat{y}_{\text{sc}}^{(k)}\big)^\top \overline{H}\big(\hat{x}_{\text{sc}}^{(k)} - \hat{y}_{\text{sc}}^{(k)}\big).$$

Furthermore, for any $\alpha > 0$ we have

$$-\big\langle \hat{\iota}_{\text{sc}}^{(k)}, \hat{x}_{\text{sc}}^{(k+1)} - \hat{x}_{\text{sc}}^{(k)}\big\rangle = -\big\langle \hat{\iota}^{(k)}, \hat{x}_{\text{sc}}^{(k+1)} - \hat{x}_{\text{sc}}^{(k)}\big\rangle$$

$$\leq \frac{1}{2\alpha}\big\|\widehat{H}^{1/2}\hat{\iota}\big\|^2 + \frac{\alpha}{2}\big\|\widehat{H}^{-1/2}\big(\hat{x}_{\text{sc}}^{(k+1)} - \hat{x}_{\text{sc}}^{(k)}\big)\big\|^2$$

$$\leq \frac{\|\iota^{(k)}\|^2}{2\alpha} + \frac{\alpha}{24\delta p_{\max}}\big\|\hat{x}_{\text{sc}}^{(k+1)} - \hat{x}_{\text{sc}}^{(k)}\big\|^2$$

by Corollary 3. It then follows that

$$\hat{g}_{\text{sc}}(\hat{x}^{(k+1)}) - \hat{g}_{\text{sc}}(\hat{x}^{(k)})$$

$$= \frac{1}{2}\big(\hat{x}_{\text{sc}}^{(k+1)} - y_{\text{sc}}^{(k)}\big)^\top \overline{H}\big(\hat{x}_{\text{sc}}^{(k+1)} - y_{\text{sc}}^{(k)}\big) - \frac{1}{2}\big(\hat{x}_{\text{sc}}^{(k)} - y_{\text{sc}}^{(k)}\big)^\top \overline{H}\big(\hat{x}_{\text{sc}}^{(k)} - y_{\text{sc}}^{(k)}\big) + \frac{\|\iota^{(k)}\|^2}{2\alpha}$$

$$+ \frac{1}{2\eta}\Big(\big\|\hat{x}_{\text{sc}}^{(k)} - \hat{y}_{\text{sc}}^{(k)}\big\|^2 - \big\|\hat{x}_{\text{sc}}^{(k+1)} - \hat{y}_{\text{sc}}^{(k)}\big\|^2 - \Big(1 - \frac{\alpha\eta}{12\delta p_{\max}}\Big)\big\|\hat{x}_{\text{sc}}^{(k+1)} - \hat{x}_{\text{sc}}^{(k)}\big\|^2\Big)$$

$$\leq -\frac{1}{2}\big(\hat{x}_{\text{sc}}^{(k)} - \hat{y}_{\text{sc}}^{(k)}\big)^\top \overline{H}\big(\hat{x}_{\text{sc}}^{(k)} - \hat{y}_{\text{sc}}^{(k)}\big) + \frac{\|\iota^{(k)}\|^2}{2\alpha}$$

$$+ \frac{1}{2\eta}\Big(\big\|\hat{x}_{\text{sc}}^{(k)} - \hat{y}_{\text{sc}}^{(k)}\big\|^2 - \Big(1 - \frac{\alpha\eta}{12\delta p_{\max}}\Big)\big\|\hat{x}_{\text{sc}}^{(k+1)} - \hat{x}_{\text{sc}}^{(k)}\big\|^2\Big),$$

where the last inequality follows from

$$\frac{1}{2}\big(\hat{x}_{\text{sc}}^{(k)} - y_{\text{sc}}^{(k)}\big)^\top \overline{H}\big(\hat{x}_{\text{sc}}^{(k)} - y_{\text{sc}}^{(k)}\big) - \frac{1}{2\eta}\big\|\hat{x}_{\text{sc}}^{(k+1)} - \hat{y}_{\text{sc}}^{(k)}\big\|^2 \leq 0.$$

since $\overline{H} \preceq I$ and $\eta \leq 1/4$. We can therefore conclude that

$$\hat{g}_{\text{sc}}(\hat{x}^{(k+1)}) - \hat{g}_{\text{sc}}(\hat{x}^{(k)}) \leq -\frac{1}{2}\big(\hat{x}_{\text{sc}}^{(k)} - \hat{y}_{\text{sc}}^{(k)}\big)^\top \overline{H}\big(\hat{x}_{\text{sc}}^{(k)} - \hat{y}_{\text{sc}}^{(k)}\big) + \frac{\|\iota^{(k)}\|^2}{2\alpha}$$

$$+ \frac{1}{2\eta}\Big(\big\|\hat{x}_{\text{sc}}^{(k)} - \hat{y}_{\text{sc}}^{(k)}\big\|^2 - \Big(1 - \frac{\alpha\eta}{12\delta p_{\max}}\Big)\big\|\hat{x}_{\text{sc}}^{(k+1)} - \hat{x}_{\text{sc}}^{(k)}\big\|^2\Big).$$

$\square$

**Lemma 22.** *Let $\eta \leq 1/4$ and $0 < \theta \leq 1$. In the case where the "if condition" in Line 7 of Algorithm 1 is triggered, we have*

$$\hat{g}_{\text{sc}}(\hat{x}^{(\mathcal{K})}) - \hat{g}_{\text{sc}}(\hat{x}^{(0)}) \leq -\frac{\theta}{4\eta}\sum_{k=0}^{\mathcal{K}-1}\big\|\hat{x}_{\text{sc}}^{(k+1)} - \hat{x}_{\text{sc}}^{(k)}\big\|^2 + \frac{\eta L_2^2 B^4 \mathcal{K}}{3\theta\delta p_{\max}}.$$

*Proof.* By Lemma 21, for any $\alpha > 0$ the following inequality holds:

$$\hat{g}_{\text{sc}}(\hat{x}^{(k+1)}) - \hat{g}_{\text{sc}}(\hat{x}^{(k)}) \leq -\frac{1}{2}\big(\hat{x}_{\text{sc}}^{(k)} - \hat{y}_{\text{sc}}^{(k)}\big)^\top \overline{H}\big(\hat{x}_{\text{sc}}^{(k)} - \hat{y}_{\text{sc}}^{(k)}\big) + \frac{\|\iota^{(k)}\|^2}{2\alpha}$$

$$+ \frac{1}{2\eta}\Big(\big\|\hat{x}_{\text{sc}}^{(k)} - \hat{y}_{\text{sc}}^{(k)}\big\|^2 - \Big(1 - \frac{\alpha\eta}{12\delta p_{\max}}\Big)\big\|\hat{x}_{\text{sc}}^{(k+1)} - \hat{x}_{\text{sc}}^{(k)}\big\|^2\Big),$$

where

$$-\frac{1}{2}\big(\hat{x}_{\text{sc}}^{(k)} - \hat{y}_{\text{sc}}^{(k)}\big)\overline{H}\big(\hat{x}_{\text{sc}}^{(k)} - \hat{y}_{\text{sc}}^{(k)}\big) \leq \frac{\theta}{2\eta}\big\|\hat{x}_{\text{sc}}^{(k)} - \hat{y}_{\text{sc}}^{(k)}\big\|^2$$

as per the definition of $\mathcal{S}_{\mathrm{sc}}$ in (24) and $\Pi_{\mathrm{sc}}$ in (25), which leads to

$$\hat{g}_{\mathrm{sc}}(\hat{x}^{(k+1)}) - \hat{g}_{\mathrm{sc}}(\hat{x}^{(k)}) \leq -\frac{1}{2\eta}\left(1 - \frac{\alpha\eta}{12\delta p_{\max}}\right)\big\|\hat{x}_{\mathrm{sc}}^{(k+1)} - \hat{x}_{\mathrm{sc}}^{(k)}\big\|^2 + \frac{\|\iota^{(k)}\|^2}{2\alpha}$$
$$+ \frac{(1+\theta)(1-\theta)^2}{2\eta}\big\|\hat{x}_{\mathrm{sc}}^{(k)} - \hat{x}_{\mathrm{sc}}^{(k-1)}\big\|^2,$$

given that $\hat{y}_{\mathrm{sc}}^{(k)} - \hat{x}_{\mathrm{sc}}^{(k)} = (1-\theta)(\hat{x}_{\mathrm{sc}}^{(k)} - \hat{x}_{\mathrm{sc}}^{(k-1)})$ as per (7). Define the potential function

$$\xi_{\mathrm{sc}}^{(k)} := \hat{g}_{\mathrm{sc}}(\hat{x}^{(k)}) + \frac{(1+\theta)(1-\theta)^2}{2\eta}\big\|\hat{x}_{\mathrm{sc}}^{(k)} - \hat{x}_{\mathrm{sc}}^{(k-1)}\big\|^2$$

and set $\alpha = 6\delta p_{\max}\theta/\eta$. Then, we have

$$\xi_{\mathrm{sc}}^{(k+1)} - \xi_{\mathrm{sc}}^{(k)} \leq -\frac{\theta}{4\eta}\big\|\hat{x}_{\mathrm{sc}}^{(k+1)} - \hat{x}_{\mathrm{sc}}^{(k)}\big\|^2 + \frac{\eta\|\iota^{(k)}\|^2}{12\delta p_{\max}\theta}.$$

Summing over all the iterations in this epoch, we can conclude that

$$\hat{g}_{\mathrm{sc}}(\hat{x}^{(\mathcal{K})}) - \hat{g}_{\mathrm{sc}}(\hat{x}^{(0)}) \leq -\frac{\theta}{4\eta}\sum_{k=0}^{\mathcal{K}-1}\big\|\hat{x}_{\mathrm{sc}}^{(k+1)} - \hat{x}_{\mathrm{sc}}^{(k)}\big\|^2 + \frac{\eta}{12\delta p_{\max}\theta}\sum_{k=0}^{\mathcal{K}-1}\|\iota^{(k)}\|^2$$
$$\leq -\frac{\theta}{4\eta}\sum_{k=0}^{\mathcal{K}-1}\big\|\hat{x}_{\mathrm{sc}}^{(k+1)} - \hat{x}_{\mathrm{sc}}^{(k)}\big\|^2 + \frac{\eta L_2^2 B^4\mathcal{K}}{3\delta p_{\max}\theta}.$$

$\square$

### E.3.2 Function Value Decrease of $\hat{g}_{\mathrm{nc}}$

**Lemma 23.** *Let $\eta \leq 1/4$ and $0 < \theta \leq 1$. In the case where the "if condition" in Line 7 of Algorithm 1 is triggered, we have*

$$\hat{g}_{\mathrm{nc}}(\hat{x}^{(\mathcal{K})}) - \hat{g}_{\mathrm{nc}}(\hat{x}^{(0)}) \leq -\frac{\theta}{2\eta}\sum_{k=0}^{\mathcal{K}-1}\big\|\hat{x}_{\mathrm{sc}}^{(k+1)} - \hat{x}_{\mathrm{sc}}^{(k)}\big\|^2 + \frac{\eta L_2^2 B^4\mathcal{K}}{6\theta\delta p_{\max}}.$$

*Proof.* The proof of this lemma follows a similar structure as the proof of [29, Lemma 3]. Denote $\hat{u} := \hat{x}_{\mathrm{nc}}^{(0)} - \overline{H}^\dagger\nabla\hat{g}_{\mathrm{nc}}(\hat{x}^{(0)})$, which allows us to rewrite $\hat{g}_{\mathrm{nc}}(\hat{v})$ as

$$\hat{g}_{\mathrm{nc}}(\hat{v}) = \frac{1}{2}(\hat{v}_{\mathrm{nc}} - \hat{u})^\top\overline{H}(\hat{v}_{\mathrm{nc}} - \hat{u}) - \frac{1}{2}\big(\nabla\hat{g}_{\mathrm{nc}}(\hat{x}^{(0)})\big)^\top\overline{H}^{-1}\nabla\hat{g}_{\mathrm{nc}}(\hat{x}^{(0)}), \quad \forall\hat{v} \in \mathbb{R}^d.$$

Then for any $0 \leq k \leq \mathcal{K} - 1$, we have

$$\hat{g}_{\mathrm{nc}}(\hat{x}^{(k+1)}) - \hat{g}_{\mathrm{nc}}(\hat{x}^{(k)})$$
$$= \frac{1}{2}(\hat{x}_{\mathrm{nc}}^{(k+1)} - \hat{u})^\top\overline{H}(\hat{x}_{\mathrm{nc}}^{(k+1)} - \hat{u}) - \frac{1}{2}(\hat{x}^{(k)} - \hat{u})^\top\overline{H}(\hat{x}^{(k)} - \hat{u})$$
$$= \frac{1}{2}\big(\hat{x}_{\mathrm{nc}}^{(k+1)} - \hat{x}_{\mathrm{nc}}^{(k)}\big)^\top\overline{H}\big(\hat{x}_{\mathrm{nc}}^{(k+1)} + 2\hat{x}_{\mathrm{nc}}^{(k)} - \hat{u}\big)$$
$$= \frac{1}{2}\big(\hat{x}_{\mathrm{nc}}^{(k+1)} - \hat{x}_{\mathrm{nc}}^{(k)}\big)^\top\overline{H}\big(\hat{x}_{\mathrm{nc}}^{(k+1)} - \hat{x}_{\mathrm{nc}}^{(k)}\big) + \big(\hat{x}_{\mathrm{nc}}^{(k+1)} - \hat{x}_{\mathrm{nc}}^{(k)}\big)^\top\overline{H}\big(\hat{x}_{\mathrm{nc}}^{(k)} - \hat{u}\big),$$

where the first term is upper bounded by

$$\frac{1}{2}\big(\hat{x}_{\mathrm{nc}}^{(k+1)} - \hat{x}_{\mathrm{nc}}^{(k)}\big)^\top\overline{H}\big(\hat{x}_{\mathrm{nc}}^{(k+1)} - \hat{x}_{\mathrm{nc}}^{(k)}\big) \leq -\frac{\theta}{2\eta}\big\|\hat{x}_{\mathrm{nc}}^{(k+1)} - \hat{x}_{\mathrm{nc}}^{(k)}\big\|^2$$

following the definition of $\mathcal{S}_{\mathrm{nc}}$ in (24) and $\Pi_{\mathrm{nc}}$ in (25). As for the second term, note that

$$\hat{x}_{\mathrm{nc}}^{(k+1)} - \hat{x}_{\mathrm{nc}}^{(k)} = \hat{y}_{\mathrm{nc}}^{(k)} - \hat{x}_{\mathrm{nc}}^{(k)} - \eta\nabla\hat{g}_{\mathrm{nc}}(\hat{y}^{(k)}) - \eta\hat{\iota}_{\mathrm{nc}}^{(k)}$$
$$= (1-\theta)\big(\hat{x}_{\mathrm{nc}}^{(k)} - \hat{x}_{\mathrm{nc}}^{(k-1)}\big) - \eta\overline{H}\big(\hat{y}_{\mathrm{nc}}^{(k)} - \hat{u}\big) - \eta\hat{\iota}_{\mathrm{nc}}^{(k)}$$
$$= (1-\theta)\big(\hat{x}_{\mathrm{nc}}^{(k)} - \hat{x}_{\mathrm{nc}}^{(k-1)}\big) - \eta\overline{H}\big(\hat{x}_{\mathrm{nc}}^{(k)} - \hat{u} + (1-\theta)\big(\hat{x}_{\mathrm{nc}}^{(k)} - \hat{x}_{\mathrm{nc}}^{(k-1)}\big)\big) - \eta\hat{\iota}_{\mathrm{nc}}^{(k)}.$$

where the second line follows from the observation that

$$\overline{H}(\hat{y}_{\text{nc}}^{(k)} - \hat{u}) = \overline{H}(\hat{y}_{\text{nc}}^{(k)} - \hat{x}_{\text{nc}}^{(0)}) + \overline{H}(\hat{x}_{\text{nc}}^{(0)} - \hat{u})$$
$$= \nabla \hat{g}_{\text{nc}}(\hat{y}^{(k)}) - \nabla \hat{g}_{\text{nc}}(\hat{x}^{(0)}) + \overline{H}\overline{H}^{\dagger}\nabla \hat{g}_{\text{nc}}(\hat{x}^{(0)}) = \nabla \hat{g}_{\text{nc}}(\hat{y}^{(k)}).$$

Hence,

$$\begin{aligned}
&(\hat{x}_{\text{nc}}^{(k+1)} - \hat{x}_{\text{nc}}^{(k)})^{\top}\overline{H}(\hat{x}_{\text{nc}}^{(k)} - \hat{u}) \\
&\quad = (1 - \theta)(\hat{x}_{\text{nc}}^{(k)} - \hat{x}_{\text{nc}}^{(k-1)})^{\top}\overline{H}(\hat{x}_{\text{nc}}^{(k)} - \hat{u}) - \eta\|\overline{H}(\hat{x}_{\text{nc}}^{(k)} - \hat{u})\|^{2} \quad\quad (27)\\
&\quad\quad - \eta(1 - \theta)(\hat{x}_{\text{nc}}^{(k)} - \hat{x}_{\text{nc}}^{(k-1)})^{\top}\overline{H}^{2}(\hat{x}_{\text{nc}}^{(k)} - \hat{u}) - \eta\langle\hat{\iota}_{\text{nc}}^{(k)}, \overline{H}(\hat{x}_{\text{nc}}^{(k)} - \hat{u})\rangle,
\end{aligned}$$

where we have

$$\begin{aligned}
&- \eta(1 - \theta)(\hat{x}_{\text{nc}}^{(k)} - \hat{x}_{\text{nc}}^{(k-1)})^{\top}\overline{H}^{2}(\hat{x}_{\text{nc}}^{(k)} - \hat{u}) \\
&\quad\quad \leq \frac{\eta(1 - \theta)}{2}\left(\|\overline{H}(\hat{x}_{\text{nc}}^{(k)} - \hat{x}_{\text{nc}}^{(k-1)})\|^{2} + \|\overline{H}(\hat{x}_{\text{nc}}^{(k)} - \hat{u})\|^{2}\right)
\end{aligned}$$

and

$$-\eta\langle\hat{\iota}_{\text{nc}}^{(k)}, \overline{H}(\hat{x}_{\text{nc}}^{(k)} - \hat{u})\rangle \leq \frac{\eta}{2(1 + \theta)}\|\hat{\iota}_{\text{nc}}^{(k)}\|^{2} + \frac{\eta(1 + \theta)}{2}\|\overline{H}(\hat{x}_{\text{nc}}^{(k)} - \hat{u})\|^{2}.$$

Combined with (27), we obtain

$$\begin{aligned}
&(\hat{x}_{\text{nc}}^{(k+1)} - \hat{x}_{\text{nc}}^{(k)})^{\top}\overline{H}(\hat{x}_{\text{nc}}^{(k)} - \hat{u}) \\
&\quad \leq (1 - \theta)(\hat{x}_{\text{nc}}^{(k)} - \hat{x}_{\text{nc}}^{(k-1)})^{\top}\overline{H}(\hat{x}_{\text{nc}}^{(k)} - \hat{u}) \\
&\quad\quad + \frac{\eta(1 - \theta)}{2}\|\overline{H}(\hat{x}_{\text{nc}}^{(k)} - \hat{x}_{\text{nc}}^{(k-1)})\|^{2} + \frac{\eta}{2(1 + \theta)}\|\hat{\iota}_{\text{nc}}^{(k)}\|^{2} \\
&\quad = (1 - \theta)(\hat{x}_{\text{nc}}^{(k)} - \hat{x}_{\text{nc}}^{(k-1)})^{\top}\overline{H}(\hat{x}_{\text{nc}}^{(k-1)} - \hat{u}) + (1 - \theta)(\hat{x}_{\text{nc}}^{(k)} - \hat{x}_{\text{nc}}^{(k-1)})^{\top}\overline{H}(\hat{x}_{\text{nc}}^{(k)} - \hat{x}_{\text{nc}}^{(k-1)}) \\
&\quad\quad + \frac{\eta(1 - \theta)}{2}\|\overline{H}(\hat{x}_{\text{nc}}^{(k)} - \hat{x}_{\text{nc}}^{(k-1)})\|^{2} + \frac{\eta}{2(1 + \theta)}\|\hat{\iota}_{\text{nc}}^{(k)}\|^{2} \\
&\quad \leq (1 - \theta)(\hat{x}_{\text{nc}}^{(k)} - \hat{x}_{\text{nc}}^{(k-1)})^{\top}\overline{H}(\hat{x}_{\text{nc}}^{(k-1)} - \hat{u}) + \frac{\eta}{2(1 + \theta)}\|\hat{\iota}_{\text{nc}}^{(k)}\|^{2},
\end{aligned}$$

where the last inequality follows from the fact that

$$(1 - \theta)(\hat{x}_{\text{nc}}^{(k)} - \hat{x}_{\text{nc}}^{(k-1)})^{\top}\overline{H}(\hat{x}_{\text{nc}}^{(k)} - \hat{x}_{\text{nc}}^{(k-1)}) + \frac{\eta(1 - \theta)}{2}\|\overline{H}(\hat{x}_{\text{nc}}^{(k)} - \hat{x}_{\text{nc}}^{(k-1)})\|^{2} \geq 0$$

since $\eta \leq 1/4$ and $\|\overline{H}\| \leq 1$. Hence,

$$\begin{aligned}
&(\hat{x}_{\text{nc}}^{(k+1)} - \hat{x}_{\text{nc}}^{(k)})^{\top}\overline{H}(\hat{x}_{\text{nc}}^{(k)} - \hat{u}) \\
&\quad \leq (1 - \theta)^{k}(\hat{x}_{\text{nc}}^{(k)} - \hat{x}_{\text{nc}}^{(k-1)})^{\top}\overline{H}(\hat{x}_{\text{nc}}^{(k-1)} - \hat{u}) + \frac{\eta}{2(1 + \theta)}\sum_{t=1}^{k}(1 - \theta)^{k-t}\|\hat{\iota}_{\text{nc}}^{(k)}\|^{2} \\
&\quad \leq \frac{\eta}{2}\sum_{t=1}^{k}(1 - \theta)^{k-t}\|\hat{\iota}_{\text{nc}}^{(k)}\|^{2},
\end{aligned}$$

where the last inequality follows from

$$(\hat{x}_{\text{nc}}^{(1)} - \hat{x}_{\text{nc}}^{(0)})^{\top}\overline{H}(\hat{x}_{\text{nc}}^{(0)} - \hat{u}) = (-\nabla \hat{g}_{\text{nc}}(\hat{x}_0)) \cdot \overline{H} \cdot (\overline{H}^{\dagger}\nabla \hat{g}_{\text{nc}}(\hat{x}^{(0)})) \leq 0.$$

Furthermore, for each $0 \leq k \leq \mathcal{K} - 1$, by Lemma 19 and Corollary 3 we have

$$\|\hat{\iota}_{\text{nc}}^{(k)}\|^{2} \leq \|\hat{\iota}^{(k)}\|^{2} = \|\widehat{H}^{-1/2}\widehat{H}^{1/2}\hat{\iota}^{(k)}\|^{2} \leq \|\widehat{H}^{-1/2}\|^{2} \cdot \|\iota^{(k)}\|^{2} \leq \frac{L_{2}^{2}B^{4}}{3\delta p_{\max}},$$

which leads to

$$\frac{\eta}{2}\sum_{t=1}^{k}(1 - \theta)^{k-t}\|\hat{\iota}_{\text{nc}}^{(k)}\|^{2} \leq \frac{\eta L_{\text{nc}}^{2}B^{4}}{6\theta\delta p_{\max}}$$

and

$$\hat{g}_{\mathrm{nc}}(\hat{x}^{(k+1)}) - \hat{g}_{\mathrm{nc}}(\hat{x}^{(k)}) \leq -\frac{\theta}{2\eta}\left\|\hat{x}_{\mathrm{nc}}^{(k+1)} - \hat{x}_{\mathrm{nc}}^{(k)}\right\|^2 + \frac{\eta L_2^2 B^4}{6\theta\delta p_{\max}}.$$

We can thus conclude that

$$\hat{g}_{\mathrm{nc}}(\hat{x}^{(\mathcal{K})}) - \hat{g}_{\mathrm{nc}}(\hat{x}^{(0)}) \leq -\frac{\theta}{2\eta}\sum_{k=0}^{\mathcal{K}-1}\left\|\hat{x}_{\mathrm{nc}}^{(k+1)} - \hat{x}_{\mathrm{nc}}^{(k)}\right\|^2 + \frac{\eta L_2^2 B^4 \mathcal{K}}{6\theta\delta p_{\max}}.$$

$\square$

### E.3.3 Function Value Decrease of $f$

**Proposition 4.** *Let $\eta \leq 1/4$ and $0 < \theta \leq 1$. In the case where the "if condition" in Line 7 of Algorithm 1 is triggered, we have*

$$f(x^{(\mathcal{K})}) - f(x^{(0)}) \leq -\frac{3\theta\delta B^2\lceil\log_2(L_1/\delta)\rceil}{\mathcal{K}\eta} + 36L_2 B^3 \leq -\sqrt{\frac{\tilde{\epsilon}^3}{L_2}},$$

*where $\tilde{\epsilon} = \epsilon/p_{\max}^8$ is defined in* (4).

*Proof.* Combining Lemma 22 and Lemma 23, we obtain

$$\hat{g}(\hat{x}^{(\mathcal{K})}) - \hat{g}(\hat{x}^{(0)}) \leq -\frac{\theta}{4\eta}\sum_{k=0}^{\mathcal{K}-1}\left\|\hat{x}^{(k+1)} - \hat{x}^{(k)}\right\|^2 + \frac{2\eta L_2^2 B^4 \mathcal{K}}{3\theta\delta p_{\max}} \leq -\frac{3\theta\delta p_{\max}B^2}{\eta\mathcal{K}} + \frac{2\eta L_2^2 B^4 \mathcal{K}}{3\theta\delta p_{\max}},$$

following the condition in Line 7. Then, we can conclude that

$$f(x^{(\mathcal{K})}) - f(x^{(0)}) = g(x^{(\mathcal{K})}) - g(x^{(0)}) + (f(x^{(\mathcal{K})}) - g(x^{(\mathcal{K})})) - (f(x^{(0)}) - g(x^{(0)}))$$

$$\leq -\frac{3\theta\delta p_{\max}B^2}{\eta\mathcal{K}} + \frac{2\eta L_2^2 B^4 \mathcal{K}}{3\theta\delta p_{\max}} + 60L_2 B^3 \leq -\sqrt{\frac{\tilde{\epsilon}^3}{L_2}}$$

given that $f$ is $L_2$-Hessian Lipschitz, and $\|x^{(\mathcal{K})} - x^{(0)}\| \leq 7B$ by Lemma 20. $\square$

### E.4 Small Gradient Case

In this subsection, we provide an upper bound on the gradient of the output of Algorithm 1 in the case where the last iterate stays close enough to $x^{(0)}$, or more concretely, the "if condition" in Line 7 is not triggered. Similar to Section E.3, we use $\{\hat{h}_1, \ldots, \hat{h}_d\}$ to denote the set of orthonormal vectors such that $\hat{h}_i$ is the eigenvector of $\overline{H}$ with eigenvalue $\lambda_i$. We decompose these coordinates into $\Theta(p_{\max})$ sets:

$$\mathcal{S}_p := \left\{i : \bar{l}_p < \lambda_i \leq \bar{l}_{p+1}\right\}, \qquad \forall p \in \mathbb{N}^+ \text{ and } p_{\min} \leq p \leq p_{\max}, \tag{28}$$

where the definition of $\bar{l}_p$ is given in (14). Then, the projector onto the eigenspace of eigenvectors with indices in $\mathcal{S}_p$ equals

$$\sum_{i\in\mathcal{S}_p}\hat{h}_i\hat{h}_i^\top = \overline{\Pi}_p, \quad \forall p \in \mathbb{N}^+ \text{ and } p_{\min} \leq p \leq p_{\max} \tag{29}$$

where $\overline{\Pi}_p$ is defined in (18). Moreover, define

$$\mathcal{S}_{\mathrm{base}} := \left\{i : \lambda_i < \bar{l}_{p_{\min}}\right\}. \tag{30}$$

Then, the projector onto the eigenspace of eigenvectors with indices in $\mathcal{S}_p$ equals

$$\sum_{i\in\mathcal{S}_{\mathrm{base}}}\hat{h}_i\hat{h}_i^\top = \overline{\Pi}_{\mathrm{base}}, \tag{31}$$

where $\overline{\Pi}_{\text{base}}$ is defined in (19). For any $\hat{v} \in \mathbb{R}^d$, denote

$$\hat{v}_{\text{base}} := \overline{\Pi}_{\text{base}}\hat{v}, \quad \text{and} \quad \hat{v}_p := \overline{\Pi}_p\hat{v}, \quad \forall p \in \mathbb{N}^+ \text{ and } p_{\min} \le p \le p_{\max}.$$

Moreover, for any $p \in \mathbb{N}^+$ and $p_{\min} \le p \le p_{\max}$, we define

$$\hat{g}_p(\hat{v}) := \langle \nabla \hat{f}(\hat{x}^{(0)}), \hat{v}_p - \hat{x}_p^{(0)} \rangle + \frac{1}{2}(\hat{v}_p - \hat{x}_p^{(0)})^\top \overline{H}(\hat{v}_p - \hat{x}_p^{(0)}), \tag{32}$$

and

$$\hat{g}_{\text{base}}(\hat{v}) := \langle \nabla \hat{f}(\hat{x}^{(0)}), \hat{v}_{\text{base}} - \hat{x}_{\text{base}}^{(0)} \rangle + \frac{1}{2}(\hat{v}_{\text{base}} - \hat{x}_{\text{base}}^{(0)})^\top \overline{H}(\hat{v}_{\text{base}} - \hat{x}_{\text{base}}^{(0)}), \tag{33}$$

Then, we have $\hat{g}(\hat{v}) = \hat{g}_{\text{base}}(\hat{v}) + \sum_{p=p_{\min}}^{p_{\max}} \hat{g}_p(\hat{v})$ and

$$\nabla \hat{g}(\hat{v}) = \nabla \hat{g}_{\text{base}}(\hat{v}) + \sum_{p=p_{\min}}^{p_{\max}} \nabla \hat{g}_p(\hat{v}.$$

### E.4.1 Small Gradient of $\hat{g}_p$ at $x^{\text{out}}$

In this part, we show that the gradient of $\hat{g}_p(x^{\text{out}})$ is small for any $p \in \mathbb{N}^+$ and $p_{\min} \le p \le p_{\max}$. We define

$$M := I - \eta\overline{H}, \quad M_p := \Pi_p \cdot M \cdot \Pi_p.$$

**Lemma 24.** *For any $v^{(k)} \in \mathbb{R}^d$ with the initial condition $v^{(-1)} = v^{(0)}$ and $\gamma^{(-1)} = 0$, that satisfies the following recursion formula*

$$v^{(k+1)} = a \cdot Mv^{(k)} - b \cdot Mv^{(k-1)} + \hat{\imath}^{(k)},$$

*for some symmetric matrix $M \in \mathbb{R}^{d \times d}$ and $a, b \in \mathbb{R}$, we have*

$$v^{(k)} = \psi_k(M)v^{(0)} + \sum_{j=0}^{k-1} \frac{\mathcal{P}(M)^{k-j} - \mathcal{Q}(M)^{k-j}}{\mathcal{P}(M) - \mathcal{Q}(M)} \cdot \gamma^{(j)},$$

*where*

$$\psi_k(M) := \frac{I - \mathcal{Q}(M)}{\mathcal{P}(M) - \mathcal{Q}(M)} \cdot \mathcal{P}(M)^{k+1} + \frac{\mathcal{P}(M) - I}{\mathcal{P}(M) - \mathcal{Q}(M)} \cdot \mathcal{Q}(M)^{k+1}, \quad \forall k \ge 0, \tag{34}$$

*and*

$$\mathcal{P}(M) := \frac{aM + \sqrt{a^2M^2 - 4bM}}{2}, \quad \mathcal{Q}(M) := \frac{aM - \sqrt{a^2M^2 - 4bM}}{2}. \tag{35}$$

*Proof.* The solution to the homogeneous part $v^{(k+1)} = a \cdot Mv^{(k)} - b \cdot Mv^{(k-1)}$ is

$$v^{(k)} = \psi_k(M)v^{(0)},$$

where

$$\psi_k(M) := \frac{I - \mathcal{Q}(M)}{\mathcal{P}(M) - \mathcal{Q}(M)} \cdot \mathcal{P}(M)^{k-1} + \frac{\mathcal{P}(M) - I}{\mathcal{P}(M) - \mathcal{Q}(M)} \cdot \mathcal{Q}(M)^{k-1}$$

with

$$\mathcal{P}(M) = \frac{aM + \sqrt{a^2M^2 - 4bM}}{2}, \quad \mathcal{Q}(M) = \frac{aM - \sqrt{a^2M^2 - 4bM}}{2}.$$

Counting in the inhomogeneous part, for each $\hat{\imath}^{(j)}$, it leads to the following additional term in $v^{(k)}$ for any $k \ge j$:

$$\frac{\mathcal{P}(M)^{k-j} - \mathcal{Q}(M)^{k-j}}{\mathcal{P}(M) - \mathcal{Q}(M)} \cdot \gamma^{(j)}.$$

We can conclude that

$$v^{(k)} = \psi_k(M)v^{(0)} + \sum_{j=0}^{k-1} \frac{\mathcal{P}(M)^{k-j} - \mathcal{Q}(M)^{k-j}}{\mathcal{P}(M) - \mathcal{Q}(M)} \cdot \gamma^{(j)}.$$

$\square$

**Lemma 25** (Properties of $\mathcal{P}(M_p)$ and $\mathcal{Q}(M_p)$). *Let $a := 2 - \theta$, $b := 1 - \theta$. If $\eta \leq 1/4$ and $\theta \leq \eta/(2p_{\max})$, for any $p \in \mathbb{N}^+$ and $p_{\min} \leq p \leq p_{\max}$, the matrices $\mathcal{P}(M_p)$ and $\mathcal{Q}(M_p)$ defined in (35) satisfy*

1. $\|\mathcal{P}(M_p)\| = \|\mathcal{Q}(M_p)\| = b\|M_p\| \leq 1 - \eta/p_{\max}$;

2. $\left\|\frac{1}{\mathcal{P}(M_p) - \mathcal{Q}(M_p)}\right\| \leq \sqrt{\frac{p_{\max}}{\eta}}$;

3. $\left\|\frac{I - \mathcal{Q}(M_p)}{\mathcal{P}(M_p) - \mathcal{Q}(M_p)}\right\| = \left\|\frac{\mathcal{P}(M_p) - I}{\mathcal{P}(M_p) - \mathcal{Q}(M_p)}\right\| \leq 2\sqrt{\frac{p_{\max}}{\eta}}$

*Proof.* Observe that

$$\mathcal{P}(M_p) := \frac{aM_p + \sqrt{a^2 M_p^2 - 4bM_p}}{2}, \quad \mathcal{Q}(M_p) := \frac{aM_p - \sqrt{a^2 M_p^2 - 4bM_p}}{2}.$$

Given the definition of $\Pi_p$ in (16), $M_p$ satisfies

$$(1 - \eta r_p)I \preceq M_p \preceq (1 - \eta l_p)I. \tag{36}$$

If $\eta \leq 1/4$ and $\theta \leq \eta/(2p_{\max})$, we have $a^2 M_p^2 - 4bM_p \preceq 0$, which leads to

$$\mathcal{P}(M_p) := \frac{aM_p + i\sqrt{-a^2 M_p^2 + 4bM_p}}{2}, \quad \mathcal{Q}(M_p) := \frac{aM_p - i\sqrt{-a^2 M_p^2 + 4bM_p}}{2}$$

Therefore, we have

$$\|\mathcal{P}(M_p)\| = \|\mathcal{Q}(M_p)\| = \frac{1}{2}\left\|(aM_p)^2 + (-a^2 M_p^2 + 4bM_p)\right\|^{1/2}$$

$$\leq \sqrt{b\|M_p\|} \leq 1 - \eta/p_{\max}.$$

As for the second entry, since

$$\mathcal{P}(M_p) - \mathcal{Q}(M_p) = i\sqrt{-a^2 M_p^2 + 4bM_p},$$

by (36) and the value of $a, b$ we have

$$\left\|\frac{1}{\mathcal{P}(M_p) - \mathcal{Q}(M_p)}\right\| \leq \sqrt{\frac{p_{\max}}{\eta}}$$

which leads to

$$\left\|\frac{I - \mathcal{Q}(M_p)}{\mathcal{P}(M_p) - \mathcal{Q}(M_p)}\right\| = \left\|\frac{\mathcal{P}(M_p) - I}{\mathcal{P}(M_p) - \mathcal{Q}(M_p)}\right\|$$

$$\leq (1 + \|\mathcal{P}(M_p)\|) \cdot \left\|\frac{1}{\mathcal{P}(M_p) - \mathcal{Q}(M_p)}\right\| \leq 2\sqrt{\frac{p_{\max}}{\eta}}.$$

$\square$

**Lemma 26** (Bound on the difference between $\nabla \hat{f}_p$ and $\nabla \hat{g}_p$). *For any $p \in \mathbb{N}^+$ and $p_{\min} \leq p \leq p_{\max}$, in the case where the "if condition" in Line 7 is not triggered, for any iteration $k$ of Algorithm 1, we have $\|\hat{\iota}_p^{(k)}\| \leq 2^{6-p/2} p_{\max} \|\iota^{(k)}\|/\sqrt{\delta}$.*

*Proof.* It follows from Proposition 2 that

$$\|\hat{\iota}_p^{(k)}\| = \|\overline{\Pi}_p \hat{\iota}\| \leq \frac{2}{\sqrt{\phi(2^p \delta)}} \|\widehat{H}^{1/2} \overline{\Pi}_p \hat{\iota}^{(k)}\|,$$

where

$$\|\widehat{H}^{1/2} \hat{\iota}\| \geq \frac{2^{-5}}{1 + 2^{-p/4} p_{\max}} \cdot \|\widehat{H}^{1/2} \overline{\Pi}_p \hat{\iota}^{(k)}\| \geq \frac{\|\widehat{H}^{1/2} \overline{\Pi}_p \hat{\iota}^{(k)}\|}{64 p_{\max}}$$

by Proposition 3, which leads to

$$\|\hat{\iota}_p^{(k)}\| \leq \frac{2^{6-p/2} p_{\max}}{\sqrt{\delta}} \|\widehat{H}^{1/2} \hat{\iota}\| = \frac{2^{6-p/2} p_{\max}}{\sqrt{\delta}} \|\iota^{(k)}\|.$$

$\square$

**Lemma 27.** *If $\eta \le \frac{1}{4}$ and $\theta \le \eta/(2p_{\max})$, then for any $p \in \mathbb{N}^+$ with $p_{\min} \le p \le p_{\max}$ and any $0 < j \le k-1$ we have*

$$\left\| \widehat{H}^{1/2}\overline{H}\left( \frac{\mathcal{P}(M)^{k-j} - \mathcal{Q}(M)^{k-j}}{\mathcal{P}(M) - \mathcal{Q}(M)} \right) \hat{\iota}_p^{(j)} \right\| \le \frac{128 p_{\max}^{5/2}}{\sqrt{\eta}} \left( 1 - \frac{\eta}{p_{\max}} \right)^{k-j} \|\iota^{(j)}\|.$$

*Proof.* It follows from Lemma 25 that

$$\left\| \left( \frac{\mathcal{P}(M)^{k-j} - \mathcal{Q}(M)^{k-j}}{\mathcal{P}(M) - \mathcal{Q}(M)} \right) \hat{\iota}_p^{(j)} \right\| \le \left\| \frac{1}{\mathcal{P}(M_p) - \mathcal{Q}(M_p)} \right\| \cdot \|\mathcal{P}(M_p)^{k-j}\| \cdot \|\hat{\iota}_p^{(j)}\|$$
$$+ \left\| \frac{1}{\mathcal{P}(M_p) - \mathcal{Q}(M_p)} \right\| \cdot \|\mathcal{Q}(M_p)^{k-j}\| \cdot \|\hat{\iota}_p^{(j)}\|$$
$$\le 2\sqrt{\frac{p_{\max}}{\eta}} \left( 1 - \frac{\eta}{p_{\max}} \right)^{k-j} \|\hat{\iota}_p^{(j)}\|$$

where

$$\|\hat{\iota}_p^{(j)}\| \le \frac{2^{6-p/2} p_{\max}}{\sqrt{\delta}} \|\iota^{(j)}\|.$$

by Lemma 26. Since

$$\overline{H}\left( \frac{\mathcal{P}(M)^{k-j} - \mathcal{Q}(M)^{k-j}}{\mathcal{P}(M) - \mathcal{Q}(M)} \right) \hat{\iota}_p^{(j)} = \overline{\Pi}_p \overline{H} \left( \frac{\mathcal{P}(M)^{k-j} - \mathcal{Q}(M)^{k-j}}{\mathcal{P}(M) - \mathcal{Q}(M)} \right) \hat{\iota}_p^{(j)},$$

we have

$$\left\| \widehat{H}^{1/2}\overline{H}\left( \frac{\mathcal{P}(M)^{k-j} - \mathcal{Q}(M)^{k-j}}{\mathcal{P}(M) - \mathcal{Q}(M)} \right) \hat{\iota}_p^{(j)} \right\|$$
$$= \left\| \widehat{H}^{1/2}\overline{\Pi}_p \overline{H}\left( \frac{\mathcal{P}(M)^{k-j} - \mathcal{Q}(M)^{k-j}}{\mathcal{P}(M) - \mathcal{Q}(M)} \right) \hat{\iota}_p^{(j)} \right\|$$
$$\le 2^{p/2} p_{\max}\sqrt{\delta} \cdot \left\| \overline{\Pi}_p\left( \frac{\mathcal{P}(M)^{k-j} - \mathcal{Q}(M)^{k-j}}{\mathcal{P}(M) - \mathcal{Q}(M)} \right) \hat{\iota}_p^{(j)} \right\|$$
$$\le 2^{1+p/2} p_{\max}^{3/2}\sqrt{\frac{\delta}{\eta}} \left( 1 - \frac{\eta}{p_{\max}} \right)^{k-j} \|\hat{\iota}_p^{(j)}\|$$
$$\le \frac{128 p_{\max}^{5/2}}{\sqrt{\eta}} \left( 1 - \frac{\eta}{p_{\max}} \right)^{k-j} \|\iota^{(j)}\|.$$

where the first inequality is by Proposition 2, and the second inequality follows from the fact that $\|\overline{H}\| \le 1$. $\qquad\qquad\square$

**Lemma 28.** *If $\eta \le \frac{1}{4}$ and $\theta \le \eta/(2p_{\max})$, then for any $p \in \mathbb{N}^+$ with $p_{\min} \le p \le p_{\max}$, any vector $\hat{v} \in \mathbb{R}^d$, and any integer $k \ge 0$, we have*

$$\left\| \widehat{H}^{1/2}\overline{H}\psi_j(M_p)\hat{v}_p \right\| \le 2^{2+p/2} p_{\max}^{3/2}\sqrt{\frac{\delta}{\eta}} \left( 1 - \frac{\eta}{p_{\max}} \right)^{k+1} \|\hat{v}_p\|.$$

*Proof.* By the definition of $\psi$ in (34) we have

$$\psi_k(M_p)\hat{v}_p = \frac{I - \mathcal{Q}(M_p)}{\mathcal{P}(M_p) - \mathcal{Q}(M_p)} \cdot \mathcal{P}(M_p)^{k+1}\hat{v}_p + \frac{\mathcal{P}(M_p) - I}{\mathcal{P}(M_p) - \mathcal{Q}(M_p)} \cdot \mathcal{Q}(M_p)^{k+1}\hat{v}_p$$

and

$$\|\psi_k(M_p)\hat{v}_p\| \le \left\| \frac{I - \mathcal{Q}(M_p)}{\mathcal{P}(M_p) - \mathcal{Q}(M_p)} \right\| \cdot \|\mathcal{P}(M_p)^{k+1}\| \cdot \|\hat{v}_p\|$$
$$+ \left\| \frac{\mathcal{P}(M_p) - I}{\mathcal{P}(M_p) - \mathcal{Q}(M_p)} \right\| \cdot \|\mathcal{Q}(M_p)^{j+1}\| \cdot \|\hat{v}_p\|$$
$$\le 4\sqrt{\frac{p_{\max}}{\eta}} \left( 1 - \frac{\eta}{p_{\max}} \right)^{k+1} \|\hat{v}_p\|$$

by Lemma 25. Since $\overline{H}\psi_k(M_p)\hat{v}_p = \overline{\Pi}_p\overline{H}\psi_k(M_p)\hat{v}_p$, we have

$$
\begin{aligned}
\left\|\widehat{H}^{1/2}\overline{H}\psi_k(M_p)\hat{v}_p\right\| &= \left\|\widehat{H}^{1/2}\overline{\Pi}_p\overline{H}\psi_k(M_p)\hat{v}_p\right\| \\
&\leq 2^{p/2}p_{\max}\sqrt{\delta}\cdot\left\|\overline{\Pi}_p\overline{H}\psi_k(M_p)\hat{v}_p\right\| \\
&\leq 2^{p/2}p_{\max}\sqrt{\delta}\left\|\psi_k(M_p)\hat{v}_p\right\| \\
&\leq 2^{2+p/2}p_{\max}^{3/2}\sqrt{\frac{\delta}{\eta}}\Big(1-\frac{\eta}{p_{\max}}\Big)^{k+1}\|\hat{v}_p\|.
\end{aligned}
$$

where the first inequality is by Proposition 2, and the second inequality follows from the fact that $\|\overline{H}\| \leq 1$. $\qquad\square$

**Lemma 29.** *If $\eta \leq 1/4$ and $\theta \leq \eta/(2p_{\max})$, for any $p \in \mathbb{N}^+$ and $p_{\min} \leq p \leq p_{\max}$, in the case where the "if condition" in Line 7 is not triggered, for any iteration $k \geq 0$ of Algorithm 1 we have*

$$
\left\|\widehat{H}^{1/2}\nabla\hat{g}_p(\hat{x}^{(k)})\right\| \leq \frac{2^{4+p/2}\delta p_{\max}^2 B}{\eta^{3/2}}\left(1-\frac{\eta}{p_{\max}}\right)^{k+1} + \frac{256L_2B^2p_{\max}^{7/2}}{\eta^{1/2}}
$$

*Proof.* For any iteration $k$, by (7) we have

$$
\begin{aligned}
\hat{x}_p^{(k+1)} &= \hat{y}_p^{(k)} - \eta\nabla\hat{f}_p(\hat{y}^{(k)}) \\
&= \hat{y}_p^{(k)} - \eta\overline{H}(\hat{y}_p^{(k)} - \hat{x}_p^{(0)}) - \eta\nabla\hat{f}_p(\hat{x}^{(0)}) - \eta\hat{\iota}_p^{(k)} \\
&= \hat{x}_p^{(k)} - \eta\nabla\hat{f}_p(\hat{x}^{(0)}) + (1-\theta)(\hat{x}_p^{(k)} - \hat{x}_p^{(k-1)}) \\
&\quad - \eta\overline{H}\big(\hat{x}_p^{(k)} - \hat{x}_p^{(0)} + (1-\theta)(\hat{x}_p^{(k)} - \hat{x}_p^{(k-1)})\big) - \eta\hat{\iota}_p^{(k)}.
\end{aligned}
$$

Denote $\tilde{x}_p^{(k)} := \hat{x}_p^{(k)} - \hat{x}_p^{(0)} + \overline{H}^\dagger\nabla\hat{f}_p(\hat{x}^{(0)})$. Then, the above equation is equivalent to

$$
\begin{aligned}
\tilde{x}_p^{(k+1)} &= \tilde{x}_p^{(k)} + (1-\theta)(\tilde{x}_p^{(k)} - \tilde{x}_p^{(k-1)}) - \eta\overline{H}\big(\tilde{x}_p^{(k)} + (1-\theta)(\tilde{x}_p^{(k)} - \tilde{x}_p^{(k-1)})\big) - \eta\hat{\iota}_p^{(k)} \\
&= \big(I - \eta\overline{H}\big)\big((2-\theta)\tilde{x}_p^{(k)} - (1-\theta)\tilde{x}_p^{(k-1)}\big) - \eta\hat{\iota}_p^{(k)}.
\end{aligned}
$$

Let $a := 2-\theta$, $b := 1-\theta$. By Lemma 24 we have

$$
\tilde{x}_p^{(k)} = \psi_k(M_p)\cdot\tilde{x}_p^{(0)} - \eta\sum_{j=0}^{k-1}\frac{\mathcal{P}(M)^{k-j} - \mathcal{Q}(M)^{k-j}}{\mathcal{P}(M) - \mathcal{Q}(M)}\cdot\hat{\iota}_p^{(j)}, \tag{37}
$$

where

$$
\psi_k(M_p) := \frac{I - \mathcal{Q}(M_p)}{\mathcal{P}(M_p) - \mathcal{Q}(M_p)}\cdot\mathcal{P}(M_p)^{k+1} + \frac{\mathcal{P}(M_p) - I}{\mathcal{P}(M_p) - \mathcal{Q}(M_p)}\cdot\mathcal{Q}(M_p)^{k+1},
$$

which leads to

$$
\begin{aligned}
\widehat{H}^{1/2}\nabla\hat{g}_p(\hat{x}^{(k)}) = \widehat{H}^{1/2}\overline{H}\tilde{x}_p^{(k)} &= \widehat{H}^{1/2}\overline{H}\psi_k(M_p)\tilde{x}_p^{(0)} \\
&\quad - \eta\sum_{j=0}^{k-1}\widehat{H}^{1/2}\overline{H}\cdot\frac{\mathcal{P}(M)^{k-j} - \mathcal{Q}(M)^{k-j}}{\mathcal{P}(M) - \mathcal{Q}(M)}\cdot\hat{\iota}_p^{(j)}. \tag{38}
\end{aligned}
$$

For the first term in (38), by Lemma 28 we have

$$
\left\|\widehat{H}^{1/2}\overline{H}\psi_k(M_p)\tilde{x}_p^{(0)}\right\| \leq 2^{2+p/2}p_{\max}^{3/2}\sqrt{\frac{\delta}{\eta}}\Big(1-\frac{\eta}{p_{\max}}\Big)^{k+1}\left\|\tilde{x}_p^{(0)}\right\|
$$

where

$$
\left\|\tilde{x}_p^{(0)}\right\| = \left\|\overline{H}^\dagger\nabla\hat{f}_p(\hat{x}^{(0)})\right\| \leq \sqrt{p_{\max}}\left\|\nabla\hat{f}_p(\hat{x}^{(0)})\right\|
$$

and

$$
\left\|\nabla\hat{f}_p(\hat{x}^{(0)})\right\| \leq \left\|\nabla\hat{f}(\hat{x}^{(0)})\right\| = \frac{\left\|\hat{x}^{(1)} - \hat{x}^{(0)}\right\|}{\eta} = \frac{4\sqrt{\delta p_{\max}}B}{\eta}
$$

which leads to

$$\left\|\widehat{H}^{1/2}\overline{H}\psi_k(M_p)\tilde{x}_p^{(0)}\right\| \le \frac{2^{4+p/2}\delta p_{\max}^2 B}{\eta^{3/2}}\left(1-\frac{\eta}{p_{\max}}\right)^{k+1}.$$

As for the second term of (38), by Lemma 27 we have

$$\left\|\eta\sum_{j=0}^{k-1}\widehat{H}^{1/2}\overline{H}\cdot\frac{\mathcal{P}(M)^{k-j}-\mathcal{Q}(M)^{k-j}}{\mathcal{P}(M)-\mathcal{Q}(M)}\cdot\hat{\iota}_p^{(j)}\right\| \le \eta\sum_{j=0}^{k-1}\left\|\widehat{H}^{1/2}\overline{H}\cdot\frac{\mathcal{P}(M)^{k-j}-\mathcal{Q}(M)^{k-j}}{\mathcal{P}(M)-\mathcal{Q}(M)}\cdot\hat{\iota}_p^{(j)}\right\|$$

$$\le 128 p_{\max}^{5/2}\eta^{1/2}\sum_{j=0}^{k-1}\left(1-\frac{\eta}{p_{\max}}\right)^{k-j}\left\|\iota^{(j)}\right\|,$$

where for each $j$ we have

$$\left\|\iota^{(j)}\right\| \le \frac{1}{2}L_2\left\|y^{(j)}-x^{(0)}\right\|^2 \le 2L_2 B^2$$

given that $f$ is $L_2$-Hessian Lipschitz. Hence,

$$\left\|\eta\sum_{j=0}^{k-1}\widehat{H}^{1/2}\overline{H}\cdot\frac{\mathcal{P}(M)^{k-j}-\mathcal{Q}(M)^{k-j}}{\mathcal{P}(M)-\mathcal{Q}(M)}\cdot\hat{\iota}_p^{(j)}\right\|$$

$$\le 256 L_2 B^2 p_{\max}^{5/2}\eta^{1/2}\sum_{j=0}^{k-1}\left(1-\frac{\eta}{p_{\max}}\right)^{k-j} \le 256 L_2 B^2 p_{\max}^{7/2}/\eta^{1/2}.$$

We can therefore conclude that

$$\left\|\widehat{H}^{1/2}\nabla\hat{g}_p(\hat{x}^{(k)})\right\| \le \frac{2^{4+p/2}\delta p_{\max}^2 B}{\eta^{3/2}}\left(1-\frac{\eta}{p_{\max}}\right)^{k+1} + \frac{256 L_2 B^2 p_{\max}^{7/2}}{\eta^{1/2}}.$$

$\square$

**Lemma 30.** *If $\eta \le 1/4$ and $\theta \le \eta/(2p_{\max})$, for any $p \in \mathbb{N}^+$ and $p_{\min} \le p \le p_{\max}$, in the case where the "if condition" in Line 7 is not triggered, we have*

$$\left\|\widehat{H}^{1/2}\nabla\hat{g}_p(\hat{x}^{\mathrm{out}})\right\| \le \frac{2^{6+p/2}\delta p_{\max}^2 B}{\eta^{3/2}}\left(1-\frac{\eta}{p_{\max}}\right)^{K/2} + \frac{2^{10}L_2 B^2 p_{\max}^{7/2}}{\eta^{1/2}}$$

*Proof.* Given that $\hat{y}^{(k)} = \hat{x}^{(k)} + (1-\theta)(\hat{x}^{(k)}-\hat{x}^{(k-1)})$ and that $\hat{g}_p$ is quadratic, for any $k \ge 1$ we have

$$\nabla\hat{g}_p(\hat{y}^{(k)}) = \nabla\hat{g}_p(\hat{x}^{(k)}) + (1-\theta)\big(\nabla\hat{g}_p(\hat{x}^{(k)}) - \nabla\hat{g}_p(\hat{x}^{(k-1)})\big),$$

which leads to

$$\left\|\widehat{H}^{1/2}\nabla\hat{g}_p(\hat{y}^{(k)})\right\| \le (2-\theta)\left\|\widehat{H}^{1/2}\hat{g}_p(\hat{x}^{(k)})\right\| + (1-\theta)\left\|\widehat{H}^{1/2}\hat{g}_p(\hat{x}^{(k-1)})\right\|$$

$$\le 2\left\|\widehat{H}^{1/2}\hat{g}_p(\hat{x}^{(k)})\right\| + \left\|\widehat{H}^{1/2}\hat{g}_p(\hat{x}^{(k-1)})\right\|$$

$$\le \frac{2^{6+p/2}\delta p_{\max}^2 B}{\eta^{3/2}}\left(1-\frac{\eta}{p_{\max}}\right)^{k} + \frac{2^{10}L_2 B^2 p_{\max}^{7/2}}{\eta^{1/2}}$$

by Lemma 29. Furthermore, since

$$\hat{x}^{\mathrm{out}} = \frac{1}{K_0+1-\lfloor K/2\rfloor}\sum_{k=\lfloor K/2\rfloor}^{K_0}\hat{y}^{(t)},$$

we have

$$\left\|\widehat{H}^{1/2}\nabla\hat{g}_p(\hat{x}^{\mathrm{out}})\right\| \le \frac{1}{K_0+1-\lfloor K/2\rfloor}\sum_{k=\lfloor K/2\rfloor}^{K_0}\left\|\widehat{H}^{1/2}\hat{g}_p(\hat{y}^{(t)})\right\|$$

$$\le \frac{2^{6+p/2}\delta p_{\max}^2 B}{\eta^{3/2}}\left(1-\frac{\eta}{p_{\max}}\right)^{K/2} + \frac{2^{10}L_2 B^2 p_{\max}^{7/2}}{\eta^{1/2}}.$$

$\square$

### E.4.2 Small Gradient of $\hat{g}_{\text{base}}$ at $x^{\text{out}}$

**Lemma 31.** *If $\eta \leq 1/4$ and $\theta \leq \eta/2$, in the case where the "if condition" in Line 7 is not triggered, we have*

$$\|\widehat{H}^{1/2}\nabla\hat{g}_{\text{base}}(\hat{x}^{\text{out}})\| \leq \frac{32\ell\sqrt{\delta p_{\max}}B}{\eta K^2} + \frac{8\theta\ell\sqrt{\delta p_{\max}}B}{\eta K} + 80p_{\max}^{3/2}L_2B^2$$

*for any $\ell$ that satisfies*

$$\|\widehat{H}^{1/2}\hat{v}_{\text{base}}\| \leq \ell \cdot \|\hat{v}_{\text{base}}\|, \quad \forall \hat{v} \in \mathbb{R}^d. \tag{39}$$

*Proof.* The proof of this lemma has a similar structure as the proof of [29, Lemma 5]. Given that $\hat{g}_{\text{base}}$ is quadratic, $\nabla\hat{g}_{\text{base}}(\hat{x}^{\text{out}})$ can be expressed as

$$\nabla\hat{g}_{\text{base}}(\hat{x}^{\text{out}}) = \frac{1}{K_0 + 1 - K/2} \sum_{k=\lfloor K/2 \rfloor}^{K_0} \nabla\hat{g}_{\text{base}}(\hat{y}^{(k)}),$$

where we have

$$
\begin{aligned}
-\nabla\hat{g}_{\text{base}}(\hat{y}^{(k)}) &= \frac{1}{\eta}\left(\hat{x}_{\text{base}}^{(k+1)} - \hat{y}_{\text{base}}^{(k)}\right) + \hat{\iota}_{\text{base}} \\
&= \frac{1}{\eta}\left((\hat{x}_{\text{base}}^{(k+1)} - \hat{x}_{\text{base}}^{(k)}) - (1-\theta)(\hat{x}_{\text{base}}^{(k)} - \hat{x}_{\text{base}}^{(k-1)})\right) + \hat{\iota}_{\text{base}}
\end{aligned}
$$

and

$$-\eta(K_0 + 1 - K/2)\nabla\hat{g}_{\text{base}}(\hat{x}^{\text{out}}) = \hat{x}_{\text{base}}^{(K_0+1)} - \hat{x}_{\text{base}}^{(K_0)} + \theta\left(\hat{x}_{\text{base}}^{(K_0)} - \hat{x}_{\text{base}}^{(K/2)}\right) + \eta\sum_{k=\lfloor K/2 \rfloor}^{K_0}\hat{\iota}_{\text{base}},$$

where the last term satisfies

$$
\begin{aligned}
\left\|\widehat{H}^{1/2}\sum_{k=\lfloor K/2 \rfloor}^{K_0}\hat{\iota}_{\text{base}}\right\| &\leq \sum_{k=\lfloor K/2 \rfloor}^{K_0}\left\|\widehat{H}^{1/2}\hat{\iota}_{\text{base}}\right\| \leq 40p_{\max}^{3/2}\sum_{k=\lfloor K/2 \rfloor}^{K_0}\left\|\widehat{H}^{1/2}\hat{\iota}\right\| \\
&\leq 80(K_0 + 1 - K/2)p_{\max}^{3/2}L_2B^2
\end{aligned}
$$

by Lemma 18. Hence,

$$
\begin{aligned}
&\left\|\widehat{H}^{1/2}\nabla\hat{g}_{\text{base}}(\hat{x}^{\text{out}})\right\| \\
&\leq \frac{4}{\eta K}\|\widehat{H}^{1/2}(\hat{x}_{\text{base}}^{(K_0+1)} - \hat{x}_{\text{base}}^{(K_0)})\| + \frac{4\theta}{\eta K}\|\widehat{H}^{1/2}(\hat{x}_{\text{base}}^{(K_0)} - \hat{x}_{\text{base}}^{(K/2)})\| + 80p_{\max}^{3/2}L_2B^2 \\
&\leq \frac{4\ell}{\eta K}\|\hat{x}_{\text{base}}^{(K_0+1)} - \hat{x}_{\text{base}}^{(K_0)}\| + \frac{4\theta\ell}{\eta K}\|\hat{x}_{\text{base}}^{(K_0)} - \hat{x}_{\text{base}}^{(K/2)}\| + 80p_{\max}^{3/2}L_2B^2 \\
&\leq \frac{4\ell}{\eta K}\|\hat{x}^{(K_0+1)} - \hat{x}^{(K_0)}\| + \frac{4\theta\ell}{\eta K}\|\hat{x}^{(K_0)} - \hat{x}^{(K/2)}\| + 80p_{\max}^{3/2}L_2B^2
\end{aligned}
$$

where the second inequality is due to the condition given in (39). By the condition $K_0 \leftarrow \text{argmin}_{\lfloor \frac{3K}{4} \rfloor \leq t \leq K-1}\|\hat{x}^{(t+1)} - \hat{x}^{(t)}\|$, we have

$$\left\|\hat{x}^{(K_0+1)} - \hat{x}^{(K_0)}\right\|^2 \leq \frac{1}{K - \lfloor 3K/4 \rfloor}\sum_{k=\lfloor 3K/4 \rfloor}^{K-1}\|x_{\text{base}}^{(k+1)} - x_{\text{base}}^{(k)}\|^2 \leq \frac{48\delta p_{\max}B^2}{K^2}$$

given that the "if condition" in Line 7 is not triggered, which leads to

$$\|\widehat{H}^{1/2}\nabla\hat{g}_{\text{base}}(\hat{x}^{\text{out}})\| \leq \frac{32\ell\sqrt{\delta p_{\max}}B}{\eta K^2} + \frac{8\theta\ell\sqrt{\delta p_{\max}}B}{\eta K} + 80p_{\max}^{3/2}L_2B^2$$

$\square$

**Corollary 5.** *If $\eta \leq 1/4$ and $\theta \leq \eta/2$, in the case where the "if condition" in Line 7 is not triggered, we have*

$$\|\widehat{H}^{1/2}\nabla\hat{g}_{\text{base}}(\hat{x}^{\text{out}})\| \leq \frac{32p_{\max}^{3/2}\delta B}{\eta K^2} + \frac{8\theta p_{\max}^{3/2}\delta B}{\eta K} + 80p_{\max}^{3/2}L_2B^2$$

*Proof.* The desired inequality follows by combining Lemma 31 with Corollary 4. $\square$

### E.4.3  Small Gradient of $f$ at $x^{\text{out}}$

**Proposition 5.** *With the choice of parameters in Theorem 2, in the case where the "if condition" in Line 7 is not triggered, we have $\left\|\nabla f(x^{\text{out}})\right\| \leq \epsilon$.*

*Proof.* By Lemma 30 and Corollary 5, we have

$$
\left\|\nabla g(x^{\text{out}})\right\| = \left\|\widehat{H}^{1/2}\hat{g}(\hat{x}^{\text{out}})\right\| = \left\|\widehat{H}^{1/2}\Big(\nabla\hat{g}(\hat{x}^{\text{out}}) + \sum_{p=p_{\min}}^{p_{\max}} \nabla\hat{g}_p(\hat{x}^{\text{out}})\Big)\right\|
$$

$$
\leq \left\|\widehat{H}^{1/2}\nabla\hat{g}_{\text{base}}(\hat{x}^{\text{out}})\right\| + \sum_{p=p_{\min}}^{p_{\max}} \left\|\widehat{H}^{1/2}\nabla\hat{g}_p(\hat{x}^{\text{out}})\right\|
$$

$$
\leq \frac{2^{6+p/2}\delta p_{\max}^3 B}{\eta^{3/2}}\Big(1 - \frac{\eta}{p_{\max}}\Big)^{K/2} + \frac{32 p_{\max}^{3/2}\delta B}{\eta K^2} + \frac{8\theta p_{\max}^{3/2}\delta B}{\eta K} + \frac{2^{10}L_2 B^2 p_{\max}^{9/2}}{\eta^{1/2}}
$$

$$
= \frac{2^{6+p/2}\delta p_{\max}^2 B}{\eta^{3/2}}\Big(1 - \frac{\eta}{p_{\max}}\Big)^{K/2} + \frac{40 p_{\max}^{3/2}\delta B}{\eta K^2} + \frac{2^{10}L_2 B^2 p_{\max}^{9/2}}{\eta^{1/2}},
$$

and

$$
\left\|\nabla f(x^{\text{out}})\right\| \leq \left\|\nabla g(x^{\text{out}})\right\| + \left\|\nabla f(x^{\text{out}}) - \nabla g(x^{\text{out}})\right\|
$$

$$
\leq \left\|\nabla g(x^{\text{out}})\right\| + 2L_2 B^2
$$

$$
\leq \frac{2^{6+p/2}\delta p_{\max}^2 B}{\eta^{3/2}}\Big(1 - \frac{\eta}{p_{\max}}\Big)^{K/2} + \frac{40 p_{\max}^{3/2}\delta B}{\eta K^2} + \frac{2^{11}L_2 B^2 p_{\max}^{9/2}}{\eta^{1/2}}. \qquad (40)
$$

Given that

$$
K \geq \frac{2p_{\max}}{\eta}\log\left(\frac{3 \times 2^{6+p/2}\delta p_{\max}^2 B}{\eta^{3/2}\epsilon}\right),
$$

the first term of (40) satisfies

$$
\frac{2^{6+p/2}\delta p_{\max}^2 B}{\eta^{3/2}}\Big(1 - \frac{\eta}{p_{\max}}\Big)^{K/2} + \frac{40 p_{\max}^{3/2}\delta B}{\eta K^2} + \frac{2^{11}L_2 B^2 p_{\max}^{9/2}}{\eta^{1/2}} \leq \frac{\epsilon}{3}.
$$

Furthermore, the second and the third term satisfy

$$
\frac{40 p_{\max}^{3/2}\delta B}{\eta K^2} \leq \frac{\epsilon}{3}, \qquad \frac{2^{11}L_2 B^2 p_{\max}^{9/2}}{\eta^{1/2}} \leq \frac{\epsilon}{3},
$$

respectively. We can thus conclude that $\left\|\nabla f(x^{\text{out}})\right\| \leq \epsilon$. $\qquad\square$

### E.5  Putting Everything Together

In this section, we give the proof of Theorem 2, and present an additional Lemma that characterizes the suboptimality of $x^{\text{out}}$, the output of Algorithm 1.

*Proof of Theorem 2.* By combining Proposition 4 and Proposition 5, we know that at least one of the two conditions in the theorem statement must hold. As for the distance between $x^{\text{out}}$ and $x^{(0)}$, in the case where the "if condition" in Line 7 is triggered, we have

$$
\left\|x^{\text{out}} - x^{(0)}\right\| = \left\|x^{(\mathcal{K})} - x^{(0)}\right\| \leq 7B
$$

by Lemma 20. Otherwise,

$$
\left\|x^{\text{out}} - x^{(0)}\right\| \leq \frac{1}{K_0 + 1 - \lfloor K/2 \rfloor} \sum_{k=\lfloor K/2 \rfloor}^{K_0} \left\|y^{(k)} - x^{(0)}\right\| \leq 2B
$$

by Lemma 19. $\qquad\square$

**Lemma 32.** *In the case where the "if condition" in Line 7 is triggered, the output $x^{\text{out}}$ of Algorithm 1 satisfies*

$$f(x^{\text{out}}) - f(x^{(0)}) \leq \frac{6\delta\epsilon}{L_2} + \frac{L_2^{1/2}\epsilon^{3/2}}{162}.$$

*Proof.* In the case where the "if condition" in Line 7 is triggered, we have

$$x^{\text{out}} \leftarrow \frac{1}{K_0 + 1 - \lfloor K/2 \rfloor} \sum_{k=\lfloor K/2 \rfloor}^{K_0} y^{(k)},$$

where in each iteration $k$ we have

$$\left\|\widehat{H}^{1/2}(y^{(k)} - x^{(0)})\right\| \leq 2\left\|\widehat{H}^{1/2}(x^{(k)} - x^{(0)})\right\| + \left\|\widehat{H}^{1/2}(x^{(k-1)} - x^{(0)})\right\| \leq 6\sqrt{3\delta p_{\max}}B,$$

which leads to

$$\left\|\widehat{H}^{1/2}(x^{\text{out}} - x^{(0)})\right\| \leq \frac{1}{K_0 + 1 - \lfloor K/2 \rfloor} \sum_{k=\lfloor K/2 \rfloor}^{K_0} \left\|\widehat{H}^{1/2}(y^{(k)} - x^{(0)})\right\| \leq 6\sqrt{3\delta p_{\max}}B.$$

Hence,

$$g(x^{\text{out}}) - g(x^{(0)}) = \frac{1}{2}(x^{\text{out}} - x^{(0)})^\top \nabla^2 f(x^{(0)})(x^{\text{out}} - x^{(0)})$$

$$\leq \frac{1}{2}(x^{\text{out}} - x^{(0)})^\top \widehat{H}(x^{\text{out}} - x^{(0)}) \leq 54\delta p_{\max}B^2 \leq \frac{6\delta\epsilon}{L_2}.$$

Meanwhile,

$$f(x^{\text{out}}) - g(x^{\text{out}}) \leq \frac{L_2}{6}\left\|x^{\text{out}} - x^{(0)}\right\|^3 \leq \frac{\epsilon^{3/2}}{6L_2^{1/2}},$$

by which we can conclude that

$$f(x^{\text{out}}) - f(x^{(0)}) \leq \frac{6\delta\epsilon}{L_2} + \frac{\epsilon^{3/2}}{6L_2^{1/2}}.$$

$\square$

# F    Analysis of Algorithm 2

**Lemma 1.** *Suppose $\epsilon \leq L_1^2/L_2$. In each iteration $t$ of Algorithm 2 before it terminates, we have*

$$f(x^{(t+1)}) - f(x^{(t)}) \leq -\tilde{p}^{-12}\sqrt{\epsilon^3/L_2}$$

*if $x^{(t+1)}$ is not $\epsilon$-critical for $f$, where we denote $\tilde{p} = \max\{\lceil\log(L_1/\tilde{\delta})\rceil, 16\}$.*

*Proof.* If the current $H$ satisfies $H \prec -2\tilde{\delta}I$, by Line 7 and Line 8 we have

$$f(x^{(t+1)}) - f(x^{(t)}) \leq \langle \nabla f(x^{(t)}), Rv \rangle + \frac{R^2}{2}v^\top \nabla^2 f(x^{(t)})v + \frac{L_2 R^3}{6}$$

$$\leq \frac{R^2}{2}v^\top \nabla^2 f(x^{(t)})v + \frac{\tilde{\delta}R^2}{6},$$

where

$$v^\top \nabla^2 f(x^{(t)})v = v^\top Hv + v^\top(\nabla^2 f(x^{(t)}) - H)v$$

$$\leq -2\tilde{\delta} + \left\|\nabla^2 f(x^{(t)}) - H\right\|$$

$$\leq -2\tilde{\delta} + \min\left\{\left\|\nabla^2 f(x^{(t)}) - \nabla^2 f(\bar{x})\right\| + \left\|\nabla^2 f(\bar{x}) - H\right\|, 2L_1\right\}$$

$$= -2\tilde{\delta} + \min\{L_2 R + \delta, 2L_1\} = -\tilde{\delta},$$

which leads to

$$f(x^{(t+1)}) - f(x^{(t)}) \le -\frac{1}{3}\tilde{\delta}R^2 \le -\frac{1}{3}L_2 R^3 \le -\frac{1}{\tilde{p}^{12}}\sqrt{\frac{\epsilon^3}{L_2}}.$$

Otherwise,

$$x^{(t+1)} = \texttt{Critical-or-Progress}\big(x^{(t)}, H, 2\tilde{\delta}, \epsilon, L_1, L_2\big),$$

which by Theorem 2 satisfies

$$f(x^{(t+1)}) - f(x^{(t)}) \le -\frac{1}{\tilde{p}^{12}}\sqrt{\frac{\epsilon^3}{L_2}}$$

if $\|\nabla f(x^{(t+1)})\| > \epsilon$. $\qquad\square$

**Lemma 33.** *Let* $0 < \epsilon \le \min\{L_1^2 L_2^{-1}, \Delta^{2/3} L_2^{1/3}\}$. *The output* $x^{\mathrm{out}}$ *of Algorithm 2 satisfies*

$$\|x^{\mathrm{out}} - x^{(0)}\| \le \frac{3\Delta}{\epsilon}\log^8\left(\frac{L_1}{c_\delta} + 16\right)$$

*and*

$$f(x^{\mathrm{out}}) - f(x^{(0)}) \le \frac{54 c_\delta \epsilon}{L_2}\log^8\left(\frac{L_1}{c_\delta} + 16\right) + \frac{\epsilon^{3/2}}{6L_2^{1/2}},$$

*where* $c_\delta := \min\{L_1, \delta + \Delta L_2/(n_H \epsilon)\}$.

*Proof.* Suppose Algorithm 2 terminates at the $\mathcal{T}$-th iteration. For any $t < \mathcal{T} - 1$, by Proposition 4 we have

$$f(x^{(t+1)}) - f(x^{(t)}) \le \frac{1}{\tilde{p}^{12}}\sqrt{\frac{\epsilon^3}{L_2}},$$

indicating that

$$\mathcal{T} \le \tilde{p}^{12}\Delta\sqrt{L_2/\epsilon^3}.$$

Since

$$\|x^{(t+1)} - x^{(t)}\| \le \frac{7}{3\tilde{p}^4}\sqrt{\frac{\epsilon}{L_2}}$$

in each iteration $t$ by (4) and Theorem 2, we have

$$\|x^{\mathrm{out}} - x^{(0)}\| = \|x^{(\mathcal{T})} - x^{(0)}\| \le 3\tilde{p}^8\Delta/\epsilon \le \frac{3\Delta}{\epsilon}\log^8\left(\frac{L_1}{c_\delta} + 16\right)$$

As for the function value change in last iteration, by Lemma 32, we have

$$f(x^{\mathrm{out}}) = f(x^{(\mathcal{T})}) \le f(x^{(\mathcal{T}-1)}) + \frac{54\tilde{\delta}\epsilon}{L_2} + \frac{L_2\epsilon^{3/2}}{6L_2^{1/2}}.$$

Summing over all the iterations, we can conclude that

$$f(x^{\mathrm{out}}) - f(x^{(0)}) \le \frac{18 c_\delta \epsilon}{L_2}\log^8\left(\frac{L_1}{c_\delta} + 16\right) + \frac{\epsilon^{3/2}}{6L_2^{1/2}}.$$

$\qquad\square$

# G  Analysis of Algorithm 3

**Lemma 34.** *Given $f := \mathbb{R}^d \to \mathbb{R}$ with $L_2$-Lipschitz Hessian, for any $x \in \mathbb{R}^d$ and any symmetric $H$ satisfying $\|H - \nabla^2 f(x)\| \le \delta$, denote*

$$y \leftarrow x - (\Pi_{\mathrm{large}} H \Pi_{\mathrm{large}})^\dagger \nabla f(x).$$

*If $\ell \ge \max\{24\Delta_x^{1/3} L_2^{2/3}, 2\delta\}$ for $\Delta_x = f(x) - \inf_{z \in \mathbb{R}^d} f(z)$, we have*

$$\|\Pi_{\mathrm{large}} \nabla f(y)\| \le 2\delta \sqrt{\frac{3\Delta_x}{\ell}} + \frac{6L_2 \Delta_x}{\ell}$$

*and*

$$\|\Pi_{\mathrm{small}} \nabla f(y)\| \le \|\Pi_{\mathrm{small}} \nabla f(x)\| + 2\delta \sqrt{\frac{3\Delta_x}{\ell}} + \frac{6L_2 \Delta_x}{\ell}.$$

*Proof.* Denote $u := y - x = -(\Pi_{\mathrm{large}} H \Pi_{\mathrm{large}})^\dagger \nabla f(x)$. We first show that $\|u\| \le \ell/L_2$. Assume the contrary, we have

$$f\left(x + \frac{\ell}{L_2} \cdot \frac{u}{\|u\|}\right) - f(x) \le -\frac{\ell}{2}\left(\frac{\ell}{L_2}\right)^2 + \frac{\delta}{2}\left(\frac{\ell}{L_2}\right)^2 + \frac{L_2}{6}\left(\frac{\ell}{L_2}\right)^3 \le -\frac{\ell^3}{12 L_2^2} \le -2\Delta_x,$$

contradiction. Then, we have

$$-\Delta_x \le f(y) - f(x) \le -\frac{1}{2} u^\top \nabla^2 f(x) u + \frac{1}{6} L_2 \|u\|^3 \le \frac{\ell \|u\|^2}{12}$$

which leads to $\|u\| \le 2\sqrt{3\Delta_x/\ell}$. Then, we have

$$\|\Pi_{\mathrm{large}} \nabla f(y)\| \le \|\Pi_{\mathrm{large}} \nabla f(x) - \Pi_{\mathrm{large}} H u\| + \|\Pi_{\mathrm{large}}(H - \nabla^2 f(x))u\| + \frac{1}{2} L_2 \|u\|^2$$

$$\le \delta\|u\| + \frac{1}{2} L_2 \|u\|^2 \le 2\delta\sqrt{\frac{3\Delta_x}{\ell}} + \frac{6L_2 \Delta_x}{\ell}.$$

Similarly, we have

$$\|\Pi_{\mathrm{small}} \nabla f(y)\| \le \|\Pi_{\mathrm{small}} \nabla f(x)\| + \|\Pi_{\mathrm{small}} H u\| + \|\Pi_{\mathrm{small}}(H - \nabla^2 f(x))u\| + \frac{1}{2} L_2 \|u\|^2$$

$$\le \|\Pi_{\mathrm{small}} \nabla f(x)\| + 2\delta\sqrt{\frac{3\Delta_x}{\ell}} + \frac{6L_2 \Delta_x}{\ell}.$$

$\square$

**Theorem 4.** *Let $\mathtt{Alg}(f_{\le L_1}, L_1, L_2, \delta, \Delta, \epsilon)$ be a procedure that, for any function $f_{\le L_1} : \mathbb{R}^d \to \mathbb{R}$ with $L_1$-Lipschitz gradient, $L_2$-Lipschitz Hessian, and $\Delta$-bounded suboptimality, uses*

- *$n_H$ queries to a $\delta$-approximate Hessian oracle for $f_{\le L_1}$, and,*
- *$n_g(L_1, L_2, \delta, \Delta, \epsilon)$ queries to a gradient oracle for $f_{\le L_1}$,*

*and returns an $\epsilon/2$-critical point $x^{\mathrm{out}}$ satisfying $\|x^{\mathrm{out}} - x^{(0)}\| \le R_{\mathrm{out}}$ and $f(x^{\mathrm{out}}) - \inf_z f(z) \le \Delta_{\mathrm{out}}$. Then, for any $f$ with $L_2$-Lipschitz Hessian and $\Delta$-bounded suboptimality, any $0 < \epsilon \le \min\{L_1^2 L_2^{-1}, \Delta^{2/3} L_2^{1/3}\}$, and any $\ell$ that satisfies*

$$\ell \ge \max\left\{\frac{800\Delta}{\epsilon^2}(L_2 R_{\mathrm{out}} + \delta)^2, \frac{48 L_2 \Delta_{\mathrm{out}}}{\epsilon}, 24\Delta_{\mathrm{out}}^{1/3} L_2^{2/3}, 2\delta\right\}. \tag{12}$$

*Algorithm 3 returns an $\epsilon$-critical point using*

- *$n_H + 1$ queries to a $\delta$-approximate Hessian oracle for $f$, and,*
- *$n_g(\ell, L_2, \delta, \Delta, \epsilon)$ queries to a gradient oracle for $f$.*

*Proof.* By Lemma 34 we have

$$\|\nabla f(y)\| \leq \|\Pi_{\text{large}}\nabla f(y)\| + \|\Pi_{\text{small}}\nabla f(y)\|$$

$$\leq \|\Pi_{\text{small}}\nabla f(x^{\text{out}}) + 4\|H - \nabla^2 f(x^{\text{out}})\|\sqrt{\frac{3\Delta_{\text{out}}}{\ell}} + \frac{12L_2\Delta_{\text{out}}}{\ell}$$

$$\leq \frac{\epsilon}{2} + 4(\delta + L_2 R_{\text{out}})\sqrt{\frac{3\Delta_{\text{out}}}{\ell}} + \frac{12L_2\Delta_{\text{out}}}{\ell} \leq \epsilon$$

given the choice of $\ell$ in (12). $\qquad\square$

**Corollary 2.** *Let $f\colon \mathbb{R}^d \mapsto \mathbb{R}$ $L_2$-Lipschitz Hessian. Given any $x^{(0)} \in \mathbb{R}^d$ with $\Delta$-bounded suboptimality with respect to $f$, any positive integer $n_H \geq 1$, and $0 < \epsilon \leq \Delta^{2/3}L_2^{1/3}$, Algorithm 3 using Algorithm 2 as the subroutine* `Alg` *outputs an $\epsilon$-critical point of $f$ with at most $n_H$ queries to a $\delta$-approximate Hessian oracle and*

$$O\left(\frac{\Delta L_2^{1/4}}{\epsilon^{7/4}}\sqrt{\delta + \frac{\Delta L_2}{n_H\epsilon}} \cdot \text{poly}\log\left(\frac{1}{c_\delta}\left(\frac{L_2^2\Delta^3}{\epsilon^4} + \frac{\Delta\delta^2}{\epsilon^2} + \delta\right)\right)\right)$$

*queries to a gradient oracle for $f$.*

*Proof.* Set

$$\Delta_{\text{out}} = 54\left(\Delta + \frac{\delta\epsilon}{L_2}\right)\log^9\left(\frac{\hat{\ell}}{c_\delta} + 16\right) + \frac{\epsilon^{3/2}}{6L_2^{1/2}},$$

$$\ell := \hat{\ell}\log^{19}\left(\hat{\ell}/c_\delta\right), \tag{41}$$

$$R_{\text{out}} = \frac{3\Delta}{\epsilon}\log^9\left(\frac{\hat{\ell}}{c_\delta} + 16\right),$$

where

$$\hat{\ell} := \max\left\{\frac{800\Delta}{\epsilon^2}\left(\frac{3L_2\Delta}{\epsilon} + \delta\right)^2, 2\delta\right\} = O\left(\frac{L_2^2\Delta^3}{\epsilon^4} + \frac{\Delta\delta^2}{\epsilon^2} + \delta\right). \tag{42}$$

Observe that the above parameters satisfy

$$\ell \geq \max\left\{\frac{800\Delta}{\epsilon^2}(L_2 R_{\text{out}} + \delta)^2, \frac{48L_2\Delta_{\text{out}}}{\epsilon}, 24\Delta_{\text{out}}^{1/3}L_2^{2/3}, 2\delta\right\},$$

which gives

$$\left\|x^{\text{out}} - x^{(0)}\right\| \leq R_{\text{out}}, \quad f(x^{\text{out}}) - \inf_{x\in\mathbb{R}^d} f(x) \leq \Delta_{\text{out}}$$

by Lemma 33, where $x^{\text{out}}$ is the output of `Restarted-Approx-Hessian-AGD` when applied to $f_{\leq\ell}$. Then by Theorem 4, Algorithm 3 outputs an $\epsilon$-critical point. Since `Restarted-Approx-Hessian-AGD` starts by querying the $\delta$-approximate Hessian oracle at $x^{(0)}$, the query in Line 3 can be reused, and there are a total of at most $n_H$ queries to a $\delta$-approximate Hessian oracle and

$$\frac{2\Delta L_2^{1/4}}{\epsilon^{7/4}}\sqrt{\min\left\{\ell, \delta + \frac{\Delta L_2}{n_H\epsilon}\right\}} \cdot \log^{18}\left(\frac{\hat{\ell}}{c_\delta} + 16\right)$$

$$= O\left(\frac{\Delta L_2^{1/4}}{\epsilon^{7/4}}\sqrt{\delta + \frac{\Delta L_2}{n_H\epsilon}} \cdot \text{poly}\log\left(\frac{1}{c_\delta}\left(\frac{L_2^2\Delta^3}{\epsilon^4} + \frac{\Delta\delta^2}{\epsilon^2} + \delta\right)\right)\right)$$

queries to a gradient oracle. $\qquad\square$

