# OpenReview forum: "Balancing Gradient and Hessian Queries in Non-Convex Optimization"
_NeurIPS.cc/2025/Conference — NeurIPS 2025 poster_

### Official Review · Reviewer_ZSvX · 2025-06-17

**Clarity:** 3
**Significance:** 3
**Originality:** 3
**Rating:** 4
**Confidence:** 4

**Summary:**

This paper studies non-convex minimization and offers new trade-offs between the number of gradient and Hessian computations.

The previous methods use $\mathcal{O}(\epsilon^{-1/4})$ gradient oracles in the subroutines to decrease the function value by a factor of $\Omega(\epsilon^{3/2})$, which leads to the $\mathcal{O}(\epsilon^{-7/4})$ queries to the gradient oracle.
This paper enhances the subroutine to $\mathcal{O}(\delta^{1/2}\epsilon^{-1/4})$ by incorporating in $\delta$-inexact Hessian oracle, which allows the reuse of the Hessian.
By properly tuning $\delta$, the proposed method can achieves better computational complexity than the lazy Hessian (Doikov et. al, 2023).

I find the theoretical results presented in this paper are interesting and novel. However, this paper lacks of experiments to check the efficiency of the proposed method, while it requires additional computational cost of the Hessian and its decomposition.

**Questions:**

1. Theorem 2 requires the condition such that $H \preceq L_1 I$ (line 156), I am wondering how can this be satisfied in Algorithm 2 when calling the Critical-or-Progress sub-routine?

**Ethical Concerns:**

["NO or VERY MINOR ethics concerns only"]

**Final Justification:**

I decided to keep my score as it is, which is positive.

**Quality:**

3

**Strengths And Weaknesses:**

**Strengths:** the theoretical results obtained by this paper is novel and significant in non-convex optimization. The new trade-off between the gradient and Hessian oracles largely improves the prior-art.

**Weakness:** the paper lacks of experimental part to verify the efficiency of the proposed methods and to validate the proper trade-off between the gradient and Hessians.

---

> ### Author Rebuttal · Authors · 2025-07-31
>
> Thank you for your positive feedback and detailed comments!
>
> Regarding your question on the efficiency of the proposed methods in practice, while this aspect is not the primary focus of our work, we agree that it is an interesting question. We hope our results open the door for such exploration. As a possible direction, we note that our algorithm only requires matrix-vector product access to $\hat{H}^{-1}$, which could admit more efficient implementations.
>
> Regarding your question on the condition $H \preceq L_1 I$, thank you for catching this minor issue: It should instead be $H \preceq 2L_1 I$. The corrected inequality is guaranteed by the gradient Lipschitzness of $f$ and the condition that $\delta\leq L_1$.

---

> > ### Comment · Reviewer_ZSvX · 2025-08-05
> >
> > Thank you for your response, which has addressed some of my questions. I'll keep my score as it is.

---

### Official Review · Reviewer_tXmx · 2025-07-02

**Clarity:** 2
**Significance:** 2
**Originality:** 3
**Rating:** 4
**Confidence:** 2

**Summary:**

The paper studies trading off gradient and Hessian computations for a faster convergence rate for computing the critical point in non-convex optimization. In particular, the paper develops a method that achieves the best known dimension-independent rate and improves upon prior works in finite dimensions.

**Questions:**

1. How easy is it to compute $\hat{H}$ and $\hat{H}^{-1}$ in Algorithm 1 for high-dimensional problems? Similarly, the Newton step in Algorithm 3?

**Ethical Concerns:**

["NO or VERY MINOR ethics concerns only"]

**Final Justification:**

The authors' response addresses my concerns and I have no further questions.

**Limitations:**

See Weaknesses.

**Paper Formatting Concerns:**

I do not notice any major formatting issues.

**Quality:**

3

**Strengths And Weaknesses:**

Strength:

1. The paper has technical contributions and significance. The proofs are based on fine-grained analysis, coupled with delicate algorithm design, e.g., the choice of parameters in Eq. (4). Moreover, the theories possess sufficient generality to cover known results for gradient methods.

2. The result is strong in improving the dimensional dependence compared to prior works under the same condition (Corollary 1, Table 1).

3. Balancing the use of gradient and Hessian information for better overall cost (measured in gradient queries) is interesting.


Weaknesses:

1. The paper only considers gradient queries and does not provide computational complexity analysis. In particular, some steps might take a significantly larger amount of time to compute, e.g., transformed Hessian $\hat{H}$ in Eq. (2). It would be clearer to provide overall computational complexity in comparison with pure gradient methods and second-order methods.

2. The writing is very technical. While this is not necessarily a weakness, it would be better to provide more intuition illustrating why a faster rate can be obtained.

---

> ### Author Rebuttal · Authors · 2025-07-31
>
> Thank you for your positive feedback and detailed comments!
>
> Regarding your question on the computational complexity of our algorithm, while this aspect is not the primary focus of our work, we agree that it is an interesting question. Although certain components of our algorithm, such as the eigen-decomposition step, may appear more involved than those in the classical Newton method, we note that these operations can also be implemented in $O(d^{\omega})$ time in the worst case, where $\omega$ denotes the matrix multiplication exponent [1]. This matches the cost of solving the linear system and the per-iteration cost of Newton method.
>
> Based on this, the overall computational complexity of our algorithm under the parameter setting of Corollary 1 is $\tilde{O}(d^{7/3}/\epsilon^{3/2})$. For comparison, the restarted AGD algorithm [2] requires $O(d/\epsilon^{7/4})$ time, while the Newton method requires $O(d^2/\epsilon^{3/2})$. Thus, our algorithm improves upon the first order method when $\epsilon=O(d^{-16/3})$, and exceeds the complexity of the Newton method by only a factor of $d^{1/3}$. We may add more discussion on this in our final version.
>
> We appreciate your thoughtful comments on the presentation of the paper. In terms of the accessibility of the paper, this is something we care about greatly and would like to improve if we can. In the submission, in Section 1.1, we included a detailed comparison with prior work along with a table that conceptualizes our contributions, in Section 2 we provided an overview of our approach, and we provided a concluding section that summarizes our algorithms and main insights. Nevertheless, we do recognize that the paper discusses trade-offs across multiple oracle models and four parameters, which inherently leads to technical complexity. If there are specific parts of the paper that you believe could benefit from clearer presentation or if you have ideas for helping the paper’s accessibility, we would be eager to hear them.
>
> In particular, for your question on the intuition regarding the faster rate we obtained, the key insight behind the improved rate lies in Algorithm 1, which is a variant of accelerated gradient descent (AGD) in the Hessian norm. In classical AGD, the step size is chosen to be as large as possible while still respecting the smoothness condition. However, if we have access to local second-order curvature information, it becomes possible to take larger steps along directions with small curvature, thereby accelerating convergence. Algorithm 1 formalizes this intuition. If you have further questions about other aspects of our work, we would be happy to address them as well.
>
> [1]. James Demmel Ioana Dumitriu, and Olga Holtz. Fast linear algebra is stable. Numerische Mathematik 2007.
>
> [2]. Huan Li and Zhouchen Lin. Restarted Nonconvex Accelerated Gradient Descent: No More Polylogarithmic Factor in the O(epsilon^(-7/4)) Complexity. ICML 2023

---

> > ### Comment · Reviewer_tXmx · 2025-08-04
> >
> > Thank you for your response. They address my concerns and I have no further questions. I'll keep my score recommending for acceptance.

---

### Official Review · Reviewer_tspV · 2025-07-03

**Clarity:** 3
**Significance:** 4
**Originality:** 3
**Rating:** 5
**Confidence:** 3

**Summary:**

This paper studies the problem of finding a first-order stationary point of a smooth non-convex function using access to a gradient oracle and an approximate Hessian oracle. The core idea is to apply the restarted nonconvex accelerated gradient method of Li & Lin (2023), preconditioned by an approximate Hessian. Moreover, the Hessian estimate is updated in a lazy manner only when the distance between the current iterate and the last reference point exceeds a predefined threshold. This threshold controls the trade-off between gradient and (approximate) Hessian queries: a smaller threshold leads to more frequent Hessian updates and also fewer gradient steps due to more accurate second-order information. As a corollary, the authors derive a new dimension-dependent complexity bound using only gradient access, as well as a bound that requires only a single Hessian query.

**Questions:**

## Clarifying Questions
- The construction of the modified approximate Hessian estimate $\hat{H}$ in (3) appears somewhat ad hoc. In particular, the origin of the logarithmic terms is unclear. While the authors mention that these terms introduce spectral gaps and facilitate the use of matrix perturbation theory, a more detailed explanation or motivation for this specific form would be helpful.
- Regarding the results in Corollary 1, since each approximate Hessian computation requires $2d$ gradient queries, I believe the complexity bound should be $\tilde{O}(d^{1/3}L_2^{1/2}\Delta \epsilon^{-3/2} +d)$. Alternatively, if this cost is not counted explicitly, then the bound applies only when $\epsilon \leq L\_2^{1/3} \Delta^{2/3} d^{-4/9}$.

**Ethical Concerns:**

["NO or VERY MINOR ethics concerns only"]

**Final Justification:**

The authors indicated that they will include a detailed discussion of the computational complexity in the revision, and they have also clarified my questions regarding the logarithmic terms in the approximate Hessian estimate and the dependence on $d$ in their complexity bound. Accordingly, I maintain my initial score for the paper.

**Limitations:**

Please refer to the "Weaknesses" section.

**Paper Formatting Concerns:**

The format of the paper appears proper to me.

**Quality:**

3

**Strengths And Weaknesses:**

## Strengths
- This paper combines several techniques from nonconvex optimization—including restarted nonconvex accelerated gradient methods, preconditioning, and lazy Hessian updates—in a non-trivial and interesting way. The result is an algorithm that explicitly balances the trade-off between gradient and Hessian computations.
- Notably, the proposed method achieves a new complexity bound of $\tilde{O}(d^{1/3}\epsilon^{-3/2}+d)$ when the function has $L_2$-Lipschitz Hessian and only gradient access. This improves the dimension dependence over Doikov et al. (2023) and yields a better dependence on $\epsilon$ than Li & Lin (2023) when the dimension is sufficiently small.

## Weaknesses
- The paper currently lacks a detailed discussion on the computational cost. Specifically, constructing the modified approximate Hessian matrix $\hat{H}$ requires full eigendecomposition, which will incur a cost of $O(d^3)$.

---

> ### Author Rebuttal · Authors · 2025-07-31
>
> Thank you for your positive feedback and detailed comments!
>
> Regarding your question regarding the computational complexity of our algorithm, while this aspect is not the primary focus of our work, we agree that it is an interesting question and thank you for asking. Although certain components of our algorithm, such as the eigendecomposition step, may appear more involved than those in the classical Newton method, we note that these operations can also be implemented in $O(d^{\omega})$ time in the worst case, where $\omega$ denotes the matrix multiplication exponent [1]. This matches the worst-case cost of solving a dense linear system and the worst-case per-iteration cost of Newton method.
>
> Regarding your question on the origin of the logarithmic terms in the approximate Hessian estimate, we first note the following intuition: our rescaled Hessian AGD framework is stable under constant-factor perturbations in the eigenvalues of $H$. That is, changing an eigenvalue $\lambda$ to $2\lambda$ or $\lambda/2$ does not affect the asymptotic convergence. To formalize this, our analysis partitions the eigenspace of $H$ into $O(\log(L_1/\delta))$ groups based on eigenvalue magnitude, as detailed in Appendix C. Within each group, eigenvalues are either $\delta$-close to zero or differ by at most a constant factor. This grouping is the source of the logarithmic term.
>
> As for how this appears in the algorithm, our analysis requires that the spectrum of the normalized matrix $\hat{H}^{-1/2} H \hat{H}^{-1/2}$ closely approximates that of $H$. In particular, the relative ordering and grouping of eigenvalues must be preserved—eigenvalues that belong to the same group in $H$ must remain in the same group after normalization. The explicit truncation steps based on this logarithmic grouping ensure that this spectral structure is maintained. We agree with you that finding a simpler construction of $\hat{H}$ that suffices is a wonderful question for future research.
>
> Regarding your question on the asymptotic $d$-dependence in the bound of Corollary 1, we thank you for pointing it out and will correct it in the final version.
>
> [1]. James Demmel Ioana Dumitriu, and Olga Holtz. Fast linear algebra is stable. Numerische Mathematik 2007.

---

> > ### Comment · Reviewer_tspV · 2025-08-05
> >
> > Thank you for your detailed responses, and I have no further questions at this point. I would encourage the authors to discuss the computational complexity of their proposed method more explicitly in the revision, which is also mentioned by several other reviewers.

---

> > > ### Author Response · Authors · 2025-08-07
> > >
> > > Thank you very much for your reply, your consideration, and this reminder. Indeed, we plan to discuss per-iteration computational costs in greater detail in the next version.

---

### Official Review · Reviewer_BCZS · 2025-07-03

**Clarity:** 4
**Significance:** 4
**Originality:** 3
**Rating:** 5
**Confidence:** 5

**Summary:**

This paper develops novel optimization algorithms that achieve improved trade-offs between gradient and Hessian oracle queries for computing ε-critical points of nonconvex functions. The key contributions are:

1. ​**Main Trade-off Result (Theorem 1):​**​ For twice-differentiable functions with L₂-Lipschitz Hessians and initial Δ-bounded suboptimality, the algorithm computes an ε-critical point using nₖ Hessian queries and $\tilde{O}(\Delta L_2^{1/4} n_H^{-1/2} \epsilon^{-9/4})$ gradient queries with δ-approximate Hessians. This generalizes prior gradient-only methods.

2. ​**Dimension-Dependent Improvement (Corollary 1):​**​ In d-dimensional space, the algorithm achieves $\tilde{O}(d^{1/3} L_2^{1/2} \Delta \epsilon^{-3/2})$ gradient-only complexity via finite differencing, improving over prior $\tilde{O}(\sqrt{d})$ results when $\epsilon$ is sufficiently small.

3. ​**Single-Hessian-Query Complexity:​**​ When limited to just one exact Hessian query, the method attains $\tilde{O}(\Delta^{3/2} L_2^{3/4} \epsilon^{-9/4})$ gradient queries, improving previous bounds by a polynomial factor.

4. ​**Technical Innovations:​**​ The approach introduces (a) a restart mechanism with adaptive Hessian updates (Algorithm 2), (b) a reduction technique to handle functions without gradient Lipschitz continuity (Algorithm 3), and (c) careful analysis of accelerated gradient descent under approximate Hessian-induced norms.

These results advance nonconvex optimization by unifying and extending prior state-of-the-art complexities while enabling new computation trade-offs between oracle types.

**Questions:**

**1. On the tightness of dimension-dependent complexity (Corollary 1):​**​
The $\tilde{O}(d^{1/3})$ gradient complexity in Corollary 1 improves prior $\tilde{O}(\sqrt{d})$ bounds but still has a polynomial dimension dependence. Could you comment on whether this exponent is theoretically tight? Specifically:
- Is there a matching lower bound for $d^{1/3}$ in this Hessian-approximation-via-finite-differencing setting?
- Does the exponent $1/3$ stem from fundamental limitations or current proof techniques?
*Actionable request*: Provide intuition for the $d^{1/3}$ term (e.g., whether it arises from the $\min(c_\delta)$ trade-off in Theorem 1 or finite differencing error propagation). A clear explanation would strengthen significance; conversely, failure to clarify may weaken perceived novelty.

​**2. Handling stochastic or inexact Hessian approximations:​**​
Theorem 1 assumes access to a deterministic $\delta$-approximate Hessian oracle. Many practical implementations (e.g., via sketching or subsampling) yield *stochastic* approximations with variance $\sigma^2$.
- Can your analysis accommodate stochastic Hessians with $\mathbb{E}[\|H_x - \nabla^2 f(x)\|^2] \leq \sigma^2$?
- If so, how would this affect the complexity (e.g., would terms like $\Delta \sigma / \epsilon$ appear)?


**3. Practicality of spectral computations for $\widehat{H}$:​**​
Algorithm 1 requires spectral decomposition of $H_{x^{(0)}}$ to construct $\widehat{H}$ (Eq. 2), which costs $O(d^3)$ and negates low-query benefits in high dimensions.
- Can $\widehat{H}$ be applied implicitly (e.g., via Krylov methods) without full decomposition?
- Does your analysis support approximate eigenvectors (e.g., from power iteration)?

**Ethical Concerns:**

["NO or VERY MINOR ethics concerns only"]

**Final Justification:**

We maintain our **Accept (5)** recommendation. The authors provided comprehensive and technically sound responses to all raised points, convincingly clarifying the origin of the $d^{1/3}$ dimension term in Corollary 1 (as a trade-off artifact rather than a fundamental limit) and outlining feasible extensions to stochastic Hessian settings. Critically, issues discussed in the rebuttal (e.g., tighter lower bounds and non-deterministic Hessian approximations) were highlighted by the authors themselves as challenging open problems worthy of standalone future work—not as resolvable weaknesses diminishing the current theoretical contribution. This paper delivers rigorously proven algorithmic advances in non-convex optimization with novel gradient/Hessian query trade-offs, meeting NeurIPS standards for significance and originality.

**Limitations:**

See weaknesses.

**Paper Formatting Concerns:**

No major formatting issues.

**Quality:**

4

**Strengths And Weaknesses:**

## Strengths

- ​**Originality:​**​ The paper introduces a fundamentally new perspective on balancing gradient and Hessian queries in non-convex optimization, an underexplored dimension in complexity analysis. The approach of using δ-approximate Hessians to define custom norms (Ĥ in Eq. 2) and the reduction technique for unbounded gradient Lipschitz (Algorithm 3) represent novel algorithmic innovations not seen in prior work.

- ​**Technical Quality:​**​ The theoretical analysis is rigorous and comprehensive. The authors provide full proofs for all claims (Appendix A–F), leveraging advanced tools like Davis-Kahan theorem for spectral perturbation analysis (Lemma 7) and carefully structured induction arguments. The main results (Theorems 1–4) strictly generalize prior bounds (e.g., recovering [28] when $\theta = L_1$).

- ​**Significance:​**​ The results meaningfully advance non-convex optimization theory by:
  (i) Establishing the first trade-off rates between gradient/Hessian queries (Theorem 1),
  (ii) Improving dimension-dependent complexity to $\tilde{O}(d^{1/3})$ (Corollary 1), beating the prior $\tilde{O}(\sqrt{d})$ barrier for $d ≥ 7$,
  (iii) Enabling non-trivial critical point computation with only one Hessian query – a practical scenario where the paper shows $\tilde{O}(\epsilon^{-9/4}) $ gradient complexity, polynomially better than prior arts.

- ​**Clarity:​**​ The paper is exceptionally well-structured. The abstract and introduction clearly contextualize contributions against 15+ cited works (Table 1 is particularly effective). Algorithm descriptions are precise, and §2 provides excellent high-level intuition before technical details. Notation is consistent and well-defined throughout.

## Weaknesses

- ​**Practical Limitations:​**​ While theoretically novel, the algorithms rely on expensive operations like spectral decomposition of $\tilde{H}$ (Eq. 2) and may incur large constant factors hidden in $\tilde{O}(\cdot)$. The logarithmic dependencies on problem parameters (e.g., $\log(L_1/\theta)$) could be prohibitive in real-world implementations, though this is common in theoretical optimization.

- ​**Optimality Questions:​**​ While the paper improves upper bounds, it doesn't establish matching lower bounds for the new trade-off regimes. The relationship to the known $\Omega(\epsilon^{-12/7}$ gradient-only lower bound [11] is discussed, but Hessian-query lower bounds remain open.

- ​**Scope Constraints:​**​ The analysis assumes exact access to a $\delta$-approximate Hessian oracle. In practice, such oracles might require auxiliary computations (e.g., via finite differencing in Corollary 1), but the complexity of implementing these oracles isn't incorporated into the main trade-off results.

---

> ### Author Rebuttal · Authors · 2025-07-31
>
> Thank you for your positive feedback and detailed comments!
>
> Regarding your comment on the practicality of our algorithm, particularly the large constants and logarithmic dependencies in the query complexity, we want to kindly note that our focus in this paper was on worst-case theoretical query complexity and showing how our improved query complexities are obtainable. We agree it is an interesting direction for future work to improve these dependencies and the practicality of our method. We hope our results open the door for such exploration.
>
> Regarding your question on the approximate Hessian oracle and its implementation, we would like to kindly clarify that each column of the approximate Hessian can be obtained using two queries to a gradient oracle via finite differences (even if the gradients are only computed approximately). As a result, the overall implementation cost is $2d$ gradient queries. This cost is implicitly reflected in the trade-off presented in Corollary 1. We will include a more detailed discussion of this point in the final version.
>
> Regarding your question on whether the $\tilde{O}(d^{1/3})$ dependence is tight, we agree that this is an excellent question. To the best of our knowledge, there is no known matching lower bound. This term arises from the trade-off between gradient queries and Hessian queries in Theorem 1. Specifically, we require $n_H \cdot d$ gradient queries to construct the $n_H$ approximate Hessians, and $O(n_H^{-1/2}\epsilon^{-9/4})$ gradient queries to perform the AGD steps in the approximate Hessian norm. Balancing these two terms yields a total query complexity of $O(d^{1/3}\epsilon^{-3/2})$ when choosing $n_H = \Theta(d^{-2/3} \epsilon^{-3/2})$. Thus, the $\tilde{O}(d^{1/3})$ term reflects a limitation of our current analysis rather than a fundamental limitation. We will expand on this discussion around Lines 274–277 in the final version.
>
> Regarding your question regarding lower bounds for the new trade-off regimes, we would like to kindly note that dimension-dependent lower bounds for critical point computation are not completely understood. To the best of our knowledge, existing dimension dependent lower bounds are limited to the low-dimensional setting and do not assume Hessian Lipschitzness of the function. Moreover, establishing tight lower bounds in our setting would require improving the current $\Omega(\epsilon^{-12/7})$ lower bound for gradient computations for a function with Lipschitz gradient and Hessian, which has remained open for several years. For this reason, we view the development of tight lower bounds in our regime as an independent and interesting open problem. We will include a more detailed discussion of lower bounds in the low-dimensional and Hessian-only settings in the final version.
>
> Regarding your question on inexact Hessian approximations. While our current algorithm is presented under the assumption that the Hessian approximation is obtained deterministically, the framework naturally extends to settings where the approximation is computed via non-deterministic means, including stochastic oracles. A straightforward approach in the stochastic setting is to compute a mini-batch average of stochastic Hessians whenever a Hessian estimate is needed, ensuring that the approximation error is bounded by $\tilde{\delta}$, as defined in Equation (8). This would introduce terms such as $\sigma / \tilde{\delta}$ in the complexity. We view it as an interesting open problem to explore whether improved bounds can be obtained in this setting using more advanced techniques.
>
> Regarding your question regarding the practicality and computational complexity of spectral computations, while this aspect is not the primary focus of our work, we agree that it is an interesting question and thank you for asking. Although certain components of our algorithm, such as the eigendecomposition step, may appear more involved than those in the classical Newton method, we note that these operations can also be implemented in $O(d^{\omega})$ time in the worst case, where $\omega$ denotes the matrix multiplication exponent [1]. This matches the worst-case cost of solving a dense linear system and the worst-case per-iteration cost of Newton method.  We agree with you that it is an interesting open question to develop an algorithm with the same theoretical guarantee and is more practical to implement. Notably, our algorithm only requires matrix-vector product access to $\hat{H}^{-1}$, which suggests potential for more efficient implementations. While directly applying Krylov subspace methods in our setting may be nontrivial, we hope our results open the door to practical variants of our approach and worth further exploration.
>
> [1]. James Demmel, Ioana Dumitriu, and Olga Holtz. Fast linear algebra is stable. Numerische Mathematik 2007.

---

> > ### Comment · Reviewer_BCZS · 2025-08-05
> >
> > Thank you for the thorough and thoughtful rebuttal addressing all points raised in the review. The clarifications on the origin of the $d^{1/3}$ dependency in Corollary 1 – specifically, your explanation that it stems from balancing finite-differencing costs and accelerated gradient steps under the Hessian-induced norm – satisfactorily resolves this question for our evaluation. Your commitment to expanding the discussion of lower bounds and implementation costs in the final version is appreciated. We also note the clear pathways outlined for extending this work to stochastic settings. Given the substantial theoretical contributions as stated, and with no further technical concerns requiring clarification, we maintain our original 5: Accept rating.

---

### Comment · Area_Chair_hxww · 2025-08-05

Dear Reviewers,

Thank you again for your time and efforts in reviewing papers for NeurIPS 2025.

I am writing to remind you that **active participation in the author-reviewer discussion phase is mandatory**. According to the guidelines from the NeurIPS program chairs, reviewers are **required to engage directly with the authors in the discussion thread**, especially in response to their rebuttals.

Please note the following important policy:

- Simply reading the rebuttal or internally considering it is **not sufficient** -- reviewers must **post at least one message to the authors**, even if it is only to confirm that their concerns were resolved. If they have not been addressed, please explain why.

- **Acknowledging the rebuttal without any engagement with the authors will be considered insufficient**. I am obligated to flag such cases using the *InsufficientReview* mechanism, which may **impact future reviewing invitations and result in desk rejection of your own submissions**.

If you have not yet responded to the authors in the discussion thread, I kindly ask you to do so **as soon as possible**, and **no later than August 8, 11:59pm AoE**.

Please don't hesitate to reach out to me if you have any questions or concerns.

Best regards,

AC

---

### Decision · Program_Chairs · 2025-09-17

**Decision:**

Accept (poster)

**Comment:**

This paper studies the problem of finding an $\varepsilon$-critical point of a non-convex function with Lipschitz continuous gradient and Hessian. The authors propose a new method that combines restarted Accelerated Gradient Descent with preconditioning and lazy Hessian updates. Theoretical analysis establishes new complexity bounds that generalize existing gradient-only results and achieve improved dimension dependence through finite-difference Hessian estimation.

All reviewers agreed that the theoretical contributions are novel and significant for the literature on non-convex optimization. Among the weaknesses noted, the most important is the requirement to compute the spectral decomposition of $\tilde H$, which should be explicitly acknowledged as a limitation (e.g., in the Conclusion section alongside other limitations). Another weakness is the absence of numerical experiments.

Overall, the strengths of this work outweigh its weaknesses. Both the reviewers and I believe that the paper makes a valuable theoretical contribution and should be accepted to NeurIPS.